



# Advances in CALIPSO (IIR) cirrus cloud property retrievals – Part 2: Global estimates of the fraction of cirrus clouds affected by homogeneous ice nucleation

David L. Mitchell[1] and Anne Garnier[2]

[1]Desert Research Institute, Reno, NV 89512-1095, USA
[2][RSES]Analytical Mechanics Associates, Hampton, VA 23666, USA

*Correspondence to*: David L. Mitchell (David.Mitchell@dri.edu)

**Abstract.** Cirrus clouds can form through two ice nucleation pathways (homo- and heterogeneous ice nucleation; henceforth hom and het) that result in very different cloud physical and radiative properties. While important to the climate system, they are poorly understood due to lack of knowledge on the relative roles of het and hom. This study differs from earlier relevant studies by estimating the relative radiative contribution of hom-affected cirrus clouds. Here, we employ new global retrievals (described in Part 1) of cirrus cloud ice particle number concentration, effective diameter ($D_e$), ice water content, shortwave extinction coefficient ($\alpha_{ext}$), optical depth ($\tau$), and cloud radiative temperature based on Imaging Infrared Radiometer (IIR) and CALIOP (Cloud and Aerosol Lidar with Orthogonal Polarization) co-located observations onboard CALIPSO (Cloud-Aerosol Lidar and Infrared Pathfinder Satellite Observation). Transition from het only to hom affected regimes are identified using $\alpha_{ext}$ and $D_e$. Over oceans outside the tropics in winter, the zonal fraction of hom affected cirrus generally ranges between 20% and 35%, with comparable contributions from in situ and liquid origin cirrus. Using $\tau$ distributions to establish a proxy for cloud net radiative effect (CRE), the $\tau$-weighted fraction for hom affected cirrus over oceans outside the tropics during winter was > 50%, indicating that hom cirrus play an important role in climate. Moreover, the climate intervention method known as cirrus cloud thinning could be an effective cooling method at high latitudes based on this $\tau$-weighted hom fraction. A conceptual model of cirrus cloud characterization is proposed from these retrievals.

## 1 Introduction

Clouds are important due to their critical role in the Earth's radiation budget (e.g., Kiehl and Trenberth, 1997) and need to be realistically predicted in climate models in terms of their coverage, altitude, temperature, and optical thickness or depth to accurately predict climate change (e.g., Stephens, 2005). The microphysical and radiative properties of cirrus clouds are poorly understood in large part due to two pathways or mechanisms through which the cirrus clouds form: heterogeneous ice nucleation (henceforth het) and homogeneous ice nucleation (henceforth hom). Since het activates before hom begins, this can result in two types of cirrus clouds: (1) cirrus formed through het (i.e., het cirrus) and (2) cirrus formed through a combination of het and hom (i.e., hom cirrus). Since hom generally results in relatively high concentrations of ice crystals



(which can in turn affect their sizes and the cloud ice water content) while het produces relatively few, the microphysical and radiative properties of het and hom cirrus could strongly differ (Barahona and Nenes, 2009; Mitchell et al., 2018; henceforth M2018).

Cirrus clouds have also been classified in terms of their origin as (1) in situ cirrus and (2) liquid origin cirrus (e.g., Krämer et al., 2016; Luebke et al., 2016; Dekoutsidis et al., 2023), with liquid origin cirrus associated with cloudy air advected from

lower levels (T > 235 K) that often contains liquid cloud droplets. When these droplets ascend past the 235 K isotherm, they immediately freeze. This occurs in anvil cirrus and often occurs in frontal systems where a "conveyor belt" of condensate is advected from warmer levels into cirrus clouds. In situ cirrus clouds are not subject to condensate advection from levels below the 235 K isotherm, and they form and grow within an ice cloud only environment. Using aircraft measurements from many field campaigns, Luebke et al. (2016) found that cirrus ice water content IWC was higher in liquid origin cirrus while

differences between ice particle number concentration $N_i$ and $D_m$ were similar for a given altitude, where $D_m$ corresponds to the ice particle size distribution (PSD) bin-size where the maximum amount of ice mass is located. This study also utilizes this cirrus cloud classification, comparing relative differences in $N_i$, effective diameter $D_e$, and IWC regarding in situ and liquid origin cirrus clouds as defined by this study.

Some studies using global climate models (GCMs) have predicted the role of hom relative to het. Using the Community

Atmosphere Model version 5 (CAM5) with the Liu and Penner (2005) ice nucleation scheme, Gettelman et al. (2010), Liu et al. (2012), Gettelman et al. (2012), and Penner et al. (2015) found that het prevailed in the Northern Hemisphere (NH) outside the tropics, with hom prevailing in the Southern Hemisphere (SH) and in the tropics. Using the ECHAM6-HAM2 GCM, Gasparini and Lohmann (2016) found hom was only relevant near the tropopause over mountainous terrain in the NH and SH outside the tropics. Muench and Lohmann (2020) used this same GCM but without the treatment for orographic gravity waves

used in Gasparini and Lohmann (2016). Their results regarding hom and het were similar to the CAM5 studies, with het prevailing over hom in the NH outside the tropics. However, hom resulting from cloud droplet freezing dominated ice production in the lower part of cirrus clouds at all latitudes. The dominance of het relative to hom in the NH outside the tropics is common to all these studies (at least in the upper cirrus cloud), and this is predicted from the higher mineral dust concentrations in the NH, which are higher due to the much greater land area. This prediction was recently validated by Froyd

et al. (2022) where global in situ measurements of upper troposphere (UT) mineral dust concentration initialized a detailed cirrus-formation model that predicted the relative contributions of het and hom cirrus during four field campaigns that each circled the world from pole-to-pole. Finally, Barahona et al. (2017) simulated hom and het in cirrus clouds using a GCM at 100 km horizontal resolution. However, standard deviations in vertical velocities (w), $\sigma_w$, were calculated at 7 km horizontal resolution in a separate simulation to drive ice nucleation processes in the lower resolution simulation. Results show that the

frequency of hom cirrus clouds was highest in the tropics, intermediate in the SH and lowest in the NH, with hom cirrus highest during winter outside the tropics in both hemispheres due to colder temperatures. However, at high latitudes (especially in the Arctic), the fraction of hom cirrus was relatively low.



It is possible that hom plays a larger role in the NH than currently predicted in most climate models having explicit microphysics. For example, the satellite remote sensing study by Sourdeval et al. (2018) shows that $N_i$ in cirrus clouds (T < 235 K) outside the tropics is relatively high during the NH and SH winter seasons, with relatively low $N_i$ for T > 235 K. Highest $N_i$ during winter was observed in the cirrus cloud remote sensing studies of M2018), which were limited to in situ cirrus clouds. These studies also found $N_i$ outside the tropics (± 30° latitude) was higher over mountainous terrain, attributing this to mountain-induced wave clouds having relatively strong and sustained updrafts (and thus greater cooling rates producing high supersaturations) conducive for hom. This was also observed in the satellite remote sensing study of Gryspeerdt et al. (2018) that explained the higher $N_i$ over mountainous terrain in a similar way. Sedimenting ice crystals originating near cloud top may promote reduced $N_i$ at lower levels by lowering the relative humidity with respect to ice (RHi) there and "quenching" hom (Spichtinger and Gierens, 2009a), which is consistent with Fig. 10 in M2018. Aircraft measurements also underscore the importance of hom. By developing an analysis scheme sensitive to hom and het influences, Patnaude and Diao (2020) found evidence that both mechanisms are important in cirrus cloud formation, depending on environmental factors.

M2018 found that $N_i$ was relatively high at high latitudes, mostly over land. This can be partly explained by the relatively low concentrations of ice nucleating particles (INPs), measured or inferred, over the Southern Ocean (Vergara-Temprado et al., 2018; McCluskey et al., 2018; Carlsen and David, 2022) and in the Arctic, especially when snow and sea ice cover is more extensive (Creamean et al., 2018; Carlsen and David, 2022). That is, when $N_i$ produced by het is relatively low, the ice surface area produced for vapor removal is often inadequate for preventing the RHi from climbing and reaching the hom threshold in a cirrus cloud updraft, generally resulting in higher $N_i$ produced by hom (Krämer et al., 2016). Results similar to M2018 over the Southern Ocean are shown in Figure 1 of Gryspeerdt et al. (2018).

As in M2018, the new CALIPSO (Cloud-Aerosol Lidar and Infrared Pathfinder Satellite Observation) satellite retrieval of $D_e$, $N_i$, and IWC presented in Part 1 (Mitchell et al., 2024; henceforth M2024) uses the Imaging Infrared Radiometer (IIR) and CALIOP (Cloud and Aerosol Lidar with Orthogonal Polarization). The new retrievals of $D_e$, $N_i$, and IWC are based on an extended set of aircraft measurements. Whereas M2018 had several sets of retrievals for several independent formulations, M2024 established a unique retrieval scheme which combines various formulations according to temperature and latitude. Nevertheless, the general findings from M2018 highlighted above still stand.

Section 2 gives an overview of the new retrievals based on M2024. Our cirrus cloud sampling is discussed, and we describe why $0.3 < \tau < 3.0$, where $\tau$ is visible optical depth, may be the most radiatively representative $\tau$ category for all cirrus clouds. Section 3 presents the rationale for the identification of hom affected cirrus clouds and methods for quantifying the fraction of hom-affected cirrus clouds are developed and discussed. Section 4 synthesizes results from this and previous studies to propose a conceptual model for cirrus cloud formation and evolution. This work is related to arguably the most relevant other study in Sect. 5. Suggestions motivated by this work are given in Sect. 6 for improving the treatment of cirrus clouds in GCMs. A summary and conclusions are given in Sect. 7.





## 2 CALIPSO retrievals

### 2.1 Cloud sampling

As described in Part 1 (M2024), the CALIPSO retrieval uses IIR passive observations at 10.6 µm and 12.05 µm with description of the atmospheric column informed by perfectly collocated CALIOP observations. Results presented in this study are based on CALIPSO data during 2008, 2010, 2012 and 2013. Microphysical properties are retrieved for atmospheric columns that contain only one semi-transparent ice cloud layer that does not fully attenuate the CALIOP laser beam to ensure that cloud base is detected. We also require the cloud layer to be classified as ice with high confidence by the CALIOP ice/water phase algorithm (Avery et al., 2020). The CALIOP in-cloud 532 nm attenuated backscatter profiles with temperatures from MERRA-2 re-analyses (Gelaro et al., 2017) allows one to estimate the cloud layer radiative temperature, $T_r$ (Garnier et al., 2021), and cirrus clouds are defined as those ice clouds having a $T_r \leq 235$ K. The upper range of optical depth for semi-transparent single-layer cirrus clouds is about 3. The lower range is driven by retrieval uncertainties. In earlier work (M2018), global retrievals over both oceans and land have been in clouds having CALIOP 532 nm layer integrated attenuated backscatter (IAB) larger than 0.01 sr$^{-1}$, generally corresponding to clouds with visible optical depth $\tau > 0.3$. However, over oceans, uncertainties are lower due to the absence of highly variable land surface emissivities, allowing all cirrus clouds with valid retrieved effective emissivities at both 10.6 µm and 12.05 µm to be considered, yielding $\tau > \sim 0.01$.

In M2018, the sampled cirrus clouds were in situ cirrus clouds having base temperature, $T_{base}$, colder than 235 K. In this work, we take advantage of the improved ice/water phase assignment in the Version 4 CALIOP products to also include cirrus clouds with $T_{base}$ warmer than 235 K (and $T_r$ colder than 235 K), hereafter called liquid origin cirrus. Even though these liquid origin cirrus clouds are identified as high confidence ice clouds by CALIOP, the presence of liquid droplets in the lower part of the layer cannot be ruled out. This method is an approximation that may underestimate liquid origin cirrus clouds somewhat (overestimating in situ cirrus) since cloud condensate from below the 235 K isotherm may be advected across this isotherm upwind of the CALIOP nadir view when there is no cloud at nadir below this isotherm. In this case the cloud would be mistakenly classified as in situ cirrus, but such cases are believed to make a minor contribution to the total number of samples.

An analysis of in situ and liquid origin cirrus clouds is presented in Appendix A in terms of their temperature dependence (Fig. A2), their physical properties (Fig. A3), and the geographic distribution of the cirrus cloud sampling (Fig. A1). The top panels in Fig. A1 (a-b) show the geographic distribution of the sampling counts of optically thicker cirrus clouds that can be sampled over both land and ocean (IAB > 0.01 sr$^{-1}$; $\sim 0.3 < \tau < \sim 3$) for two seasons: December-January-February (DJF) (a) and June-July-August (JJA) (b). The bottom panels (c-d) show these same periods with the sampling count geographic distribution over oceans only for all samples where optically thinner cirrus can now be evaluated due to less uncertainty in surface emissions ($\sim 0.01 < \tau < \sim 3$). Cloud temperature distributions are shown in Fig. A2 (a-c) in terms of the number of samples (i.e., IIR 1-km$^2$ pixels) against $T_r$. The red curves with triangles characterize the situ cirrus with IAB > 0.01 sr$^{-1}$ (i.e., the cloud sampling used in M2018 over land (dashed) and over oceans (solid)) while red curves with diamonds indicate liquid



origin cirrus. The blue curves are based on all samples over oceans ($\sim 0.01 < \tau < \sim 3$). The cold samples with $T_r < 200$ K at high latitude over land are polar stratospheric clouds classified as ice clouds by the CALIOP algorithm.

Figure A2 (d-f) describe the transition from liquid origin to in situ cirrus clouds for all samples over ocean only (blue curves) as well as the optically thicker cirrus over both land and ocean (red curves). When only clouds with $\tau > \sim 0.3$ are sampled over oceans, liquid origin cirrus clouds prevail at mid- and high latitude (60 % and 70 %, respectively), but not in the tropics (32 %). When all clouds are considered, the percentage of in situ cirrus increases by 18 to 25 % and they always prevail. For these blue curves, it is seen that the liquid origin cirrus prevail at $T_r$ larger than about 227 K, which is $\sim 6$ K higher than shown in Luebke et al., 2016 (their Fig. 13) and Dekoutsidis et al., 2023 (their Fig. 4). This may be partly due to the different sampling method used here as described above. The blue curves in Figure A3 indicate that while liquid origin cirrus have relatively higher IWCs, the main difference between them and in situ cirrus clouds is the greater ice water path (IWP) and $\tau$ associated with liquid origin cirrus.

## 2.2 Realism of the CALIPSO cloud sampling for representing cirrus clouds

As in M2018, global retrievals over both oceans and land are in clouds having IAB > 0.01 sr$^{-1}$ or $\tau > \sim 0.3$, and we are now sampling both in situ and liquid origin cirrus clouds. This raises the question "Is this subcategory of cirrus cloud adequate for testing cirrus clouds in climate models?" It was argued in Sect. 6.3 of M2018 that cirrus clouds in this $\tau$ range should dominate the overall cirrus cloud net radiative forcing, but more can be said about this. Table 4 in Hong et al. (2016) gives global mean top-of-atmosphere (TOA) net ice cloud radiative effect (CRE) values for the following five $\tau$ categories: $\tau < 0.03$; $0.03 < \tau < 0.3$; $0.3 < \tau < 3.0$; $3 < \tau < 20$; $\tau > 20$. The corresponding net CRE values in W m$^{-2}$ are: 0.05, 0.96, 4.98, -0.49 and -0.43. These CRE values are weighted by ice cloud frequency of occurrence. Relative contributions of net CRE can be calculated by taking absolute values, in which case ice clouds having $0.3 \leq \tau \leq 3.0$ contribute 72% of the total "absolute" CRE. Moreover, the global mean net CRE for all ice cloud categories in their Table 4 was 5.07 W m$^{-2}$ whereas the global mean net CRE for $0.3 \leq \tau \leq 3.0$ is 4.98 W m$^{-2}$, further suggesting that this category is representative for all ice clouds. However, Hong et al. (2016) addressed all ice clouds (i.e., $T_r < 273$ K) whereas our study addresses only cirrus clouds ($T_r \leq 235$ K).

To further investigate this question for our cirrus sampling, we examine the distributions of retrieved $\tau$ (M2024) over oceans for the high and midlatitudes and the tropics. The tropics are defined here as $\pm 30°$, while the mid- and high latitudes are $30°$ – $60°$ and $60°$ - $90°$ in each hemisphere, respectively. The $\tau$ distributions are built using bins equal to 0.1 in log($\tau$) space. The number of IIR 1-km$^2$ cloudy pixels, p, and the product p x $\tau$, are plotted against $\tau$ in Fig. 1. This is done over oceans for all sampled cirrus clouds and for the subcategory of cirrus having IAB > 0.01 sr$^{-1}$ that have lower retrieval uncertainties. The p x $\tau$ product serves as a proxy for the radiative impact of cirrus clouds since $\tau$ and frequency of occurrence are the principal factors determining cloud radiative effects. The right panels in Fig. 1 show the cumulative distributions over $\tau$ (panel b) and for this proxy or radiative weighting (panel d). The cumulative distributions are normalized with respect to all cirrus clouds sampled. The cumulative distributions for IAB > 0.01 sr$^{-1}$ indicate that cirrus clouds having $\tau > \sim 0.3$ represent 31 % to 48 % (depending on latitude band) of the cirrus clouds sampled. However, they represent 78% to 87% of the radiative weight for all





cirrus clouds sampled (i.e., cirrus clouds having τ < 0.3 represent 13 % to 22 % of the radiative weight). This compares

favourably with results from Hong et al. (2016) described above, where the ice cloud CRE fraction for τ < 0.3 relative to τ <

3 is 0.17, or 17%. The occurrence frequency of cirrus having τ > 3 is negligible here because only clouds that are semi-

transparent to CALIOP are sampled.

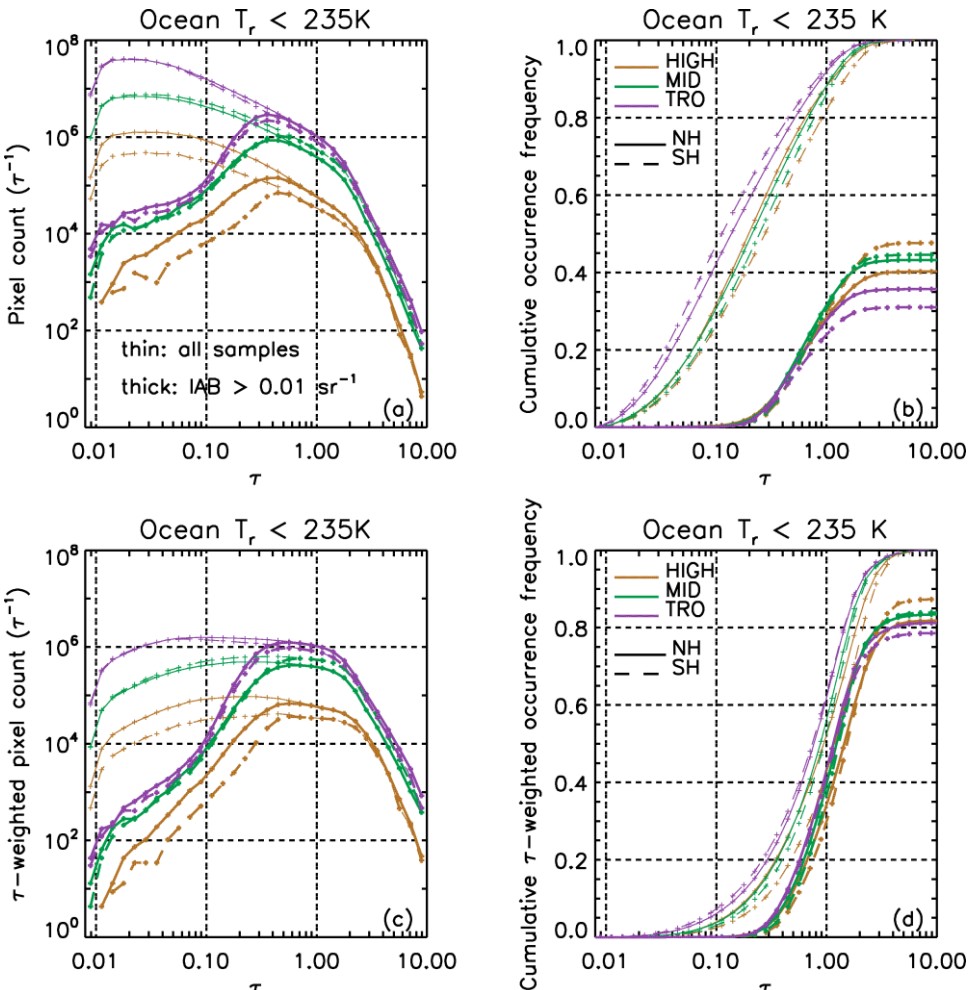


**Figure 1.** The upper panels (a and b) show IIR cirrus sampling statistics during 2008, 2010, 2012 and 2013 over visible optical depth (τ) while the lower panels (c and d) show the same distribution of sampled pixels multiplied by τ. All samples have a cloud radiative temperature $T_r \leq 235$ K for the tropics (purple), midlatitudes (green) and high latitudes (brown). Solid and dashed patterns refer to the northern and southern hemispheres, respectively. The product of the number of IIR pixels × τ is a proxy for the radiative

impact of cirrus clouds. The thin curves indicate all IIR samples while the thicker curves indicate only samples having IAB > 0.01 $sr^{-1}$. The right panels (b and d) show the cumulative distributions. The IIR pixels count is normalized per τ bin-size. Retrievals are only over oceans to minimize uncertainties at the lowest τ.



## 2.3    Global maps of cirrus cloud properties

Global maps for each season are shown for median $N_i$, $D_e$, IWC, and $T_r$ using the cloud sampling criteria described in section 2.1 and IAB $\geq$ 0.01 sr$^{-1}$ (i.e., $\sim 0.3 < \tau < \sim 3$) in Figs. 2, 3, 4, and 5, respectively.  Since this sampling criteria appears to render radiative properties representative of all cirrus clouds, the cloud property values predicted in GCMs for cirrus clouds should be similar to those shown in these figures.

Relatively high $N_i$ are found at high latitudes over land, as well as over mountainous regions in the mid-latitudes, especially

during winter.  These observations appear consistent with hom since hom tends to produce higher $N_i$ in stronger updrafts (e.g., orographic gravity waves induced by mountains) and during winter (Joos et al., 2014).  The seasonal dependence of $N_i$ in Fig. 2 was also found by Sourdeval et al. (2018) at mid-to-high latitudes for winter vs. summer.

For a given season, regions having relatively high $N_i$ in Fig. 2 have relatively small $D_e$ in Fig. 3.  An exception is in the tropical western Pacific during winter where $D_e$ is relatively small and $N_i$ is relatively low.  The deepest convection occurs in

this region during winter (Danielsen, 1993), producing the coldest cloud top temperatures where ice crystal growth rates are bound to be minimal, likely resulting in smaller $D_e$.  This phenomenon is more evident in Fig. 4, showing retrieved median IWCs, where IWC is lowest during winter in the tropical western Pacific due to these minimal temperatures.  The correspondence between $T_r$ (Fig. 5) and IWC (Fig. 4) is clearly seen.  This is an expected result predicted by the Clausius-Clapeyron equation.

Figures 6 and 7 show global maps of median $N_i$ and $D_e$ for in situ cirrus clouds only, where $N_i$ is higher and $D_e$ considerably smaller relative to Figs. 2 and 3 (where liquid origin cirrus are included). To interpret these figures, microphysical properties of in situ and liquid origin cirrus clouds are investigated in Fig. A3.  Although corresponding to a specific season and latitude zone, these Fig. A3 results are globally representative.  Median $D_e$ (panel b) increases as $T_r$ increases, with no marked difference between in situ and liquid origin cirrus clouds at a given $T_r$, suggesting that the $D_e$ differences seen in Figs. 3 and 7

are due to temperature differences.  Temperature could also be a factor in explaining $N_i$ differences between liquid origin and in situ cirrus since the later tends to form at lower temperatures (Fig. A2) and hom depends strongly on temperature (e.g., Lamb and Verlinde, 2011).  Figure A3 shows that both the geometric thickness (panel d) and the equivalent thickness (panel c) seen by IIR are on average greater for liquid origin cirrus clouds than for in situ cirrus clouds.  This is the main reason that IWP and $\tau$ tend to be substantially greater for liquid origin cirrus clouds, as shown in Fig. A3(g-h).  Figure A3 shows that IWC

and $N_i$ (panels f and e) are both larger in liquid origin cirrus, in agreement with Luebke et al. (2016), only when clouds with $\tau < \sim 0.3$ are included (blue curves).  Figure A3 highlights the importance of cloud sampling in optically thick vs. thin cirrus when interpreting cirrus cloud field campaign measurements of $N_i$ and IWC, and ultimately in terms of the contribution of het and hom to cloud formation.





**Figure 2.** Global-seasonal maps of the median ice particle number concentration $N_i$ for IAB > 0.01 sr$^{-1}$ (~ 0.3 < $\tau$ < ~ 3) that include both liquid origin and in situ cirrus clouds. The color legend gives $N_i$ in L$^{-1}$. The four panels are for (a) December-January-February (DJF), (b) March-April-May (MAM), (c) June-July-August (JJA), and (d) September-October-November (SON) during 2008, 2010, 2012 and 2013.






**Figure 3. Global-seasonal maps of the median effective diameter $D_e$ for IAB > 0.01 sr$^{-1}$ (~ 0.3 < $\tau$ < ~ 3) that include both liquid origin and in situ cirrus clouds. The color legend gives $D_e$ in microns. Otherwise, as in Fig. 2.**






**Figure 4.** **Global-seasonal maps of the median ice water content IWC for IAB > 0.01 sr⁻¹ (~ 0.3 < τ < ~ 3) that include both liquid origin and in situ cirrus clouds. The color legend gives IWC in mg m⁻³. Otherwise, as in Fig. 2.**





Figure 5. Global-seasonal maps of the median cirrus cloud radiative temperature $T_r$ (K) for IAB > 0.01 sr$^{-1}$ (~ 0.3 < τ < ~ 3) that include both liquid origin and in situ cirrus clouds. Otherwise, as in Fig. 2.





**Figure 6. Global-seasonal maps of the median ice particle number concentration $N_i$ for IAB > 0.01 sr$^{-1}$ (~ 0.3 < $\tau$ < ~ 3) based on only in situ cirrus clouds. The color legend gives $N_i$ in L$^{-1}$. Otherwise, as in Fig. 2.**



**Figure 7. Global-seasonal maps of the median effective diameter D$_e$ for IAB > 0.01 sr$^{-1}$ (~ 0.3 < τ < ~ 3) based on only in situ cirrus clouds. The color legend gives D$_e$ in microns. Otherwise, as in Fig. 2.**



## 3 Contribution of hom-affected cirrus clouds

### 3.1 Sensitivity of cirrus cloud physical properties to hom

As discussed in M2024, when all IIR samples over oceans were used ($\sim 0.01 < \tau < \sim 3$), retrieved median $N_i$, $R_v$ (mean volume radius), and IWC were in reasonable agreement with corresponding values in the cirrus cloud property climatology of Krämer et al. (2020) that is based on many field campaigns throughout the world. However, when the CALIPSO retrievals were restricted to $\sim 0.3 < \tau < \sim 3$, the agreement degraded for $N_i$ and IWC. Again, this raised the question of whether cirrus clouds having $\tau$ smaller and larger than 0.3 are characterized by different physical properties.

Figures 8-11 provide a means of investigating this question, evaluating $N_i$, IWC, $D_e$ and the sampling densities for two different $\tau$ categories: $\sim 0.01 < \tau < \sim 0.3$ (IAB < 0.01 sr$^{-1}$) and $\sim 0.3 < \tau < \sim 3$ (IAB > 0.01 sr$^{-1}$); henceforth categories 1 and 2. Only retrievals over ocean are considered since variable land emissivities preclude retrievals over land in Category 1. These quantities are plotted in $T_r$ vs. $T_r - T_{top}$ space where $T_{top}$ is temperature at cloud top and $T_r$ is most of the time near mid-cloud (M2024). Thus, $T_r - T_{top}$ is an index for cloud geometrical thickness. Figure 8 shows that for Category 2 (panels a to f), the highest $N_i$ (resulting from hom due to its magnitude) is found in relatively thin clouds (consistent with Fig. 10 in M2018), while Category 1 (panels g to l) is almost featureless with $N_i < 100$ L$^{-1}$ in general. Note that the retrievals characterize cloud layers (i.e., no vertical profiling within a cloud is possible). The cirrus cloud remote sensing results from Dekoutsidis et al. (2023) and the cirrus cloud modelling results of Spichtinger and Gierens (2009a,b) both show that hom is primarily active only near cloud top, with the later showing relatively low $N_i$ in the mid- to lower cloud where sedimenting ice crystals quench hom by lowering the RHi. Based on this, geometrically thicker cirrus should have vertically averaged $N_i$ much lower than $N_i$ in the upper cloud and in geometrically thinner cirrus (assuming comparable thicknesses for the ice generating regions). These findings are consistent with our results for Category 2. IWC in Fig. 9 is higher in Category 2 (panels a-f) than in Category 1 (panels g-l) clouds which exhibit much lower values, typically smaller than 4 mg m$^{-3}$. While the differences in magnitude may be expected due to differences in $\tau$, the different distribution patterns, with highest values in thinner clouds in Category 2 for a given $T_r$, were not anticipated. Differences in $D_e$ between the two $\tau$ categories are considerably less, as shown in Fig. 10. For a given $T_r$, $D_e$ tends to be quasi-constant, although usually decreasing for the thinnest clouds in both categories, possibly due to entrainment. But this $D_e$ decrease could also be due to hom in Category 2 since Figs. 3 and 7 show that hom acts to decrease $D_e$, and hom is very active in Category 2 for relatively thin clouds. Figure 11 shows the number of pixels in $T_r$ vs. $T_r - T_{top}$ space where it is apparent that Category 1 cirrus clouds (panels g-l) tend to be geometrically thinner relative to Category 2 (panels a –f). This seems consistent with our visual experience of thin cirrus that have a minor impact on sunlight attenuation. To summarize, these results show that hom is active in Category 2 (due to the high $N_i$ in relatively geometrically thin cirrus) but not Category 1. Moreover, they show that hom mostly affects $N_i$ and IWC, with some impact on $D_e$ (referring to Figs. 3 and 7 as well). This suggests that the ratio IWC/$D_e$ may be sensitive to hom.







**Figure 8.** **Cirrus cloud median ice particle number concentration $N_i$ plotted in $T_r$ vs. $T_r - T_{top}$ space (for both liquid origin and in situ cirrus) where $T_{top}$ is cloud top temperature and $T_r - T_{top}$ is related to cloud geometrical thickness. Each row features the tropics (TRO, panels a, d, g, j), midlatitudes (MID, panels b, e, h, k), and high latitudes (HIGH, panels c, f, i, l) in both hemispheres either for winter (WIN, panels a-c and g-i) or summer (SUM, panels d-f and j-l) during 2008, 2010, 2012 and 2013. The top half panels (a-f) feature cirrus having IAB > 0.01 sr$^{-1}$ (~ 0.3 < τ < ~ 3) while the bottom half panels ( g-l) feature cirrus having IAB < 0.01 sr$^{-1}$ (~ 0.01 < τ < ~ 0.3). The numbers in the color legend are to be multiplied by 100 to give $N_i$ in L$^{-1}$.**





**Figure 9.** Cirrus cloud median ice water content (IWC) plotted in $T_r$ vs. $T_r - T_{top}$ space (for both liquid origin and in situ cirrus) where $T_{top}$ is cloud top temperature and $T_r - T_{top}$ is related to cloud geometrical thickness. The top half panels (a-f) feature cirrus having IAB > 0.01 sr⁻¹ (~ 0.3 < τ < ~ 3) while the bottom half panels (g-l) feature cirrus having IAB < 0.01 sr⁻¹ (~ 0.01 < τ < ~ 0.3). The color legend gives IWC is mg m⁻³. Otherwise, the labeling convention is the same as in Fig. 8.





**Figure 10.** Cirrus cloud median effective diameter $D_e$ plotted in $T_r$ vs. $T_r - T_{top}$ space (for both liquid origin and in situ cirrus) where $T_{top}$ is cloud top temperature and $T_r - T_{top}$ is related to cloud geometrical thickness. The top half panels (a-f) feature cirrus having IAB > 0.01 sr$^{-1}$ ($\sim 0.3 < \tau < \sim 3$) while the bottom half panels (g-l) feature cirrus having IAB < 0.01 sr$^{-1}$ ($\sim 0.01 < \tau < \sim 0.3$). The color legend gives $D_e$ in microns. Otherwise, the labeling convention is the same as in Fig. 8.





**Figure 11.** The cirrus cloud sampling density (i.e., number of IIR pixels filled with cirrus) plotted in $T_r$ vs. $T_r - T_{top}$ space (for both liquid origin and in situ cirrus) where $T_{top}$ is cloud top temperature and $T_r - T_{top}$ is related to cloud geometrical thickness. The top half panels (a-f) feature cirrus having IAB > 0.01 sr$^{-1}$ ($\sim 0.3 < \tau < \sim 3$) while the bottom half panels (g-l) feature cirrus having IAB < 0.01 sr$^{-1}$ ($\sim 0.01 < \tau < \sim 0.3$). The color legend gives sample counts in a $\log_{10}$ scale. Otherwise, the labeling convention is the same as in Fig. 8.



### 3.2 Using the extinction coefficient to identify hom-affected and het-only cirrus clouds

As described in M2018 and M2024, one of the cloud properties retrieved by this method is the extinction coefficient for visible light, $\alpha_{ext}$. For visible light and cirrus ice particles, the extinction efficiency $Q_{ext}$ can be approximated as $Q_{ext} \approx 2$. In this context, IWC was derived by rearranging the $D_e$ expression in Mitchell (2002) as IWC = $(\rho_i/3) D_e \alpha_{ext}$, where $\rho_i$ = density of bulk ice and $\alpha_{ext}$ is twice the PSD projected area. Here we use retrieved $\alpha_{ext}$ as a means for measuring the activity of hom since it contains the ratio IWC/$D_e$, which above was shown to be sensitive to hom activity. For a given $T_r$, relatively high values of

$\alpha_{ext}$ should correspond to a substantial impact from hom on the cloud microphysics.

To test this idea, $N_i$ is plotted in $T_r$ vs. log($\alpha_{ext}$) space in Fig. 12 for retrievals over ocean using all IIR samples, with $\tau$ ranging from ~ 0.01 to ~ 3, for the tropics and mid- to high latitudes during winter and summer (for both hemispheres). White regions in these plots are where the number of samples is 0 or smaller than 10. Characteristic in all plots is a broad region on the left side (relatively low $\alpha_{ext}$) where $N_i < 30$ L$^{-1}$, apparently corresponding to het only. To the right of this region is a gradient of

300 increasing $N_i$, culminating in values of $N_i > 2000$ L$^{-1}$. This gradient region is likely produced by varying degrees of hom activity. To the right of this gradient, the boundary between peak $N_i$ values and the "no data" region has a curious diagonal orientation (as well as the gradient) that may be explained from first principles.

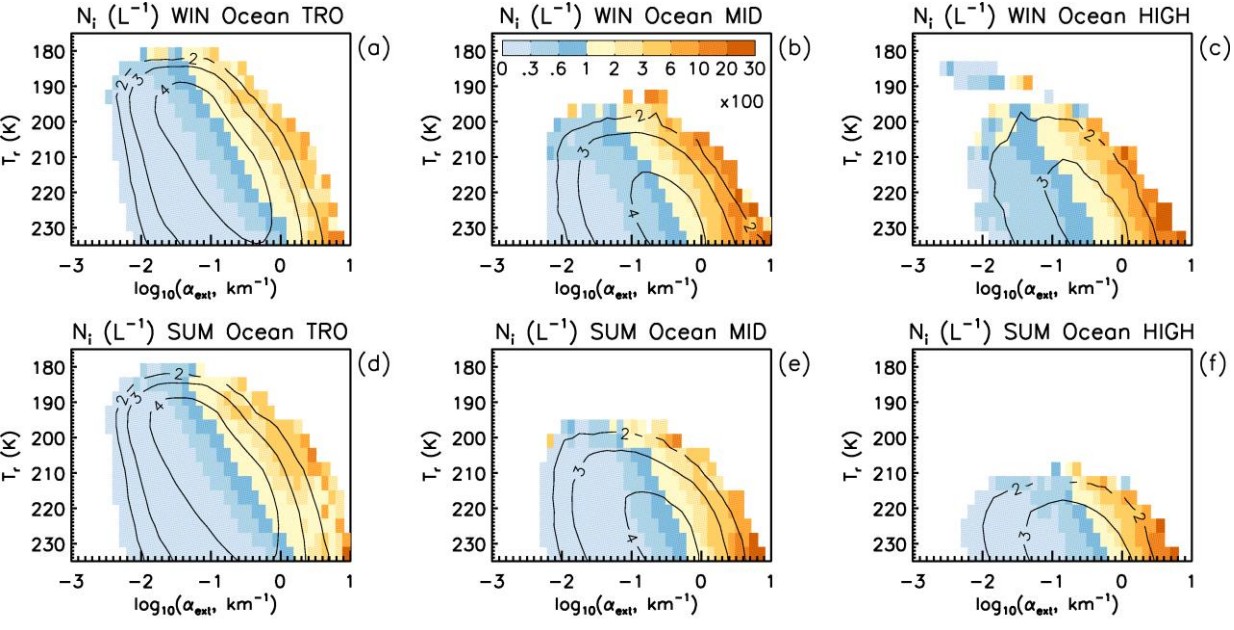

**Figure 12. Median retrieved ice particle number concentration $N_i$ plotted in $T_r$ vs. log($\alpha_{ext}$) space to separate the het- and hom-regimes. The black curves show the log$_{10}$ of the number of samples. The numbers in the color legend are to be multiplied by 100 to give $N_i$ in L$^{-1}$. Retrievals are over ocean using all IIR samples, with $\tau$ ranging from ~ 0.01 to ~ 3, for the tropics (TRO, panels a and d), midlatitudes (MID, panels b and e), and high latitudes (HIGH, panels c and f) during winter (WIN, panels a-c) and summer (SUM, panels d-f) using both hemispheres during 2008, 2010, 2012 and 2013.**



To explore this further, two formulations of the Clausius-Clapeyron equation and the supersaturation equation for hom (Lamb and Verlinde, 2011) were used:

$$e_{si} = e_{s0} \, exp\left[\frac{L_s}{R_g} \left(\frac{1}{T_0} - \frac{1}{T}\right)\right]$$  (1)

where $e_{si}$ is water vapor pressure at ice saturation, $L_s$ is latent heat of sublimation, $R_g$ is the gas constant, T is temperature and $e_{so}$ and $T_0$ refer to water vapor pressure and temperature at the ice-point of water (6.11 hPa and 273.15 K, respectively). Reformulating Eq. (1) gives an expression useful in calculating the supersaturation required for homogeneous ice nucleation:

$$e_s/e_{si} = exp\left[\frac{L_f}{R_g} \left(\frac{1}{T} - \frac{1}{T_0}\right)\right]$$  (2)

where $e_s$ = water vapor pressure at water saturation and $L_f$ = latent heat of fusion. Finally, the supersaturation threshold where hom occurs, $S_i^f$, is predicted by

$$S_i^f = 1.0 + 0.305 \, \frac{e_s}{e_{si}}$$  (3)

Noting that the water vapor pressure at $S_i^f$, $e_{hom}$, is simply $e_{hom} = S_i^f \, e_{si}$, water vapor densities are obtained from the Gas Law to predict the maximum IWC resulting from hom:

$$IWC_{hom} = \rho_{hom} - \rho_{si}.$$  (4)

But to relate $IWC_{hom}(T)$ to retrieved IWC in $T_r$ vs. $\alpha_{ext}$ space, a corresponding $\alpha_{ext}$ is needed which requires $D_e$. For the tropics, the PSD mean ice particle maximum dimension $D_{mean}$ was related to T using the CEPEX PSD scheme described in Mitchell et al. (1999), where only the large ice particle mode was used. For outside the tropics, a $D_{mean} - T$ relationship based on the SPARTICUS field campaign (Mishra et al., 2014) was used, which is $log(D_{mean}) = 0.0307 \, T + 3.169$ with $D_{mean}$ in microns and T in °C. Using the ice particle mass- and projected area-dimensional relationships from Erfani and Mitchell (2016) and an exponential gamma PSD, $D_e$ was calculated from $D_{mean}$ as described in that study. This gives $D_e$ under "normal" conditions. To estimate $D_e$ under maximum hom conditions (corresponding to Eq. (4)), we use panels h, i, and l of Fig. 16 over land for high latitudes and for mid-latitudes during winter which show that when hom is most active, $D_e$ decreases from "normal values" ~ 55 μm to ~ 35.5 μm for T between 219 and 235 K, or by a factor of 0.646. Thus, we apply this factor to $D_e$ estimated from $D_{mean}$ to estimate $D_e$ for maximum hom conditions. Finally, we calculate $N_i$ by converting this $D_e$ back to $D_{mean}$ for an exponential PSD using Eq. (7) in Mitchell et al. (2020), and then calculating the PSD slope parameter λ from $D_{mean}$ (λ = $1/D_{mean}$). For the region where hom is most active, $N_i$ is $N_{max}$ calculated as

$$N_{max} = \frac{\Gamma(\nu+1) \, IWC_{hom} \, \lambda^\beta}{\alpha \, \Gamma(\beta+\nu+1)}$$  (5)





where ν is a parameter in the gamma PSD $N(D) = N_0 \, D^\nu \, exp(-\lambda D)$ and α and β define the ice particle mass relationship m = $\alpha D^\beta$, where ν = 0 here. The values of α and β were obtained from Erfani and Mitchell (2016) in the same way $D_e$ was calculated.

Panels a to f of Fig. 13 show the retrieved median IWC plotted in $T_r$ vs. $log(\alpha_{ext})$ space for all samples over oceans, with triangles predicted from (4) overlaying these retrieved IWCs. The different $D_{mean}$-T relationships for the tropics and for outside the tropics yield slightly different $\alpha_{ext}(T)$ associated to $IWC_{hom}(T)$ (since $\alpha_{ext}$ = 3 IWC/($\rho_i \, D_e$)). The color within the triangle indicates the approximate $IWC_{hom}$ value via the color bar legend. It is seen that the color within the triangles matches the background color in almost all cases, showing good agreement between basic theory and the retrieved IWC where hom is

expected to be most active. $IWC_{hom}$ was also calculated from Eq. (1) in Schneider et al. (2021) shown by the squares in Fig. 13, where the hom onsets of aqueous sulfuric acid aerosol particles were measured in a cloud chamber and are expressed in terms of $S_i^f$ and temperature. There is close agreement between the theoretical and experimental methods. In the tropics the margin between the triangles and the "no data" region is wider. This may be due to the deep convective origin of most tropical cirrus, with convective plumes overshooting the temperature level where hom is first activated, carrying ice to lower

temperatures. Outside the tropics, this phenomenon is less evident. In panels g to l, retrieved median $D_e$ is plotted in $T_r$ vs. $log(\alpha_{ext})$ space for all samples over oceans, with theoretical and experimental hom $D_e$ indicated by the triangles and squares, respectively, with their color based on the same color legend as used for the retrievals. The predicted and retrieved $D_e$ values are in general agreement for this hom-active region.

The upper portion of Fig. 14 (panels a-f) is the same as Fig. 12 except that $N_{max}$ predicted from Eq. (5) and Eq. (1) of

Schneider et al. (2021) is overplotted as shown by the triangles and squares, respectively, with the color inside the triangles indicating the concentration range (based on the color bar). Outside the tropics (especially during winter), there is a close correspondence between the triangle (or square) color and adjacent retrieval colors that show maximum $N_i$ for given $T_r$ (thus indicating the region of maximum hom activity). The predicted colors (triangles and squares) tend to indicate higher $N_i$ than the adjacent background retrieved $N_i$ colors, which is consistent with expectations that hom events are diluted through mixing

and entrainment (Jensen et al., 2012). The diagonal orientation of this hom region is a consequence of the Clausius-Clapeyron equation that predicts lower water vapor pressure and density as T decreases. In the tropics, the predicted colors indicate considerably higher $N_i$ values than the adjacent retrieval values. This may indicate that hom is less active in the tropics for our cloud sampling. The lower portion of Fig. 14 (panels g to l) shows the pixel count densities in $T_r$ vs. $log(\alpha_{ext})$ space along with predicted black triangles and squares that indicate the region where hom is most active. Outside the tropics, the

triangles/squares coincide with relatively low pixel counts, indicating the frequency of occurrence for hom events is quite low, consistent with theoretical expectations (e.g., Krämer et al., 2016). Figures complementing Figs. 13 and 14 are given in the Supplement as Figs. S1 and S2 over land for IAB > 0.01 sr$^{-1}$. To be noted is the strong contribution from polar stratospheric clouds over Antarctica at very cold temperatures in the winter season (panels c and i).







**Figure 13.** Median retrieved IWC (upper half, panels a -f) and $D_e$ (lower half, panels g-l) plotted in $T_r$ vs. $\log(\alpha_{ext})$ space to separate the het- and hom-regimes. The black curves show $\log_{10}$(number of samples). The color-bars give IWC (mg m$^{-3}$) and $D_e$ (μm). Retrievals are over oceans using all IIR samples, with τ ranging from ~ 0.01 to ~ 3, for the tropics (TRO), midlatitudes (MID), and high latitudes (HIGH) during winter (WIN) and summer (SUM) using both hemispheres during 2008, 2010, 2012 and 2013. The triangles and the squares represent predictions from simple hom theory using Eq. (4) and Schneider et al. (2021), respectively, and the color within each triangle or square indicates the approximate IWC$_{hom}$ (panels a-f) and hom $D_e$ (panels g-l) values via the color bar legends.








**Figure 14.** Same as Fig. 13 except that the upper half panels (a-f) show median ice number concentration $N_i$ (as in Fig. 12) and the lower half panels (g-l) show $\log_{10}$(number of samples). The triangles and squares represent predictions from simple hom theory as in Fig. 13. The color within each triangle in panels a-f indicates the approximate $N_{max}$ in hom conditions via the color bar legends.



### 3.3 Microphysical attributes of het and hom-affected cirrus clouds

It has now been shown that $\alpha_{ext}$ seems to be a suitable metric to identify varying degrees of hom activity, but how can this be used to quantify the contribution of hom-affected cirrus clouds in terms of frequency of occurrence and radiative impact? To address this question, a method is presented that estimates the fraction of hom-affected cirrus clouds such that the clouds are strongly altered microphysically relative to het cirrus.

The median $N_i$ corresponding to various temperature intervals in Figs. 12 and 14 was plotted against $\alpha_{ext}$ in Fig. 15 (panels a to f), along with the corresponding median $D_e$ plotted against $\alpha_{ext}$ in panels g to l. As with Figs. 12-14, these retrievals are over oceans for all IABs (~ 0.01 < $\tau$ < ~ 3), but where median $D_e$ is smaller than at the sensitivity limit (M2024). Colors indicate the 4 K temperature interval corresponding to the retrievals. Outside the tropics, it is seen that two populations exist, characterized by different $d\log(N_i)/d\log(\alpha_{ext})$ slopes in the case of $N_i$ or trends in the case of $D_e$, mostly for the higher temperatures. If one were to describe the "warm-colored" (T > 215 K) $N_i$ data with two linear fits, a change in slope occurs around $\alpha_{ext} \approx 0.3$ km$^{-1}$. The increased slope for $\alpha_{ext} > 0.3$ km$^{-1}$ is where median $D_e$ increases slowly or starts decreasing. For $T_r < 215$ K, these trends are less coherent. This behaviour suggests a transition from het-only cirrus to hom-affected cirrus (henceforth het cirrus and hom cirrus). For $N_i$ in the tropics, there is no apparent change in slope, which may indicate that hom is less active in the tropics. Note that this analysis is based on clouds having $\tau$ < ~ 3 and thus does not consider deep convective columns where $\tau > 3$ and hom is probably quite active.

A maximum in $D_e$ is expected to occur after a slope change regarding $N_i$ due to the competition for water vapor among a relatively high concentration of newly formed ice crystals. This competition reduces the growth rates of these ice crystals, keeping them relatively small. Other than hom, we know of no process that can explain the decrease in $D_e$ at higher extinction shown in Fig. 15. Since for spherical ice the spherical volume radius $R_v$ is proportional to $(1/N_i)^{1/3}$, it is apparent that even for non-spherical ice crystals, it will take a larger change in $N_i$ to affect a significant change in $R_v$ or $D_e$ through vapor competition effects. Thus, regarding $D_e$ and $N_i$, a decrease in $D_e$ caused by hom would be a more conservative criterion for defining hom cirrus. This would be more pragmatic if one was primarily interested in the impact of hom on radiation since cloud radiative properties are parameterized in terms of $D_e$ and IWC (e.g., Gettelman et al., 2010; Emde et al., 2016). For this reason, we define hom cirrus as cirrus having an extinction exceeding that at the $D_e$ maximum.

### 3.3.1 Hom intensity

To better understand these principles, these ideas can be tested in an environment where INP concentrations are higher, namely over land. Figure 16 is like Fig. 15 except the retrievals are over land and therefore only where IAB > 0.01 sr$^{-1}$ ($\tau$ > ~ 0.3). Despite the different ranges of optical depth in Figs. 15 and 16, observations are qualitatively similar. Regarding the $N_i$ plots (panels a –f), there is a clear inflection point around $\alpha_{ext} = 0.2$ to 0.3 km$^{-1}$ where the slope changes for 215 < $T_r$ < 235 K in all latitude bands and around $\alpha_{ext} = 0.1$ to 0.2 km$^{-1}$ for $T_r < 215$ K in three latitude bands. Referring to Fig. 14, this change in







**Figure 15.** The dependence of median log $N_i$ (upper half, panels a-f) and median $D_e$ (lower half, panels g-l) on $\alpha_{ext}$ as derived from Fig. 12 and Fig. 13 (panels g-l), respectively for all samples over oceans ($\sim 0.01 < \tau < \sim 3$). The various $T_r$ intervals (4 K/interval) are given by the color legend. As in Figs. 12 and 13, each row features the tropics (TRO), midlatitudes (MID), and high latitudes (HIGH) in both hemispheres for either winter (WIN) or summer (SUM). The vertical dashed line at $\alpha_{ext} = 0.3$ km⁻¹ in panels a – f marks where a change in slope generally occurs for the 231-235 K interval. These same dashed line positions are also indicated in panels g – l showing their relation to the $D_e$ maximum.







**Figure 16.** Same as in Fig. 15 but for retrievals over land where IAB > 0.01 sr$^{-1}$ (~ 0.3 < τ < ~ 3).





slope is due to hom with hom affecting the cirrus microphysics for $\alpha_{ext} \geq 0.3$ km$^{-1}$. The slope of the N$_i$ data for 235-215 K for

$\alpha_{ext} \geq 0.5$ km$^{-1}$ appears related to the production rate of ice crystals through hom (i.e., N$_i$ is higher for steeper slopes) and as such can be viewed as a measure of the intensity of hom. The mid-latitude slope over land is greater during winter relative to summer, consistent with N$_i$ in Figs. 2 and 6. For $\alpha_{ext} < 0.3$ km$^{-1}$, het is expected to dominate, and the N$_i$ distributions are almost flat, with N$_i$ increasing slightly for decreasing $\alpha_{ext}$ for some latitude bands/seasons.

The concept of hom intensity (related to ice crystal production rates via hom) may also be described through the slope or

trend in plots of D$_e$ vs. log $\alpha_{ext}$ within the hom regime described above. Comparing panels g to l of Fig. 15 and Fig. 16, we see that this slope tends to be more negative over land than over oceans, consistent with corresponding dlog(N$_i$)/dlog($\alpha_{ext}$) slopes that tend to be more positive over land than over oceans. This may be due to higher cirrus cloud updrafts over land, especially over mountainous terrain. Figures 2 and 6 show higher N$_i$ over mountainous terrain.

### 3.3.2   A cirrus cloud Twomey effect?

Most evident when comparing Figs. 15 and 16 for $\alpha_{ext} < 0.3$ km$^{-1}$ (where het is expected to prevail) is that median N$_i$ is higher over land (up to a factor of 10), presumably due to higher INP concentrations over land. For direct comparison with Fig. 16, results similar to Fig. 15 but for IAB > 0.01 sr$^{-1}$ are shown in Fig. S3 of the Supplement. In the lower half of Figs. S3 and 16, D$_e$ is related to $\alpha_{ext}$ where it is evident that D$_e$ is substantially smaller over land relative to oceans for $\alpha_{ext} < 0.3$ km$^{-1}$. Since N$_i$ is higher over land (although perhaps not at high latitudes, based on Fig. S3), this suggests that higher INP over land (which

can also be enhanced by stronger updrafts) may be producing a "Twomey effect" in het cirrus clouds over land.

### 3.4   Quantifying the contribution of hom-affected cirrus clouds based on extinction and D$_e$

As shown in Figs. 15 and 16, D$_e$ exhibits a maximum where d(D$_e$)/d(log($\alpha_{ext}$)) $\approx$ 0. This maximum generally occurs for $\alpha_{ext} >$ 0.3 km$^{-1}$ and serves as a more conservative estimate of the transition from het-only to hom-affected conditions. Fig. 17 shows plots of N$_i$ in T$_r$ vs. log($\alpha_{ext}$) space for the tropics, midlatitudes, and high latitudes over oceans for DJF and JJA in each

hemisphere, where ~ 0.01 < $\tau$ < ~ 3. The transition from het to hom cirrus at each T$_r$ value is where D$_e$ is maximum, and this maximum is marked by the black "+" signs. However, in the tropics and in the summer NH midlatitudes, D$_e$ maxima were ambiguous and the transition from het to hom cirrus at a given T$_r$ was determined by a N$_i$ threshold, N$_{thres}$. That is, N$_{thres}$ is the mean value of N$_i$ that correspond to all the D$_e$ maxima (51 in total) in the extra-tropics. The value and standard deviation of N$_{thres}$ is 118 $\pm$ 57 L$^{-1}$ where the standard deviation is likely subject to variations in INP concentration (Haag et al., 2003). The

hom threshold was assigned to the $\alpha_{ext}$ having the corresponding N$_i$ closest to N$_{thres}$. In two extra-tropical datasets (DJF MID SH and DJF MID NH), N$_{thres}$ was used to estimate the hom threshold for T$_r$ < 219 K and T$_r$ < 207 K, respectively.

The horizontal dashed line in Fig. 17 indicates the temperature T$_{min}$, below which the number of cirrus cloud samples was too low for statistically meaningful analysis. The region of hom-affected cirrus clouds is thus defined as the region bounded by the + signs in the vertical and the dashed line in the horizontal, with the right vertical boundary defined by the absence of

data and the lower horizontal boundary defined by the 235 K isotherm. Also plotted in Fig. 17 are alternating solid and dashed







**Figure 17. Procedure for identifying hom affected cirrus over oceans using all samples (~ 0.01 < τ < ~ 3). The hom affected region is bounded by the $D_e$ maxima (+ signs) in the vertical and in the horizontal by the 235 K isotherm and $T_{min}$ (dashed line) below which sampling is too low for meaningful analysis. Other features are as in Fig. 14 a-f. The alternating solid-dashed contour lines are varied with increments of 0.10 (i.e., 10%) from right to left to show the approximate fraction of samples (at a given $T_r$) within the hom affected region. Each row features the tropics (TRO), midlatitudes (MID), and high latitudes (HIGH) for either the northern (NH, panels a-c and g-i) or southern (SH, panels d-f and j-l) hemisphere in DJF and JJA during 2008, 2010, 2012, and 2013.**




curves indicating the fraction of samples, in increments of 0.10 from right to left, having a larger extinction than the extinction
corresponding to the curve (at a given $T_r$). This provides estimates of the fraction of hom cirrus for a given $T_r$. For reference, the triangles predicted from hom theory are also shown (same as in Fig. 14). Overall, $\alpha_{ext}$ at het-hom transitions is found between ~ 1.8 km$^{-1}$ (in the tropics) and 0.02 km$^{-1}$ (in the winter Arctic) and tends to decrease as $T_r$ decreases (similar to the temperature dependence of $N_i$ predicted from hom theory). Similar results were found for cirrus clouds over land (~ 0.3 < τ < ~ 3), which are shown in Fig. S4 in the Supplement.

These hom region boundaries were incorporated into look-up-tables (LUTs) that relate 4 K temperature intervals to latitude band, season, and surface type (oceans or land) for determining whether a sample was het-only or hom-affected. When $D_e$ > sensitivity limit, we set the sample as het-only.

### 3.4.1   Results

Results from this analysis over oceans (~ 0.01 < τ < ~ 3) are shown in Figs. 18 and 19, where the hom fraction vs. $T_r$ is
shown in red. The hom fraction, which is defined as the number of hom cirrus pixels divided by the total number of cirrus pixels, varies with $T_r$ and is typically smaller than 0.4 (with 5 exceptions at the lowest $T_r$ outside the tropics). In addition, Fig. 18 shows median $N_i$ vs. $T_r$ for all cirrus (blue), for het cirrus (black), and for hom cirrus (orange). The "all cirrus" and "het cirrus" profiles are identical where hom fraction is null and they are close when hom fraction is relatively small. Conversely, when the hom fraction is relatively large, the "all cirrus" and "hom cirrus" profiles become closer. For the latter, median $N_i$ is
most of the time larger than 200 L$^{-1}$. In this way, Fig. 18 shows how the $T_r$ dependence of the hom fraction affects median $N_i$. Figure 19 is similar to Fig. 18 except cloud optical depth τ is evaluated instead of $N_i$. Noting the log scaling of τ, it is clear that hom is having a non-trivial impact on median τ. That is, aerosol-cloud-radiation interactions for cirrus clouds are likely greatest for optically thicker cirrus where hom is active. The maximum CRE change is expected when INP concentrations are just enough to completely prevent hom but do not exceed this concentration by more than a factor of ~ 3 (Storelvmo et al.,
2013). Under this condition, the blue median τ profile for all cirrus and the black het cirrus τ profile should be the same. Thus, the difference between the blue and black τ profiles reveals the potential relative CRE impact of INPs (e.g., Mitchell and Finnegan, 2009; Storelvmo et al., 2013; Gruber et al., 2019). Finally, it should be noted that the relatively low hom fractions in the tropics are believed to partly result from sampling restrictions, where only cirrus having τ < ~ 3 are sampled. That is, hom is probably very active where deep convection occurs, but these clouds are characterized by τ >> 3.
Global geographical distributions of hom fractions in the atmospheric column are shown for DJF and JJA in Fig. 20. Because hom clouds tend to have an overall larger optical depth than het clouds, the hom fraction over land based on samples with IAB > 0.01 sr$^{-1}$ (i.e., τ > ~ 0.3) is expected to be overestimated. Therefore, we applied a zonal correction factor at 2° resolution which was estimated from samples over oceans as:

$$\text{Correction factor} = \frac{\text{fraction}_{All}}{\text{fraction}_{IAB > 0.01\ sr^{-1}}} \qquad (6)$$






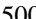

**Figure 18. The $T_r$ dependence of the hom fraction (red dashed) over oceans ($\sim 0.01 < \tau < \sim 3$). Each row features the tropics (TRO), midlatitudes (MID), and high latitudes (HIGH) for either the northern (NH) or southern (SH) hemisphere in DJF and JJA during 2008, 2010, 2012, and 2013. Also shown are median $N_i$ profiles for all cirrus (blue), for het cirrus (black), and for hom cirrus (orange). The 25 to 75 percentile range is indicated by darker grey shading for het cirrus and darker orange shading for hom cirrus, while the 10 to 90 percentile range is indicated by lighter grey shading for het cirrus and lighter orange shading for hom cirrus.**







**Figure 19. As in Fig. 18, but instead of showing median $N_i$ profiles, median $\tau$ profiles are shown for all cirrus (blue), for het cirrus (black), and for hom cirrus (orange).**





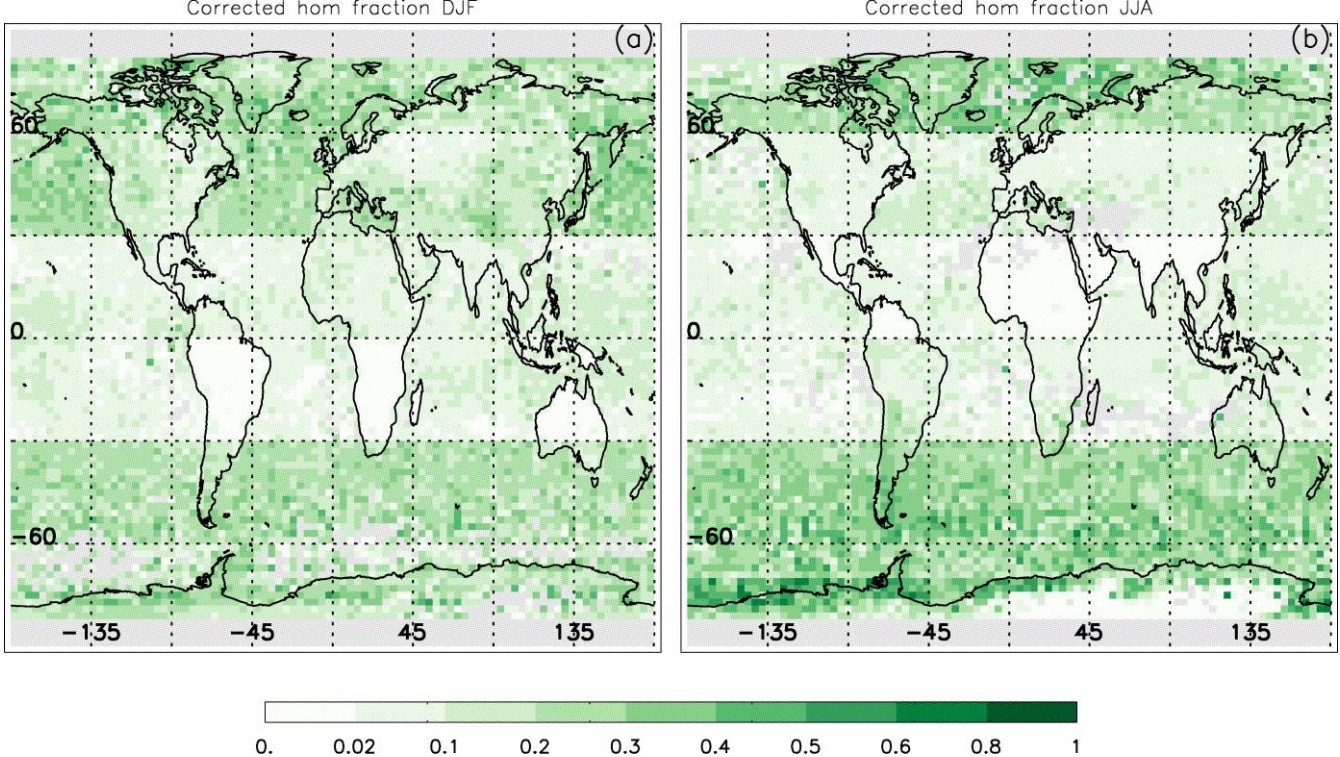

**Figure 20. Geographical distribution of the fraction of cirrus clouds affected by hom during (a) DJF and (b) JJA where the color bar indicates the hom fraction. See text for details.**

where fraction$_{All}$ is the hom fraction using all samples and fraction$_{IAB > 0.01 \, sr-1}$ is the hom fraction using samples with IAB > 0.01 sr$^{-1}$. This correction factor was multiplied by the hom fraction derived from samples having IAB > 0.01 sr$^{-1}$ to produce the corrected hom fractions shown in Fig. 20. For a given grid cell (i.e., column of atmosphere), the $T_r$-dependent hom fractions were weighted by the normalized fraction of samples at each level and vertically integrated to yield the mapped hom fraction. The hom fraction ranges between 0.02 in the tropics and 0.6 at high latitudes in the SH. Relatively few grid cells are light grey to indicate that less than 10 samples constitute that grid cell (see also panels a and b of Fig. A1 for cloud sampling). Since this methodology is based on $T_r$-dependent hom thresholds determined for 30° latitude bands, discontinuities in the hom fraction are common between latitude bands. Thus, the hom fractions in Fig. 20 are qualitative, and there would be a more gradual transition between latitude bands where discontinuities occur.

A case in point is the discontinuity at 60°N in Fig. 20 for JJA, with relatively high hom fractions north of 60°N. As seen in Fig. 17i, the maximum $D_e$ values at the four warmest temperature levels between 221 and 233 K occur at low $\alpha_{ext}$ relative to the other panels, corresponding $N_i$ values are relatively small, and hom fractions in Figs. 18i and 19i range between 0.3 and 0.4. Because this was unexpected, we examine in Fig. 21 the dependence of $N_i$, $D_e$, and IWC on extinction for the four seasons





**Figure 21. Seasonal analysis of the region 60-90° N over oceans for four 4K-temperature levels (219-223 K: panels a-c, 223-227 K: panels d-f, 227-231 K, 231-235 K, panels j-l) in terms of the extinction dependence of $N_i$ (panels a, d, g, h)$D_e$(panels b, e, h, k), and IWC (panels c, f, i, l). Extinction at maximum $D_e$, indicative of het-hom transition, is the smallest in JJA (dark pink) compared to MAM (light pink), SON (light green) and DJF (dark green) in the four temperature levels, yielding the greatest hom fraction during JJA. The internal consistency of $N_i$, $D_e$, and IWC helps to verify this method.**

and the same four temperature levels. Regardless of temperature, the maximum $D_e$ value at low $\alpha_{ext}$ occurs only in JJA, and no marked difference between the three other seasons is observed. In addition, $N_i$ is lowest in JJA (consistent with largest $D_e$ in JJA), with change in slope consistent with the $D_e$ maximum, and IWC is consistent with the $N_i$-$D_e$ behaviour. A possible reason for this low het-hom $\alpha_{ext}$ threshold and resulting high hom fraction during JJA is relatively low INP concentrations




which would allow the RHi in an ascending air parcel to reach the hom threshold more frequently due to lower water vapor removal rates by ice (Krämer et al., 2016). Evidence for lower INP concentrations can be found in the CALIOP dust aerosol optical depth (DAOD) climatology of Song et al. (2021, Fig. 6) where JJA has the lowest DAOD north of 60°N. While dust concentrations were not vertically resolved in this figure, aircraft measured dust concentrations at cirrus levels reported in
Froyd et al. (2022) were lowest during August and October at these latitudes regarding the four months (also including Feb. and May) of the ATOM field campaign. However, estimated dust mixing ratios in Kok et al. (2021) poleward of 60°N from 200-400 hPa show slightly higher values for JJA relative to DJF. Overall, if this interpretation is correct, it suggests that changes in relatively low levels of dust concentration can significantly affect the $D_e$ of cirrus clouds and hence their optical properties.

Also of interest are the seasonal changes in hom fraction between 30°N and 60°N in Fig. 20. Relative dust contributions of the world's main dust source regions are estimated in Kok et al. (2021) and Froyd et al. (2022), with dust from Asian deserts (e.g., Taklimakan and Gobi) more likely to reach cirrus cloud levels in the UT due to ascent within frontal systems, orographic uplift, and dry convection. Asian dust emissions to the UT were maximum in summer (JJA), contributing 59-73% of dust to the NH UT, but with minimal contributions during winter (also see Supplement of Kok et al., 2021). Regions downwind of
these Asian dust sources in Fig. 20 (into North America) undergo a strong seasonal change regarding their hom fraction, being relatively high in winter and relatively low in summer. This may be due to the above noted changes in mineral dust, with higher UT dust concentrations promoting het cirrus clouds. This phenomenon is discussed in greater detail in Mitchell and Finnegan (2009), where a climate intervention method known as cirrus cloud thinning (CCT) was proposed. A similar phenomenon may be occurring in the SH over the Southern Ocean (30-60°S), where the main UT dust source is South America.
Estimated dust concentrations between 200 and 400 hPa are highest during summer (DJF) and lowest during winter (Kok et al., 2021, Supplement). In Fig. 20, hom fractions over the Southern Ocean tend to be lower in summer and higher in winter, consistent with these estimated seasonal changes in dust concentration. To summarize, it appears that an increase in INP concentration pre-empts the formation of hom cirrus by preventing the RHi from reaching the threshold for hom. This phenomenon is the physical basis of CCT.

### 3.4.1 Estimating the fraction of het and hom cirrus comprising in situ and liquid origin cirrus clouds and their estimated radiative impacts

The remote sensing study by Dekoutsidis et al. (2023) found that RHi values in both in situ and liquid origin cirrus clouds peak near cloud top at ~ 140%, high enough to activate hom. The case studies they examined further underscored the importance of hom in liquid origin cirrus clouds, regardless of whether they were convectively generated or produced through frontal
systems (as part of a moisture "conveyor belt"). The methodology described above (Sect. 3.4.1) for identifying het and hom cirrus clouds was applied to in situ and liquid origin cirrus. Only cirrus over oceans were analysed to include optically thin cirrus (so that ~ 0.01 < τ < ~ 3). Zonal means of the cloud fractions are shown in the upper panels of Fig. 22 for DJF (a) and JJA (b), where the thick black histogram separates the het and hom cirrus cloud regimes. Within these two regimes, the fraction





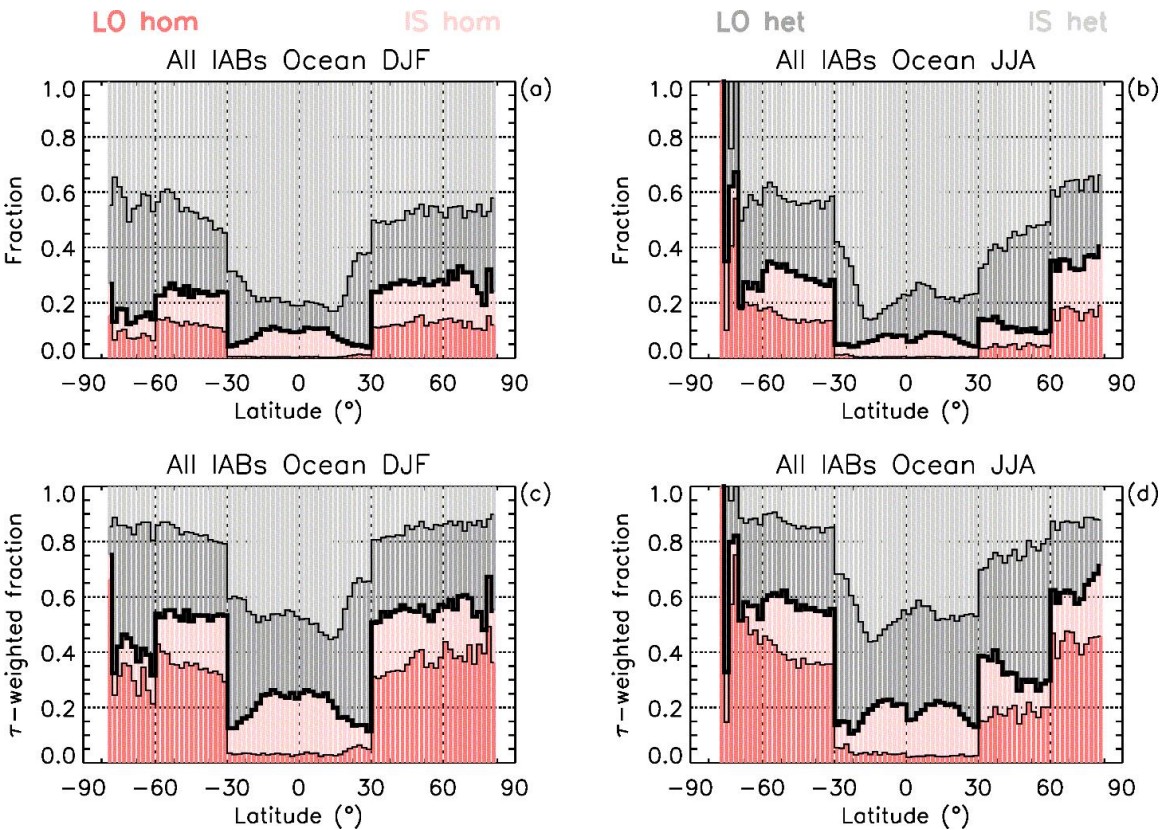

**Figure 22. Upper panels (a and b): the fraction of liquid origin (red) and in situ (pink) hom affected cirrus and of liquid origin (dark grey) and in situ (light grey) het only cirrus vs. latitude (negative in SH) in (a) DJF and (b) JJA based on retrievals over oceans using all IABs (~ 0.01 < τ < ~ 3). The thick black histogram gives the total fraction of hom affected cirrus. Lower panels (c and d): same as upper panels except that the fractions are weighted by cloud optical depth (τ).**

of in situ (IS) and liquid origin (LO) cirrus are indicated. In situ het cirrus are most common, followed by liquid origin het cirrus, although these two cloud types are comparable over the Southern Ocean and Antarctica during summer (DJF). Outside the tropics (± 30°), the hom fraction is generally between 10% and 35%, although over Antarctica during winter it can be much higher. For the hom fraction outside the tropics, the fractions of in situ and liquid origin cirrus are often comparable, although liquid origin dominates south of 60°S latitude during winter.

Panels c (DJF) and d (JJA) of Fig. 22 show for each type of cirrus cloud (i.e., LO hom, IS hom, LO het, IS het) the zonal fraction weighted by optical depth. As in Fig. 1, τ distributions are computed for $\log(\tau)$ between -3 and 1 using bins equal to 0.1 in log space. The τ-weighted fraction of a cirrus type is defined as

$$\tau - \text{weighted fraction (type)} = \frac{\left(\Sigma_i \ p_i \cdot \tau_i\right)_{\text{type}}}{\left(\Sigma_i \ p_i \cdot \tau_i\right)_{\text{all cirrus}}} \tag{7}$$





where $p_i$ is the number of pixels in bin $\tau_i$. The $\tau$-weighted fraction of a cirrus type is a proxy for its relative CRE contribution,
following the same reasoning given for the CRE proxy in Fig. 1. The thick black histograms in panels c and d give the relative
CRE proxy associated with hom cirrus clouds while the relative contributions from in situ and liquid origin cirrus are indicated by
the pink and red colors, respectively. Similarly, the relative CRE proxy associated with het cirrus clouds is indicated by
the grey colors, with relative contributions from liquid origin and in situ cirrus indicated by the darker and lighter shades,
respectively. The main finding here is that, outside the tropics, the radiative impact of hom cirrus clouds appears comparable
to het cirrus clouds. During winter in both hemispheres and over the Southern Ocean in summer, the radiative impact of hom
cirrus could even be greater than het cirrus clouds.

### 3.5     Relating IWC to het and hom cirrus clouds

The results in this study appear generally consistent with those reported in Krämer et al. (2016), titled "A microphysics guide
to cirrus clouds – Part 1: Cirrus types" as shown in their Fig. 5 and summarized in their Fig. 9. That study combined
measurements of cirrus cloud IWC from 17 aircraft campaigns (94 sampling hours) with predictions of cirrus cloud properties
from a detailed microphysical box model. The modelling revealed that the measured median IWC effectively divided cirrus
clouds into two updraft (w) regimes; slow (0.01 to 0.1 m s$^{-1}$) and fast (0.5 to 3.0 m s$^{-1}$). The slow w regime was characterized
by IWC < median IWC, relatively low $N_i$, and relatively large mean mass volume radius $R_v$. Conversely, the fast w regime
was characterized by IWC > median IWC, relatively high $N_i$, and relatively small $R_v$. Interpreting these w regimes as het and
hom regimes, respectively (which was not done in Krämer et al., 2016), this study reports similar findings in Figs. 15 and 16.
Figure 23 compares the temperature-dependence of the retrieved median IWC (and variance as percentiles) with the median
IWC usually near the temperature-dependent het-hom transition (+ signs) as described in Sect. 3.4.1 for cirrus over oceans (~
$0.01 < \tau < \sim 3$). In most cases there is a close correspondence between the median IWC and the het-hom transition IWC. The
tropics and JJA MID NH are exceptions where the $D_e$ maximum was ambiguous (suggesting hom is less active) and the hom
threshold was determined by $N_{thres}$. Excluding these exceptions, the median IWC tends to partition the het and hom cirrus
cloud regimes. This description is different than the explanations given in Krämer et al. (2016), where the median IWC roughly
separates in situ cirrus from liquid origin cirrus (their Fig. 13).

### 4     Conceptualizing cirrus cloud formation and evolution

Figure 24 relates $T_r - T_{top}$ to $\alpha_{ext}$ for various 4 K $T_r$ bins (indicated by the color legend), with the vertical dashed lines giving
the $\alpha_{ext}$ at which the corresponding $D_e$ maximum occurs for the three "warmest" $T_r$ bins. This is based on all IIR samples over
oceans, where $T_r - T_{top}$ is a measure of cloud geometric thickness as discussed earlier. For display purposes, and since hom is
mostly active outside the tropics for $\tau < 3$, only the mid- and high latitudes are shown. It is seen that the correspondence
between the $D_e$ maximum and the $T_r - T_{top}$ maximum for a given $T_r$ curve is usually very close; so close that it is hard to
dismiss it as a coincidence.





**Figure 23.** **Comparing the median IWC (navy blue curves) over oceans (using all samples) with the temperature-dependent het-hom transition point (black triangles) based on the $D_e$ maximum and the corresponding extinction coefficient $\alpha_{ext}$ for a given 4 K temperature interval. Latitude zones and seasons (DJF and JJA) are denoted as in Figs. 18 and 19. The light-blue shade lies between the 10th and the 90th percentiles and the overplotted medium-blue shade lies between the 25th and 75th percentiles**





**Figure 24. Dependence of $T_r - T_{top}$ (which is related to cirrus cloud geometrical thickness) on the extinction coefficient $\alpha_{ext}$ for various 4 K temperature bins indicated by the color bar. Retrievals were over oceans ($\sim 0.01 < \tau < \sim 3$) for the mid- and high latitude zones and two seasons (DJF & JJA) in both hemispheres (NH & SH) during 2008, 2010, 2012, and 2013. Vertical lines indicate $D_e$ maxima for the three "warmest" temperature bins, revealing a correspondence between cloud thickness and $D_e$ maxima. Panels having only two vertical lines result from two $D_e$ maxima having the same $\alpha_{ext}$.**

As for the underlying reason accounting for the close correspondence between the $D_e$ and $T_r - T_{top}$ maxima, the following is proposed. From Eqs. (24) and (26) in Mitchell (1988), it can be shown that for ice clouds at quasi-steady state conditions, the mean cloud updraft w is directly proportional to the downward ice mass flux $\chi_f$, and since IWC $\approx \chi_f/(v_m - w)$ where $v_m$ is the mass-weighted ice fall speed, IWC is also directly proportional to w when $v_m > w$. As shown in Fig. 23, the het regime increases in IWC as it approaches the het-hom transition region, suggesting an increase in w as well. As w increases (for a





given INP concentration), RHi increases within the cloud until the RHi reaches the hom threshold. As IWC increases within the het regime, $D_e$ and $v_m$ also increase, exporting more ice through cloud base that sublimates, humidifying the sub-cloud layer. Over time this increased sedimentation tends to lower cloud base, deepening the cloud. But this cloud deepening ends

once hom becomes important, with $v_m$ decreasing due to smaller ice particle sizes. As w (and thus IWC) increase further beyond the het- hom transition, hom becomes more vigorous and $D_e$ further decreases, decreasing $v_m$ and sedimentation rates, raising cloud base, with geometric cloud thickness becoming minimal at the highest w, IWCs and $N_i$, and at the lowest $D_e$, in the hom regime. This conceptual model is expected to be most apparent over land where the hom-regime decrease in $D_e$ (with increasing w and IWC) is more obvious. This simple explanation is consistent with the retrievals of these quantities and $T_r -$

$T_{top}$. This proposed explanation of Figs. 23 and 24 is summarized in the schematic in Fig. 25, showing the evolution of cirrus cloud thickness with increasing cloud updraft and IWC. Since upward air movements are balanced by downward movements, w in Fig. 25 can be better viewed as the standard deviation of w, $\sigma_w$.

An Arctic cirrostratus modelling study by Spichtinger and Gierens (2009a,b) simulated hom cirrus where ice nucleation occurred at cloud top, with cirrus beginning as geometrically thin cirrus that rapidly deepen with time (i.e., over the course of


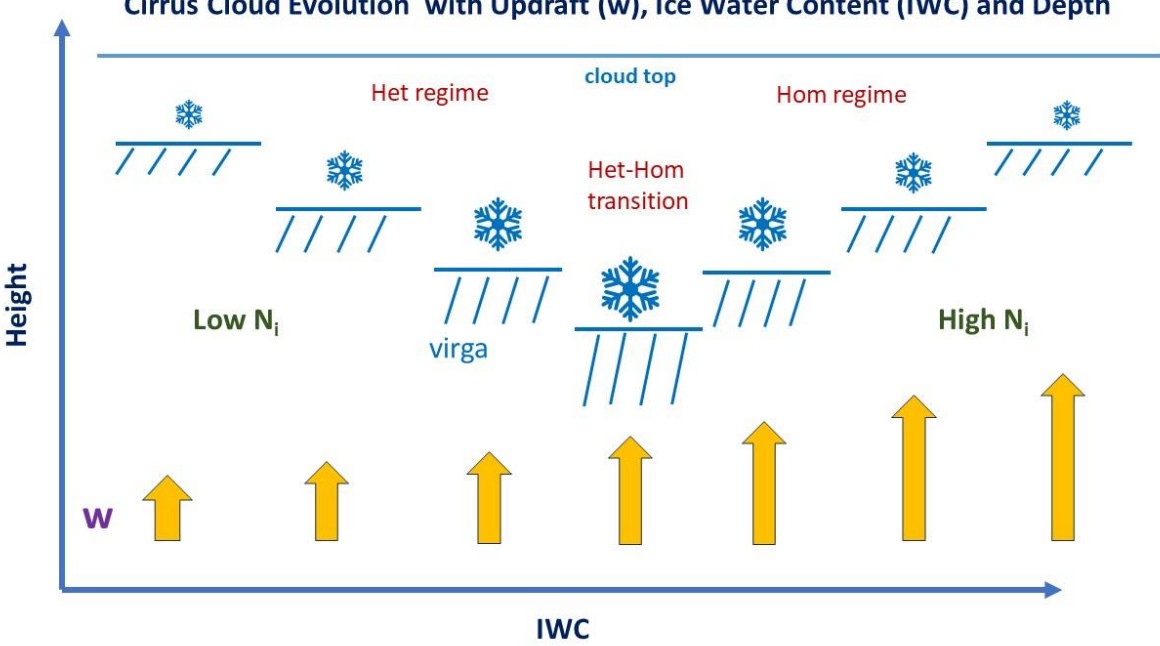

**Figure 25. Proposed evolution of cirrus cloud thickness or depth with increasing cloud updraft (arrows) and IWC, based on results in Figs. 23 and 24. The relative cloud depth is the height difference between the cloud top line and the line segments below (with virga shown below these line segments). The size of the ice crystal symbol represents the relative size of $D_e$. As $D_e$ increases,**
**sedimentation or virga increase, lowering cloud base. Hom acts to decrease $D_e$, decreasing sedimentation and thus raising cloud base.**





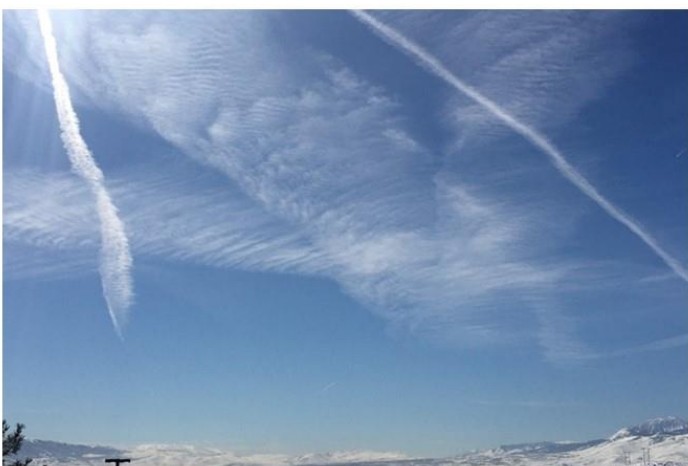
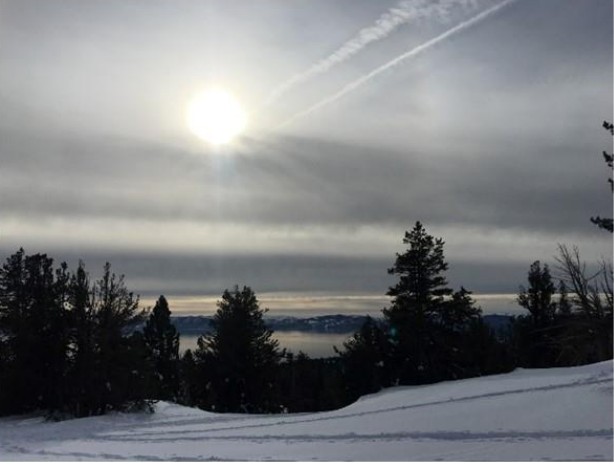

**Figure 26. Examples of het cirrus (and a few contrails) looking south from the Desert Research Institute (left) and hom**
**cirrus associated with a warm front moving over the Sierra Nevada above Lake Tahoe (right).**

an hour). As the cloud deepens, mean $N_i$ (for the cloud layer) decreases since sedimenting and growing ice from cloud top reduces RHi in the mid- to lower cloud, preventing the RHi there from reaching the hom threshold. With only het capable of producing new ice crystals in the mid- to lower cloud, mean layer $N_i$ decreases. This appears generally consistent with the
above description of hom cirrus and with the upper panel results (a-f) in Figs. 8 and 9 (where $N_i$ and IWC are lower in the geometrically thicker cirrus). This explanation is also consistent with Fig. A3(e-f) for $\tau > \sim 0.3$ (where hom cirrus are common), with lower $N_i$ and IWC in the geometrically thicker liquid origin cirrus relative to the thinner in situ cirrus.

     To better relate this work to our experience, example photos of het cirrus and hom cirrus clouds are shown in Fig. 26 where the hom cirrus are optically much thicker. Fig. 27 shows wave cloud hom cirrus forming over the Sierra Nevada mountains.
The thermal infrared (11.2 µm) GOES (Geostationary Operational Environmental Satellites) image of these wave clouds was acquired about 10 minutes before the photo was taken, where these hom cirrus wave clouds extend over much of Nevada and Idaho. This is consistent with Fig. 2 that shows relatively high $N_i$ extending far downwind of mountainous regions.

## 5    Another relevant study

Comparisons between this and another retrieval are discussed in M2024, while one study that relates to this Part 2 study is
presented here. The Particle Analysis by Laser Mass Spectrometry (PALMS) instrument can identify aerosol particles by their chemical "fingerprint," including the residue from sublimated ice crystals, and thus determine whether the ice was nucleated by het or hom (Cziczo et al. 2013; Voigt et al., 2017). But the only PALMS study we know of having global coverage and thus being most relevant for comparison with satellite data is by Froyd et al. (2022), where the PALMS instrument discriminated between mineral dust and other aerosol types. Dust mass and number concentration were determined




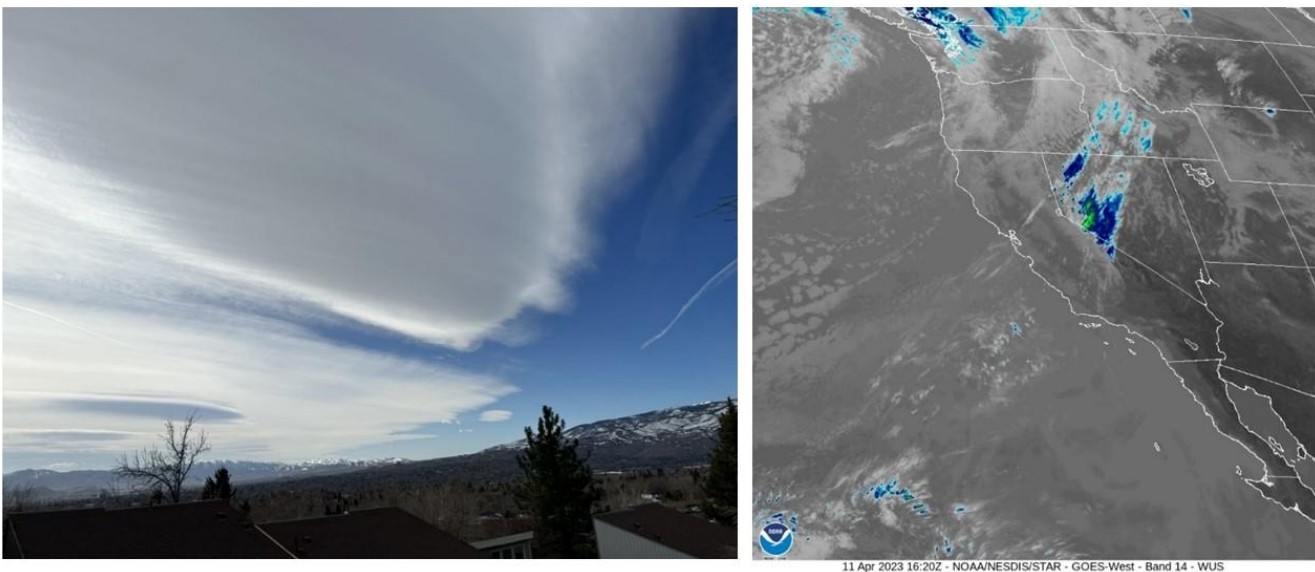

**Figure. 27. Left: A wave cloud example of hom cirrus, forming over the Sierra Nevada mountains next to Reno, Nevada, that are associated with a warm front on 11 April 2023. Right: GOES-18 satellite image (11.2 μm channel) of these cirrus wave clouds acquired about 10 minutes before the photo was taken, revealing their extensive coverage.**


by simultaneous measurements by PALMS and of size-resolved aerosol number concentration. Such measurements were made during four global flights mostly over the Pacific and Atlantic Ocean, from the Arctic to Antarctica and back. To estimate the contribution of het and hom to cirrus formation, these UT dust measurements initialized a detailed cirrus-formation model. It was found that het on mineral dust was responsible for 71% of the cirrus clouds modelled. This is generally consistent with

Fig. 22 outside the tropics. However, comparing Fig. 22 with Fig. 5 and S12 in Froyd et al. (2022) shows that regional differences are larger, where the fraction of hom cirrus in the NH for this study is greater than in Froyd et al. (2022). Note that these two studies are based on very different and somewhat complementary approaches. This CALIPSO study is based exclusively on retrieved cirrus cloud properties while Froyd et al. (2022) use a cirrus-formation model initiated by global dust measurements in the UT. In addition, this study estimates the relative radiative impact of het and hom cirrus clouds through

the τ-weighted hom fraction.

## 6      Representing cirrus clouds in climate models and cirrus cloud thinning

Climate modelling investigations into the climate intervention method known as cirrus cloud thinning or CCT have exposed our lack of understanding about cirrus clouds. CCT is based on the principle that hom cirrus can be seeded with higher INP concentrations (relative to ambient) to convert them to het cirrus. This would allow more outgoing longwave radiation (OLR)



to escape to space since a conversion to het cirrus would (1) reduce the cirrus cloud τ and lifetime (due to larger ice crystals falling faster), (2) cause the cirrus to form at lower altitudes (since the augmented INP require lower RHi to nucleate) where their warming effect is weaker, and (3) the enhanced ice sedimentation rates would enhance the removal of water vapor (a powerful greenhouse gas) from the UT (Mitchell and Finnegan, 2009; Lohmann and Gasparini, 2017). Thus, the physical processes affecting CCT are complex and difficult to model, and numerous CCT modelling studies have produced mixed

results, with some predicting a cooling effect (Storelvmo et al., 2013; Storelvmo and Herger, 2014; Storelvmo et al., 2014; Gruber et al., 2019; Gasparini et al., 2020; Liu and Shi, 2021; Muri et al., 2014; Crook et al., 2015; Kristjánsson et al., 2015; Jackson et al., 2016; Cao et al., 2017; Muri et al., 2018; Duan et al., 2018, 2020) and others predicting a negligible or warming effect regarding the CCT net CRE (Penner et al., 2015; Gasparini and Lohmann, 2016; Gasparini et al., 2017; Tully et al., 2022, 2023). But perhaps the greatest uncertainty in these studies is the role of hom in cirrus cloud formation, since if hom is

not an important process, no significant cooling from CCT is possible. To quote Lohmann and Gasparini (2017), "For the time being, cirrus cloud thinning should be viewed as a thought experiment that is helping to understand cirrus cloud–formation mechanisms". Similarly, the United Nations report (2023) titled "One Atmosphere: An independent expert review on Solar Radiation Modification research and deployment", states "The feasibility of CCT is uncertain, in part because of the larger uncertainties associated with the ice nucleation processes in high clouds". By advancing our understanding of ice nucleation

in cirrus clouds, this study may advance our understanding of CCT's potential.

      The CCT simulations that use the treatment of pre-existing ice (Shi et al., 2015) tend to predict a relatively small or no CCT cooling effect while those CCT simulations that do not use this pre-existing ice treatment predict a substantial cooling (Gasparini et al., 2020; Tully et al., 2023). The pre-existing ice treatment described in Shi et al. (2015) is based on the supersaturation development equation that can be written as:

$\frac{dS_i}{dt} = a_1 S_i W - (a_2 + a_3 S_i)\left(\frac{dq_{i,nuc}}{dt} + \frac{dq_{i,pre}}{dt}\right)$                                   (8)

where $q_{i,nuc}$ is the ice mass mixing ratio due to ice nucleation and $q_{i,pre}$ is the ice mass mixing ratio of pre-existing ice, parameters $a_1$, $a_2$, and $a_3$ depend only on the ambient temperature and pressure, $S_i$ is the supersaturation with respect to ice, W is the updraft velocity and t is time. From this equation it is seen that the greater $q_{i,pre}$ is, the smaller the increase in $S_i$ is. The study by Dekoutsidis et al. (2023) indicates that hom is occurring in a relatively thin layer near cloud top, and the study by Diao et al.

(2015) shows that ice nucleation in cirrus occurs near cloud top. This suggests that $q_{i,pre}$ may be overestimated in GCMs since $q_{i,pre}$ is based on layer mean IWC or $q_i$ values, whereas the actual $q_{i,pre}$ should correspond to a relatively thin layer near cloud top that the vertical resolution in a climate model cannot accommodate. When $q_{i,pre}$ is overestimated, it is much more difficult for $S_i$ to reach the threshold for hom. As stated in Shi et al. (2015), "The pre-existing ice crystals significantly reduce ice number concentrations in cirrus clouds, especially at mid- to high latitudes in the UT (by a factor of ~ 10). Furthermore, the

contribution of heterogeneous ice nucleation to cirrus ice crystal number increases considerably." To address this issue, $q_{i,pre}$ could be attenuated by a factor that best represents $q_i$ near cloud top in the "nucleation zone". The global/seasonal distribution of the predicted hom fraction could be compared against results from this study.





The Community Atmosphere Model version 6 (i.e., CAM6) uses the standard treatment of pre-existing ice, and also considers turbulence kinetic energy as the only dynamic source for in situ ice formation (Lyu et al., 2023). That is, orographic

gravity waves (OGWs) are not considered in ice nucleation. Lyu et al. (2023) demonstrated the importance of including OGWs in CAM6, which triggered more frequent hom in orographic cirrus, increasing $N_i$ and IWC while decreasing $D_e$.

The impact of OGWs on cirrus clouds is evaluated in Appendix A of Tully et al. (2022), where a version of the ECHAM6-HAM GCM is used. Figure A3 of that paper evaluates $N_i$ at 200 hPa for vertical motions with and without orographic effects active. When orographic effects are included, predicted global distributions of $N_i$ appear similar to Fig. 6 of this study featuring

$N_i$ for in situ cirrus clouds.

CCT is potentially most effective at high latitudes during fall and winter when solar insolation is relatively weak (Storelvmo et al., 2014; Mitchell et al., 2020). Based on this study, CCT could have a cooling effect at high latitudes based on the $\tau$-weighted hom fraction. But this cooling could be compromised by a corresponding warming effect resulting from (1) a het cirrus Twomy effect (i.e., where higher INP concentrations produce higher $N_i$ and smaller $D_e$) and (2) a het cirrus IWC/IWP

effect (i.e., where smaller $D_e$ from the Twomey effect produce lower fall speeds, increasing the IWC and IWP as demonstrated in Mitchell et al. (2008)). These two effects may increase $\tau$ and cloud lifetime (Mitchell et al., 2008), increasing the warming effect of het cirrus clouds during Arctic winter. The two opposing radiative effects resulting from higher INP concentrations, the hom cirrus cooling effect and the het cirrus warming effect, need to be investigated by empowering climate modelling with satellite observations like these to better understand aerosol-cirrus cloud-climate interactions. Moreover, radiative transfer

models may employ the CALIPSO retrievals presented here to better understand INP-cirrus cloud-radiation interactions, but these topics are beyond the scope of this study.

## 7    Summary and conclusions

This study dealt with the identification of relevant cloud physical parameters to identify hom-affected cirrus clouds and further quantify their frequency of occurrence. From a global observational purview, this was done for the first time by Froyd et al.

(2022) who used global measurements of dust in the UT to initialize a detailed cirrus cloud formation model. In contrast, our study is based on retrieved cirrus cloud properties and does not involve modelling.

In Section 2, global-seasonal maps for IAB > 0.01 sr$^{-1}$ ($\tau$ > ~ 0.3) show relatively high $N_i$ and relatively low $D_e$ at high latitudes during winter, especially over mountainous terrain. Orographic gravity waves induced by mountains provide stronger updrafts that promote hom, which may help explain these results. Gryspeerdt et al. (2018) and M2018 reported similar findings

and explanations. This winter phenomenon was also observed at midlatitudes over mountainous regions. Similar maps for IWC and $T_r$ show that IWC tends to track $T_r$, consistent with expectations from the Clausius-Clapeyron equation. Similar maps of $N_i$ and $D_e$ restricted to in situ cirrus exhibit these same patterns but with higher $N_i$ and smaller $D_e$, possibly resulting from their lower temperatures and lower geometrical thickness as described in Sect. 4.



The attribution study of het and hom affected cirrus clouds (Sect. 3) began with determining which cloud properties were
most sensitive to hom by exploiting a finding that hom was most active (based on $N_i$) in the geometrically thinnest cirrus
clouds (i.e., having the smallest $T_r - T_{top}$). Other than $N_i$, IWC was most sensitive to hom (with IWC increasing with hom
activity), followed by $D_e$ (mostly over land, with $D_e$ decreasing with increasing hom activity). Therefore, the extinction
coefficient $\alpha_{ext}$ for visible radiation was used to separate het and hom cirrus clouds since $\alpha_{ext}$ includes the IWC/$D_e$ ratio.
Comparisons of our retrievals of $N_i$, IWC, and $D_e$ in the $T_r$ vs. $\alpha_{ext}$ space with predicted corresponding values for pure hom
conditions (based on the Clausius-Clapeyron equation and simple hom theory) revealed that maximum retrieved values for
IWC and $N_i$ (for a given $T_r$) coincided with theory, verifying the success of $\alpha_{ext}$ for separating het and hom-affected cirrus
clouds (as well as validating the retrievals).

With this understanding, a method was developed to quantify the contribution of cirrus clouds affected by hom. This
involved analysing variations of $D_e$ against $\log(\alpha_{ext})$ where $D_e$ exhibited a maximum typically at $\alpha_{ext} > 0.3$ km$^{-1}$ for $T_r = 233$ K,
with the maxima tending to occur at lower $\alpha_{ext}$ with decreasing $T_r$. This $D_e$ maximum was interpreted as a transition from het-
only cirrus to hom-affected cirrus, with $D_e$ decreasing with increasing $\alpha_{ext}$ in the hom regime due to higher $N_i$ resulting from
hom. That is, the greater competition for water vapor due to higher $N_i$ in the hom regime reduces ice crystal growth rates,
reducing $D_e$ with increasing $N_i$. This method insures that hom always has an unambiguous impact on cloud microphysical and
radiative properties. Most of the quantitative results are over oceans where retrieval uncertainties are smaller than over land,
owing to smaller uncertainties in surface parameters. The main findings of this het-hom attribution study are as follows:

1.  Outside the tropics over oceans, zonal means of the fraction of hom-affected cirrus clouds during winter were
    generally between 25% and 30%, with a comparable contribution from in situ and liquid origin cirrus, similar to the
    findings of Dekoutsidis et al. (2023).

2.  The relative contribution of hom affected cirrus in terms of radiation impact was estimated using a proxy for the cloud
net radiative effect (CRE) inferred from $\tau$ distributions as a $\tau$-weighted occurrence frequency. Using this proxy, we
    found that cirrus clouds with $0.3 < \tau < 3$ represent 78% to 87% of the radiative weight for all cirrus clouds sampled
    depending on latitude, which compared favorably with results from Hong et al. (2016). Using this proxy, the $\tau$-
    weighted fraction for hom affected cirrus over oceans outside the tropics during winter was > 50%, indicating that
    hom cirrus may contribute substantially to the Earth's radiation budget.

3.  Based on two earlier studies that evaluate mineral dust contributions from the main dust source regions to different
    latitude zones during different seasons, the fraction of hom cirrus was lowest downwind of the Asian dust source
    regions during summer when UT dust concentrations were highest. Similarly, the fraction of hom cirrus over the
    Southern Ocean was lowest during summer when dust concentrations were highest (relative to winter). This suggests
    that the hom fraction is sensitive to mineral dust concentration and this may be a natural analogue to the climate
intervention proposal known as cirrus cloud thinning (Mitchell and Finnegan, 2009).





4.  Hom affected cirrus were found to have a median $N_i$ typically $\geq 200$ L$^{-1}$ (see Fig. 18) and $\tau$ is much larger than in het-only cirrus (Fig. 19). To a first approximation, the median IWC divides the het and hom cirrus cloud regimes outside the tropics (Fig. 23).

5.  When plotting $D_e$ and $T_r - T_{top}$ against $\alpha_{ext}$, it was found that their maxima occurred at roughly the same $\alpha_{ext}$. A simple theory of cirrus cloud development was postulated to explain this observation.

6.  Contrasting $N_i$ and $D_e$ vs. $\alpha_{ext}$ plots for oceans and land in the het regime, it appears that higher INP over land may be producing a "Twomey effect" in het cirrus clouds over land. However, in the hom regime, higher INP concentrations appear to reduce the hom fraction and thus substantially reduce cirrus cloud optical depth.

## 8    Appendix A: Global distributions and properties of in situ and liquid origin cirrus clouds

Figure A1 shows the geographic sampling distribution of 1-km$^2$ IIR pixels for two periods: DJF and JJA. Optically thicker cirrus (IAB > 0.01 sr$^{-1}$) that can be measured over both land and oceans are featured in the upper panels (a – b) while sample counts for all cirrus over oceans with valid retrieved effective emissivities at both 10.6 µm and 12.05 µm (~ 0.01 < $\tau$ < ~ 3) are featured in the lower panels (c – d). The higher counts in (c – d) are mostly due to in situ cirrus clouds.

Figure A2 (a-c) shows the abundance of in situ and liquid origin cirrus as a function of cloud radiative temperature $T_r$ for three latitude bands (combining NH and SH results), discriminating between optically thick cirrus (1) over land and (2) over oceans (IAB > 0.01 sr$^{-1}$; ~ 0.3 < $\tau$ < ~ 3) and (3) all retrieved cirrus clouds over oceans (~ 0.01 < $\tau$ < ~ 3). Figure A2 (d-f) shows the fraction of in situ cirrus clouds [in situ/(in situ + liquid origin)] as a function of $T_r$. The overall (total) fraction of in situ cirrus clouds for the above three cloud classes is indicated in each panel by the fraction and adjacent line pattern (or color). Temperature bins are 4 K resolution. Note that we count 1-km$^2$ IIR pixels, and that a cirrus cloud extending 20 km horizontally along a portion of the lidar track is counted 20 times whereas a cirrus cloud extending 5 km along this track is counted only five times. Using the above definitions, Fig. A2 (d-f) shows that the fraction of all retrieved in situ cirrus clouds over oceans reaches ~ 50% for $T_r \approx 227$ K for all latitude zones. These findings can be compared with those reported in Fig. 4 of Dekoutsidis et al. (2023) over Europe, where it is evident that the occurrence of in situ cirrus prevails over liquid origin cirrus for $T_r < 221$ K. Similarly, Fig. 13 in Luebke et al. (2016) shows that for Europe, the transition from in situ cirrus dominating to liquid origin cirrus dominating occurs between 218 K and 222 K. These studies are fairly consistent with this one given the different methods used to discriminate between in situ and liquid origin cirrus clouds. As noted in Sect. 2.1, our method may overestimate the occurrence frequency of in situ cirrus clouds. The overall fraction of in situ cirrus for all sampled cirrus over oceans ranges from 86% in the tropics, to 62% in the midlatitudes, and 55% in the high latitudes.





**Figure A1. Geographic sampling distribution of optically thicker cirrus clouds (IAB > 0.01 sr⁻¹) during DJF (a) and JJA (b), where the number of sample counts is given by the color bar. Similarly, this sampling distribution is given for all cirrus clouds over oceans only (~ 0.01 < τ < ~ 3) during DJF (c) and JJA (d). Analysis for years 2008, 2010, 2012, and 2013.**







**Figure A2. Top panels (a-c):** Global annual mean radiative temperature distributions of sampled in situ (triangles) and liquid origin (diamonds) cirrus clouds arranged horizontally by latitude zone: (a) tropics, (b) mid- and (c) high latitudes, NH and SH combined. Distributions were built with a 4 K temperature step. Red curves indicate IAB > 0.01 sr⁻¹ over land (dashed) and oceans (solid), while blue curves show results for all samples over oceans. **Bottom panels (d-f):** Red curves show the fraction of in situ cirrus clouds derived from upper panel results for IAB > 0.01 sr⁻¹ over land (dashed) and oceans (solid) while blue curves show the same but for all samples over oceans only. Numbers indicate the overall fraction of in situ cirrus clouds for each cloud category (indicated by corresponding line pattern or color). Analysis for years 2008, 2010, 2012, and 2013.

A major motivation for investigating these two types of cirrus cloud is that their different origins may endow them with different microphysical and radiative properties. This assumption is tested in Fig. A3 which contrasts the $T_r$ dependence of $N_i$, $D_e$ and IWC for in situ and liquid origin cirrus clouds over oceans for two $\tau$ classes: (1) $0.01 < \tau < 3$ and (2) $0.3 \leq \tau \leq 3$, where (1) is for all IIR samples and (2) provides the lowest uncertainties. This analysis is for winter at 30° N-60° N latitude. Class (1) may be more representative of aircraft measurements that do not discriminate between thin and thick cirrus, while class (2) is for relatively optically thick cirrus. These results are discussed in the last paragraph of Sect. 2.3. Based on all samples over oceans (blue curves), the main differences between in situ and liquid origin cirrus are their IWPs and $\tau$, which





**Figure A3.** Retrieval results contrasting in situ (triangle symbols) with liquid origin (diamond symbols) cirrus cloud properties for the NH midlatitudes over oceans during DJF. The blue curves are based on all samples (~ 0.01 < τ < ~ 3) while the red curves have IAB > 0.01 sr$^{-1}$ (~ 0.3 < τ < ~ 3). Shown for each cirrus cloud type and range of τ are (a) number of retrieval samples, (b) median $D_e$ (μm), (c) median IIR equivalent thickness (km), (d) median geometric thickness (km), (e) median $N_i$ (L$^{-1}$), (f) median IWC (mg m$^{-3}$), (g) median τ, and (h) median IWP (g m$^{-2}$).

are greater for liquid origin cirrus. The geometric thickness ($\Delta z$) in Fig. A3 (panel d) is the vertical geometrical thickness of the cloud layer measured by CALIOP while the IIR equivalent thickness ($\Delta z_{eq}$, panel c) accounts for the fact that the IIR instrument does not equally sense all of the cloud profile that contributes to thermal emission (M2018; Garnier et al., 2021; M2024). This is accounted for through the IIR weighting profile as illustrated in M2024, which gives more weight to higher extinctions in the cloud profile. Both median $\Delta z$ and $\Delta z_{eq}$ are larger for liquid origin cirrus, partly explaining their higher IWPs and $\tau$.

## 9 Data availability

The CALIPSO IIR Level 2 data products used in this study are available at the NASA Langley Atmospheric Science Data Center and can be retrieved from https://doi.org/10.5067/IIR/CALIPSO/CAL_IIR_L2_Track-Standard-V4-51. The CALIPSO Lidar Level 2 cloud profiles used in this study are available at the NASA Langley Atmospheric Science Data Center and can be retrieved from https://doi.org/10.5067/CALIOP/CALIPSO/CAL_LID_L2_05kmCLay-Standard-V4-51. These CALIPSO data products are also available from the AERIS/ICARE Data and Services Center in France (https://www.icare.univ-lille.fr/).

## 10 Author contribution

DM and AG conceived the study; AG accessed and analysed the CALIPSO data and contributed to the writing of the paper.

## 11 Competing interests

The authors declare that they have no conflict of interest.

## 12 Acknowledgements

This research was supported by NOAA grant NA22OAR4690640 and by the NASA CALIPSO project. The authors are grateful to the Centre National d'Etudes Spatiales (CNES) and to the AERIS/ICARE Data and Services Center in France for their support with the CALIPSO IIR data. We are grateful to Dr. Martina Krämer for asking us to produce Fig. 23. We also thank Dr. Ehsan Erfani for his assistance with the SPARTICUS data analysis and for his participation and suggestions in scientific discussions.



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
