# Peer review of "Advances in CALIPSO (IIR) cirrus cloud property retrievals – Part 2: Global estimates of the fraction of cirrus clouds affected by homogeneous ice nucleation"

_EGUsphere, 2024_

## Referee Comment (RC2)

Advances in CALIPSO (IIR) cirrus cloud property retrievals – Part2:
Global estimates of the fraction of cirrus clouds
affected by homogeneous ice nucleation

David L. Mitchell and Anne Garnier

**General:**
The manuscript contains a new global climatology of cirrus clouds derived from satellite observations. It is an impressive work with very extensive analyses of various properties of cirrus, as cloud ice particle number concentration ($N_i$), effective diameter ($D_e$), ice water content (IWC), shortwave extinction coefficient ($\alpha_{ext}$), optical depth ($\tau$), and cloud radiative temperature. The study includes innovative data analyses that lead to new perspectives and a deeper understanding of cirrus clouds. In particular, the observations are analyzed to determine whether the cirrus formed homogeneously or heterogeneously. Further, the fraction of hom-affected cirrus clouds is determined and $\tau$ distributions are used to establish a proxy for cloud net radiative effect (CRE) of the hom affected cirrus. Finally, a conceptual model of cirrus cloud characterization is proposed. Altogether, this study has the potential to become a new standard work on cirrus properties.

Unfortunately, however, I have some concerns, which I will list in the following. I know that the authors are experienced scientists with many publications and therefore write their articles the way they like it best. Nevertheless, I would like to add some comments, because I feel that otherwise the extensive and thorough study may not get the attention it deserves.

**(G 1)** It took me quite a while to work through the long, sometimes complicated text and the equally complicated figures. To my opinion, the interesting, but complex results could be presented more simply and shorter to make them easier for the reader to understand. Otherwise, I fear readers will be discouraged from reading the article.

So, overall, I think it might be good to consider shortening the main part of the paper and only showing the most important figures in that part. Everything else could be moved to the Appendix or Supplementary Material.

**(G 2)** I have some suggestions for simplifications - but not for the text, for that I can only ask the authors to go through the manuscript again and simplify and shorten the descriptions.

For example, I recommend

**(a)** to introduce a small table with abbreviations, which contains for example:

$T_r$ : cloud layer radiative temperature, approximately in the middle between $T_{top}$ und $T_{bottom}$

IAB: CALIOP 532 nm layer integrated attenuated backscatter

IAB < 0.01 sr -1   =   ~ 0.01 < $\tau$ < ~ 0.3   optically thinner cirrus – thin - subvisible cirrus*
IAB > 0.01 sr -1   =   ~ 0.3 < $\tau$ < ~ 3     optically thicker cirrus – opaque cirrus

* subvisible cirrus ($\tau$ < 0.03), thin cirrus (0.03 < $\tau$ < 0.3)

**(b)** to use either the τ ranges throughout the manuscript to identify the cirrus type, or (I think even better) the name (subvisible cirrus, thin cirrus, opaque cirrus), IAB doesn't give an intuitive impression on the type.

**(c)** to simplify the figures:
I strongly recommend revising all figures so that the recurring headings above each panel be incorporated into a general figure title, so that only the specific information appear above the panels (in the current version it is hard to find out the differences between the panels). Further, I would also include information that is now somehow hidden in figure captions in the Figure title.
As an example of the simplification of the figures here **Figure 18 (**I modified the figure for my own understanding):

[Figure]

**(G 3)**   Retrieval of liquid origin cirrus

Line 110 ff: … *cirrus clouds with $T_{base}$ warmer than 235 K (and $T_r$ colder than 235K), hereafter called liquid origin cirrus.* …
*This method is an approximation that may underestimate liquid origin cirrus clouds somewhat (overestimating in situ cirrus) since cloud condensate from below the 235 K isotherm may be advected across this isotherm upwind of the CALIOP nadir view when there is no cloud at nadir below this isotherm.'*

This method sorts not only liquid origin as in-situ origin cirrus, but likely also in-situ origin as liquid origin, as explained in the following: Warm conveyer belts (but also convective systems) consist from bottom to top of layers of  liquid, mixed-phase and cirrus clouds. The mixed-phase clouds appear in the cirrus region as liquid origin clouds, but above these, in-situ cirrus usually also form due to the lifting of the air masses. An example is shown by Luebke et al. (2016) (see Figure below, top panel). The vertical structure of liquid origin and in-situ origin cirrus is clearly recognizable.

 If these clouds were classified as described in this paper (i.e. if the clouds reach down to temperatures warmer than -38C they are liquid origin cirrus), the whole in-situ origin cirrus umbrella would be misclassified as liquid origin cirrus.

 I would recommend doing some case studies to test the classification. A trajectory analysis, as  done for example by Luebke et al. (2016), would be best suited for this. This is the most reliable method to classify cirrus of in-situ and liquid origin. I think it is crucial to check the classification method, as all results on in-situ origin and liquid origin cirrus depend on the correctness of the sorting - and my concern is that many of the in-situ cirrus at the top  of WCBs, MCS or convective cells will be classified as liquid origin.

[Figure]

**Figure 4.** Examples of CLaMS-Ice simulations from ML-CIRRUS showing a liquid origin cloud sample (top) from the 11 April flight and an in situ origin sample (bottom) from the 7 April flight. The flight path is illustrated by the black line and represents the pressure at which the aircraft was flying (*y* axis) and the distance since take-off (*x* axis). The colors in each plot represent the simulated IWC (orange: high IWC, blue: low IWC). Grey areas indicate $T > 238$ K and do not contain simulated clouds.

**(G 4)** Hom-affected and het-only cirrus clouds (Sections 3.2 and 3.3 etc.)

E.g. Line 293ff: *Characteristic in all plots* (Figure 12 and subsequent figures) *is a broad region on the left side (relatively low $\alpha_{ext}$) where $N_i < 30$ $L^{-1}$, apparently corresponding to het only. To the right of this region is a gradient of increasing $N_i$, culminating in values of $N_i > 2000$ $L^{-1}$. This gradient region is likely produced by varying degrees of hom activity.*

I am not convinced by this classification and would interpret this central point of the paper differently, as I will explain in the following.

I agree that there are two cirrus regimes, as described in Krämer et al. (2016, 2020). Here $\alpha_{ext}$ (color coded by $N_i$) is used to make this visible. The region of high $\alpha_{ext}$ and $N_i$ ('hom acitivity') corresponds approximately to the area of high IWC and Ni in Fig. 6 (top panel) of Krämer et al. (2020). In this region, both in-situ and liquid-origin cirrus clouds are present. For the in-situ cirrus clouds, the interpretation that they are formed by hom (of soluble aerosol particles) is correct, but not for the liquid-origin cirrus. Hom (of cloud drops) can occur within liquid origin cirrus clouds, but is rather rare outside the tropics. The predominant freezing mechanism of liquid origin cirrus is het, nevertheless, they can have high $N_i$.

The region with low $\alpha_{ext}$ and $N_i$ is defined here as 'het only'. However, I think the composition of the cirrus clouds in this region are much more complex. First to mention, in this region there are also also both in-situ and liquid origin cirrus present. Further, the in-situ origin cirrus could have formed either hom or het, since hom also produces only few ice crystals at warm temperatures and low updrafts. But, most importantly, the concentration of hom cirrus with initially high $N_i$ (and thus $\alpha_{ex}$) decreases quite rapidly in the warming phases of the ubiquitous mesoscale temperature fluctuations where the environment is subsaturated (Jensen et al., 2024). This means that they are moving from the hom affected regime to what is now defined as het only. This can be seen also in Fig. 6 (bottom panel) of Krämer et al. (2020) - the thinner the cirrus (and the lower $N_i$) the more frequent the cirrus clouds are in a subsaturated environment.
The fact that the cirrus clouds with low $\alpha_{ext}$ and $N_i$ are in a subsaturated environment is also indicated by the decreasing $D_e$ to the left of $D_{max}$ (Figures 15 and 16), because under this condition, the thinner the cirrus clouds and the lower the $N_i$, the smaller the ice particles.
In an at least saturated or supersaturated environment, the ice particles would be larger with decreasing Ni, i.e. there would be no maximum in $D_e$, but an increase, maybe with a change of the slope during the transition from one to the other regime .
In summary, I believe that this region is a mixture of in-situ origin cirrus clouds of different ages, which could have formed either het or hom, and aged liquid origin in the dissolution stage.
I recommend reconsiding the naming and the discussion of the 'het only' cirrus regime.

What I wonder (although I know it would be a lot of work) is whether this analysis would be better done separately for in-situ and liquid origin cirrus (the derivations presented in section 3.2 only apply to in-situ cirrus anyway)? Especially for in-situ cirrus, the interpretation of the freezing mechanisms would be much clearer, now the liquid origin probably blurs their features.

**Specific comments:**

**(S 1) Line 35f:** … *liquid origin cirrus associated with cloudy air advected from lower levels (T>235 K) that often contains liquid cloud droplets.*

I don't think cirrus of liquid origin often contains liquid droplets - they are usually completely glaciated by the Wegener-Bergeron-Findeisen process when they enter the cirrus temperature regime (< -38C; see Costa et al., 2017, their Figure 15). Only at high vertical velocities (as in convective cirrus), or in the absence of INPs (e.g. in the Arctic), liquid droplets can rise to temperatures as low as -38 C, where they freeze homogeneously.

**(S 2) Line 51f**: *However, hom resulting from cloud droplet freezing dominated ice production in the lower part of cirrus clouds at all latitudes.*

This is not consistent with the measurements of Costa et al. (2017), see point (2).

**(S 3)** All kinds of studies are cited in the introduction, but not with a specific focus. Then it is said - even without focus - what is in the paper.

**(S 4)  Line 130-132:** *When only clouds with τ>~S0.3 are sampled over oceans* (solid red lines), *liquid origin cirrus clouds prevail at mid- and high latitude (60 % and 70 %, respectively), but not in the tropics (32%).*

I think these numbers 60, 70 and 32% can be derived from those in the respective panels by taking the difference to 1, right? That is not easy to understand - please mention it in the text or write both numbers (for in-situ and liquid origin) in the panels.

*When all clouds are considered* (solid blue lines)*, the percentage of in situ cirrus increases by 18 to 25 % and they always prevail.*

I can't find these numbers in the panels …

**(S 5) Line 132ff:** *For these blue curves, it is seen that the liquid origin cirrus prevail at Tr larger than about 227 K, which is ~ 6 K higher than shown in Luebke et al., 2016 (their Fig. 13) and Dekoutsidis et al., 2023 (their Fig. 4). …*

Note that the analysis of  Luebke et al. (2016) and Dekoutsidis et al. (2023) are based on the same field experiment (ML-Cirrus) and represent only the meteorological conditions that prevailed   during   that   time.
I would recommend to compare the in-situ / liquid origin fractions with the analysis of Wernli et al. (2016), which covers 10 years of ERA5 data.

**(S 6) Line 175:** *Global maps for each season are shown for median Ni, De, IWC, and Tr using the cloud sampling criteria described in section 2.1 and IAB ≥ 0.01 sr-1 (i.e., ~ 0.3 < τ < ~ 3,* thick cirrus*) …*

Please introduce the cirrus category ‚thick cirrus', also in the Figure captions (or even more simply as the title of the figures: ‚Median Ni / De / IWC / Tr ,  Tr < 235 K, thick cirrus (~ 0.3 < τ < ~ 3)', then only the time of year appears above the individual panels.

**(S 7) Line 193ff (Figure A3):**

  **\* panel b:** I wonder why the in-situ $D_e$ is of comparable size to that of  liquid origin?
                I think it should be smaller, because in-situ ice particles cannot grow  as large as
                ice particles of liquid origin.
   **\* panel e:** I wonder why the in-situ $N_i$ of thick cirrus (red curves) increases at warm T -
                could this be a misclassification?
   **\* panel f and h** (IWC and IWP): I wonder why the in-situ IWC and IWP  (red curves) is of
                comparable size to the liquid origin ones? I think they should be smaller.
                Misclassification?

**(S 8) Line 237f:**  *... two different τ categories: ~ 0.01 < τ < ~ 0.3 (IAB < 0.01 sr-1) and ~0.3<τ<~3 (IAB>0.01sr-1); henceforth categories 1 and 2.*

Instead of categories 1 and 2, you could say 'thin cirrus' and ‚thick cirrus', which is more specific  and  informative.

**(S 9)  Line 251f:**  *… highest* (IWC) *values in* (geometrically) *thinner clouds in Category 2 for a given Tr, were not anticipated.*

Possibly the higher IWC, especially at warmer temperatures in the geometrically  thinner cirrus clouds, indicates that these are young cirrus which have not yet lost any ice particles through evaporation in temperature fluctuations  (see Jensen et al., 2023). During aging ice particles are lost by  evaporation and sedimentation, so the geometrical thickness increases and the IWC decreases.

**(S 10)  Line 253ff:** *For a given $T_r$, $D_e$ tends to be quasi-constant, although usually decreasing for the thinnest clouds in both categories,  possibly due to entrainment. But this $D_e$ decrease could also be due to hom in Category 2* (thick cirrus)...

It could also be that the larger ice crystals in geometrically thicker cirrus clouds, especially in the tropics at warm temperatures, indicate liquid origin cirrus clouds. I think that is more likely than the occurrence of  hom.

**(S 11)  Line 440ff:** *Most evident when comparing Figs. 15 and 16 for $α_{ext}$ < 0.3 km$^{-1}$ (where het is expected to prevail) is that median $N_i$ is higher over land (up to a factor of 10), presumably due to higher INP concentrations over land. ...*

Or stronger updrafts → enhanced hom over land ?

*…. higher INP over land (which can also be enhanced by stronger updrafts) may be producing a "Twomey effect" in het cirrus clouds over land.*

This is very speculative (over-interpreted?) .... to make this hypothesis  more information about INP and updrafts  would be necessary.

**(S 12)  Line 476f:**  *When $D_e$ > sensitivity limit, we set the sample as het-only.*

Here I have strong concerns, as outlined in point **G 4**.

**(S 13)  Line 550ff:**  *Also of interest are the seasonal changes in hom fraction between 30°N and 60°N in Fig. 20. Relative dust contributions of the world's main dust source regions are … more likely to reach cirrus cloud levels in the UT due to ascent within frontal systems, orographic  uplift, and dry convection. ...*

As in point S 11,  seasonal changes  in hom fraction might also be due to changes in updrafts and not only to be related to INP.

**(S 14)  Line 575, Figure 22:**   I wonder if it wouldn't be easier to understand if LO and IS were the same color (LO grey, IS red) and hom / het the color shades ? And then to plot LO (het/hom) over IS (het/hom).

**(S 15)  Line 604f:** *Interpreting these w regimes as het and hom regimes, respectively (which was not done in Krämer et al., 2016) ….*

It has been discussed by Krämer et al. (2016) that het dominates in slow updrafts with low IWCs and hom in fast updrafts with high IWCs, see their Figure 6 and corresponding text.

**(S 16)  Line 650, Figure 25:**  Considering point G 4, does this scheme fit?

I'll stop commenting here (but have read the rest of the paper); I think there are enough points to be revised, after which the remaining parts could have changed. So I will wait for the next version of the manuscript.

**References:**

Costa, A., Meyer, J., Afchine, A., Luebke, A., Günther, G., Dorsey, J. R., Gallagher, M. W., Ehrlich, A., Wendisch, M., Baum gardner, D., Wex, H., and Krämer, M.:
Classification of Arctic, midlatitude and tropical clouds in the mixed-phase temperature regime,
Atmospheric Chemistry and Physics, 17, 12 219–12 238,
https://doi.org/10.5194/acp-17-12219-2017, 2017.

Jensen, E. J., Kärcher, B., Woods, S., Krämer, M., & Ueyama, R. (2024).
The impact of gravity waves on the evolution of tropical anvil cirrus microphysical properties.
Journal of Geophysical Research: Atmospheres, 129, e2023JD039887.
https://doi.org/10.1029/2023JD039887

Wernli, H., M. Boettcher, H. Joos, A. K. Miltenberger, and P. Spichtinger (2016),
A trajectory-based classification of ERA-Interim ice clouds in the region of the North Atlantic storm track, Geophys. Res. Lett., 43, 6657–6664,  doi:10.1002/2016GL068922.

---

## Author Comment (AC1)

**Author responses to the Reviewer 1**

Format: The reviewers' comments are in normal font while author responses are in red font. Text in red font italics indicates revised/added text in the revised manuscript.

We understand that reviewing this paper took a lot of time and effort, and we sincerely thank you for your comments that have improved this paper. Below are our responses to the general and specific comments:

**General comments from Reviewer 1:**

This manuscript analyzes cirrus cloud properties using a satellite retrieval product. It introduces a criterion for distinguishing between microphysical formation mechanisms of cirrus, often based on simple thermodynamic theory. Notably, the study not only estimates the fraction of homogeneously formed cirrus but also assesses their optical depth-weighted contribution, providing a more accurate estimate of their radiative and climatic relevance.

This manuscript has the potential to make an important contribution to our understanding of cirrus clouds. However, several issues undermine its impact, mainly related with the manuscript length, logic, and figure presentation.

Key comments:
1. Excessive manuscript length (particularly figure number)

The manuscript includes 27 figures, many with 8-12 subplots. This abundance, coupled with occasional topic detours, dilutes the key messages and diminishes the paper's relevance for the community. Reducing the number of figures (or at least figure panels) while focusing on the paper's key results is recommended. For instance, tropical data could often be excluded, as it is not the main focus of the study.

- For the maps (Figs. 2 – 7 in preprint), only DJF and JJA are now shown in the main paper. Maps for MAM and SON have been moved to the Supplement. Colormap concerns have been addressed in these figures. Figures 6 and 7 have been moved to the Supplement.
- Figures 8 – 11 have been condensed into a single figure (Fig. 6) that illustrates the impact of hom on $N_i$, IWC, & $D_e$ for midlatitude winter only. Other latitude bands and seasons are similar (noted in text).

- Figures 12 & 14 have been replaced by Fig. 7 (comparing hom theory with $N_i$ retrievals + variation in sample density), featuring winter only, and by separating in situ and warm base clouds.  Land retrievals are featured in the Supplement (previous Fig. S2 replaced with new Fig. S7).
- Figure 13 is replaced by Fig. 8 (comparing hom theory with IWC and $D_e$ retrievals + variation in sample density), featuring winter only.  Land retrievals are featured in the Supplement (previous Fig. S1 replaced with new Fig, S8).
- Figure 21 (Fig. 15 in revised manuscript) has been changed to show only one temperature bin (i.e., 229 K) and the text now states that this same behavior is seen at the other temperatures:

*"Because this was unexpected, we examine in Fig. 15 the dependence of $N_i$, $D_e$, and IWC on extinction for the four seasons for the temperature bin at 229 K.  Other temperature intervals (having a mid-temperature of 233, 225, and 221 K) exhibited the same behavior."*

- Figure 26 has been removed.

This reduces the number of figures in the main paper from 27 to 20 and restricts all seasonal comparisons to winter vs. summer or DJF vs. JJA.

To improve readability/simplify the manuscript, the authors could limit the main text's analysis to a cloud optical depth range of 0.3 to 3, moving additional analysis to an appendix.

We have simplified the manuscript, but to restrict the analysis to an optical depth (OD) range of 0.3 to 3 would reduce the fraction of cirrus clouds sampled to less than half (relative to the original study as shown in Fig. 1b), restricting this study to the properties of only relatively thick cirrus clouds.  This would make the findings of this study much less meaningful since we find cirrus cloud properties change with extinction which is related to OD.  Considerable time and effort were devoted in Part 1 to extending the sampling range down to an OD of 0.01 over oceans to make the hom fractions in Part 2 representative of all cirrus clouds over oceans.

2. Colormap use:
The manuscript frequently uses a broken colormap, which is (in my opinion) visually appealing but requires justification. If the switch between cold and warm colors aligns with a physically meaningful threshold, this should be explicitly stated. Otherwise, a perceptually uniform colormap should be used to ensure clarity and accessibility.

All figures now use a perceptually uniform colormap whenever relevant (i.e., shades of a single color or gradations between two colors) to ensure clarity and accessibility.

3. Seasonality figures:
The current plots make it challenging for readers to discern seasonality in cirrus cloud properties. Key figures could represent seasonality more intuitively, for example, by showing relative anomalies from annual means.

Seasonal differences are now easier to discern by using maps for only DJF and JJA and headings have been improved for more clarity.  Maps for SON and MAM are now in the Supplement.  This preserves the magnitude of a variable (important findings we believe) while also showing seasonal changes.

4. Key science issue: Homogeneous freezing of solution droplets vs. homogeneous freezing of cloud droplets at homogeneous freezing temperature of water:
The manuscript treats homogeneous ice nucleation in situ and homogeneous freezing of cloud droplets as equivalent, which they are not.

If homogeneous freezing of cloud droplets was important, we would see evidence of this in Fig. 18 (new Fig. 12) as a large peak of $N_i$ in the 231 – 235 K temperature bin.  In general, outside the tropics during non-summer months, updrafts in cirrus clouds are not strong enough to move this cloud droplet freezing zone to much lower temperatures.  Perhaps this study advances our knowledge of cloud physics in this way.  Text has been added to Sect. 3.4.1 to indicate that our results do not provide evidence that homogeneous freezing of cloud droplets is an important process:

…In this way, Fig. *12* shows how the $T_r$ dependence of the hom fraction affects median $N_i$.
   *Hom may occur through (1) the freezing of haze solution droplets (Koop et al., 2000) and (2) the freezing of supercooled cloud droplets advected across the isotherm ~ 235 K (e.g., Rosenfeld and Woodley, 2000).  If (1) and (2) are comparable in their frequency of occurrence, an abrupt increase in median $N_i$ should be evident in Fig. 12 in the 231 – 235 K range (given typical non-convective cirrus updrafts of 10 to 30 cm s$^{-1}$).  Since such an abrupt increase is not evident in Fig. 12, it appears that (2) does not contribute significantly to $N_i$, even in the tropics.  This is consistent with Avery et al. (2020), Costa et al. (2017), and Mitchell and d'Entremont (2012) where it was shown that liquid water is rare in clouds over the range 239 – 235 K.  This applies to both in situ cirrus and WBC clouds and is consistent*

*with the definition of LOC in Luebke et al. (2016) where it is stated that LOC are restricted to pure ice clouds having T < 250 K (-23° C).*

*Figure 13 is similar to Fig. 12 except ...*

This distinction is critical due to its implications for cirrus cloud thinning. While in situ nucleated homogeneous clouds are promising targets for seeding, current thinning methods cannot modify clouds forming at water's freezing temperature. Although alternative modification strategies (e.g., convective invigoration or, possibly, the opposite, weakening; e.g. Varble et al., 2023, https://doi.org/10.5194/acp-23-13791-2023) could be developed, they fall outside the scope of this study.

As noted above, to a first approximation, homogeneous freezing within liquid origin cirrus clouds (LOC) can be viewed as proceeding through the freezing of solution droplets and not cloud droplets. Since the way we define LOC differs slightly from the literature (e.g., Luebke et al., 2016; Dekoutsidis et al., 2023), we now describe our LOC category as warm-base cirrus or WBC. As shown in Fig. 9 of Avery et al. (2020, AMT) and in Fig. 8 of Mitchell and d'Entremont (2012, AMT), clouds are generally glaciated for T < -34° C, except those having strong convection (Rosenfeld and Woodley, 2000, Nature). Moreover, LOC as defined in Luebke et al. (2016, ACP) are restricted to pure ice clouds at T < 250 K (thus excluding mixed-phase clouds, although LOC cloud properties may be influenced by "upstream" mixed-phase microphysics where T > 250 K). Thus, WBC should generally exhibit the solution droplet homogeneous freezing that characterizes in situ cirrus and LOC clouds. This is consistent with our finding that the fraction of hom cirrus is comparable for in situ cirrus and WBC clouds. Moreover, results pertaining to both in situ cirrus and WBC clouds appear to be relevant to cirrus cloud thinning or CCT. The text below has been added to Sect. 3.4.2 to indicate this:

For the hom fraction outside the tropics, the fractions of in situ and liquid origin cirrus are often comparable, although liquid origin dominates south of 60°S latitude during winter. *Although hom in WBC clouds (and thus liquid origin cirrus) has been predicted to occur mostly through the freezing of cloud droplets (Gasparini et al., 2018), evidence for this was not found in Fig. 12 (as discussed in Sect. 3.4.1). Thus, it appears that hom proceeds through the freezing of solution haze droplets for both hom cirrus categories in Fig. 16, making both in situ hom cirrus and hom WBC clouds susceptible to modification by increasing the concentration of INPs, which is the physical basis of CCT (discussed in Sect. 5).*

5. Additional key comment - data accessibility:
To enhance the utility of the dataset for the community, the authors could consider publishing their post-processed data in a user-friendly format, such as NetCDF following CF conventions. This would facilitate its use by climate modelers.

The CALIPSO project plans to include all retrievals described in Part 1 and used in Part 2 in the upcoming version of the IIR Level 2 products, which will be publicly released by the end of this year. The post-processed data used to create the figures shown in this paper are currently stored in ascii format and are available upon request (which was added to the "Data Availability" section).

**Author responses to the specific comments from Reviewer 1:**

I'm sorry if some comments repeat points already made in the key comments section.

Line 117 and Appendix A:

Why should such cases be rare? This is hardly justified. It certainly cannot be the case for cirrus originating from deep convection, where anvils spread far from the deep convective core, see Gasparini et al. 2018, Fig. 5 (10.1175/JCLI-D-16-0608.1). In any case, it's hard to discriminate the origin with only snapshot data, without any means of tracking the cloud evolution.
There are more studies that discriminate between liquid and in situ origin cirrus that could be mentioned, e.g. Wernli et al. 2016 (10.1002/2016GL068922), which uses a large statistical sample of clouds (relying on the imperfect reanalysis data, but at least with good statistics).

We agree that evidence is lacking to justify this sentence (that misclassified in situ cirrus probably make a minor contribution to our in situ cirrus category).   Following Gasparini et al. (2018), we retain these two cirrus categories but redefine the LOC category as warm-base cirrus clouds or WBC clouds to acknowledge the difference between LOC and WBC. This paragraph in Sect. 2.1 has been rewritten, describing these two categories (in situ and WBC) accordingly:
In this work, we take advantage of the improved ice/water phase assignment in the Version 4 CALIOP products to also include cirrus clouds with $T_{base}$ warmer than 235 K (and $T_r$ colder than 235 K), hereafter called *warm base cirrus clouds or WBC clouds*. *Even though these WBC clouds are identified as high confidence ice cloud layers by CALIOP, this assessment*

*does not rule out the possibility of liquid droplets in the lower part of the layer.  This classification method is an attempt to qualitatively contrast the properties of in situ and liquid origin cirrus (LOC) clouds, using WBC as a proxy for LOC clouds.  A similar approach was used in Gasparini et al. (2018).  This approximation may underestimate LOC clouds (overestimating in situ cirrus) since cloud condensate from below the 235 K isotherm may be advected across this isotherm upwind of the CALIOP nadir view when there is no cloud at nadir below this isotherm. In this case the cloud would be mistakenly classified as in situ cirrus. On the other hand, the modeling study by Wernli et al. (2016) estimates that roughly 50 % of in situ cirrus clouds occur on top of LOC, indicating a strong dynamical linkage. Relative to an air parcel back trajectory analysis as used in Wernli et al. (2016) and other LOC studies cited below, our approach should underestimate in situ cirrus if there is no clear layer separating in situ from WBC clouds.  This classification scheme is evaluated in Appendix A in Figs. A2 and A3.  Figure A2 shows the dependence of the in situ fraction on temperature, where this fraction is ~ 0.5 (indicating a transition from in situ to WBC) at about 227 K over oceans when all clouds are considered.  In the LOC studies by Dekoutsidis et al. (2023) and Luebke et al. (2016), which are both based on the same field campaign, this transition occurs around 221 K and between 218 – 222 K, respectively.  This suggests that the WBC approximation overestimates the in situ fraction somewhat (shifting the transition temperature by ~ 6 K relative to these measurement-based studies), but that WBC may still serve as a qualitative proxy for LOC.*

Line 175, but related to many of the figures:

Wouldn't it be enough to show only DJF and JJA in the main manuscript and the other seasons in the supplement?

Yes, we agree and that has now been done.

Line 177: "Since this sampling criteria appears to render radiative properties representative of all cirrus clouds, the cloud property values predicted in GCMs for cirrus clouds should be similar to those shown in these figures."

Should be predicted by the GCM, but only in the sampled range of cloud optical depths. The clouds at COD<0.3 are very common, and while not as radiatively important, will contribute significantly to the mean ice cloud properties.

This sentence has been changed to: "*From Sect. 2.2, this sampling criteria appears to provide cloud property values of cirrus clouds whose radiative properties are representative of all cirrus clouds*".

Line 188: "The correspondence between $T_r$ (Fig. 5) and IWC (Fig. 4) is clearly seen. This is an expected result predicted by the Clausius Clapeyron equation."

But also the agreement with ICNC
Figures 2-7:

1. If one does not pay attention and focus on some features, all subpanels generally look the same.
2. In general, I like the choice of the colormap. However, it is a discontinuous colormap with a very sharp transition. This can only be used if there's a reason for such a choice. Otherwise, the reader will see patterns that aren't real, but just a result of the sharp colormap discontinuity.

The color bars were changed to show gradations between one or two colors without discontinuities or artificial sharp transitions. For seasons, only DJF and JJA are shown.

Section 3.1 is, in my opinion, not important to the main story of the paper. It could be moved to the Appendix, along with the corresponding figures (or at least parts of some of the figures).

As mentioned in our reply under General Comments, Figures 8 – 11 have been condensed into a single figure (Fig. 6) that illustrates the impact of optical depth on $N_i$, IWC, & $D_e$ for midlatitude winter only. The main point is to show that hom mostly affects $N_i$ and IWC, with some impact on $D_e$. Since this finding provides the rationale for Sect. 3.2 (which uses extinction, proportional to IWC/$D_e$, to separate het and hom cirrus; a central result of this study), Section 3.1 still remains, although the number of figures has been greatly reduced.

Section 3.2/Figure 12:
If the relevant threshold is 30 ICNC/liter, then the discontinuity in the color map should be set at that level.

There is no threshold mentioned at 30 ICNC ($L^{-1}$); the text is merely describing $N_i$ regions strongly dominated by het and hom nucleation, and how a $N_i$ gradient exists between these two regions. The colormap for Fig. 12 (now Fig. 7) now follows the new convention shown in

Fig. 2 for $N_i$ (showing gradations between two colors without discontinuities or artificial sharp transitions).

Section 3.2:
The use of extinction coefficient is not motivated

The purpose of Sect. 3.1 was to explain why the extinction coefficient was used to separate predominantly het formed cirrus from predominately hom formed cirrus (i.e., het and hom cirrus). Sect. 3.1 has been modified to make this point more clearly. The opening sentence for Sect. 3.1 is now this:

*The main purpose of this section is to justify the use of the extinction coefficient $a_{ext}$ as a means of separating cirrus clouds formed primarily through het from those formed primarily through hom.*

And the last sentence of Sect. 3.1 now reads:
*This suggests that the ratio IWC/$D_e$ and therefore the extinction coefficient ($a_{ext}$ = 3 IWC/($\rho_i$ $D_e$, where $\rho_i$ = bulk density of ice = 0.917 g cm$^{-3}$) may be sensitive to hom.*

Line 320:
How is this equation derived? What is its source?

Lines 311-212 in the EGUsphere preprint state that "two formulations of the Clausius-Clapeyron equation and the supersaturation equation for hom (Lamb and Verlinde, 2011) were used", and lines 318-319 state: "Finally, the supersaturation threshold where hom occurs, $S_i^f$, is predicted by", followed by the equation in question. This equation for $S_i^f$ comes from the cloud physics textbook by Lamb and Verlinde (2011) and is easily found in their chapter on nucleation (Eq. 7.33). To make things clearer, lines 311-312 are modified to read:

*To explore this further, two formulations of the Clausius-Clapeyron equation and the supersaturation threshold equation for hom, first derived in Lamb and Verlinde (2011), were used.*

Line 347:
Does "theoretical method" refer to the formula in Verlinde? I'm not sure where it comes from, but it doesn't seem like a theoretical relationship, more like an empirical fit to data?

Yes, "theoretical method" refers to Eq. 7.33 in Lamb and Verlinde (2011). The lead author emailed Dr. Lamb, who confirmed that he built upon the findings of Koop (2000, Nature) to present those findings in a simpler theoretical context that is more convenient to use. Note that Koop et al. (2000) "present a thermodynamic theory for homogeneous ice nucleation" and that Lamb and Verlinde are using the water activity offset of 0.305 found in Fig. 1 of Koop et al. (2000) that is based on laboratory data. Koop et al. (2000) does not give the $S_i^f$ formula found in Lamb and Verlinde (2011, Eq. 7.33) which includes this water activity offset. Therefore, this equation is an original contribution from Lamb and Verlinde (2011). Line 347 has been modified to read:

There is close agreement between the theoretical *(i.e., Eq. 3)* and experimental methods*, where both methods address the homogeneous freezing of solution haze droplets based on the activity of water in the solution droplet (Koop et al., 2000) and do not address the homogeneous freezing of activated cloud droplets*.

Lines 348-350:
The authors here assume that homogeneous freezing of cloud droplets is the dominant source of ice in convective plumes. It's hard to know how important it really is, but given that most convective clouds freeze at T of about -30°C, freezing of cloud droplets in the mixed-phase regime should be very important. There are quite a few references on this, although it's harder to find good information on deep convective clouds (e.g. Coopmann et al., 2020 https://agupubs.onlinelibrary.wiley.com/doi/10.1029/2019JD032146; but at least in the tropics, most mixed-phase clouds are of convective origin).
Moreover, while associated with a large uncertainty, secondary ice production is thought to contribute substantially to ICNC in deep convective clouds, see e.g. Hu et al., 2021 (https://doi.org/10.1175/JAS-D-21-0015.1)

Lines 347-350 state: In the tropics the margin between the triangles and the "no data" region is wider. This may be due to the deep convective origin of most tropical cirrus, with convective plumes overshooting the temperature level where hom is first activated, carrying ice to lower temperatures.

There is no mention of homogeneous freezing of cloud droplets here, which occurs when activated, spontaneously growing cloud droplets are advected across the 235 K isotherm. Rather, the process addressed here and in Koop et al. (2000) is the homogeneous freezing of solution haze droplets based on the activity of water in the solution droplet (henceforth

hom).  This is what the Lamb-Verlinde and Schneider et al. relationships describe.  New text has been added, associated with line 347 above (in italics), to clarify this issue.

We agree that ice multiplication processes in convective clouds, such as those described in Lawson et al. (2015, JAS; 2017, JAS), can produce Ni levels similar to hom conditions, but this generally occurs when T < -30° C (e.g., Hu et al., 2021) as the reviewer indicates.  It seems very speculative to invoke ice multiplication and long-range vertical transport to explain CALIPSO samples above the theoretical/experimental hom activation regime (i.e., the triangles and squares) in Figs. 13 and 14 of the preprint (Figs. 7 and 8 in the revised manuscript) when hom is producing relatively high IWC and Ni in closer $T_r$ proximity to these CALIPSO samples.

Figure 14:
Not sure we need panels g-l, since they show the same as panels a-f.

Agreed.  We have removed panels g-l from Fig. 14.  Fig, 12 and Fig. 14 are now combined into the new Fig. 7.

Line 400: "Note that this analysis is based on clouds having $\tau$ < ~ 3 and thus does not consider deep convective columns where $\tau$ > 3 and hom is probably quite active."

We cannot know how active hom is, and in any case it would be homogeneous freezing of cloud droplets.

As indicated above, the process described is homogeneous freezing of solution haze droplets, not cloud droplets, as indicated in the additional new text associated with line 347.  Nonetheless, line 400 has been clarified to read:

Note that this analysis is based on cirrus clouds having $\tau$ < ~ 3 and thus does not consider *thick cirrus clouds originating from deep convection* where $\tau$ > 3 and hom is probably *active in the strong updrafts*.

Figures 15 and 16:
Is there a physical meaning to the colormap boundary at 211 K?

There was no physical meaning to the colormap boundary at 211 K. The colormap was modified to have the darkest colors corresponding to the coldest and warmest bins and lighter colors around the transition from cold to warm colors to attenuate the color

boundary contrast.  In addition, because there is often more coherency in the $N_i$ – and $D_e$ – extinction relationships for T > 215 K, which are referred to as "warm-colored" data in the manuscript, we modified to colormap to have only warm colors for T > 215 K.  The color change makes this easier to recognize, but we do not advocate mechanistic reasons for this coherency change.

Lines 440-441:
What if updrafts are the key difference between land and ocean?

Good question.  Higher updrafts are associated with higher IWC (Hu et al., 2021; Mitchell, 1988, JAS) and higher extinction in Figs. 15 & 16 (now Figs. 9 and 10) is associated with higher IWCs, noting that $α_{ext}$ = 3 IWC/($ρ_i$ $D_e$).  Thus, updrafts should tend to increase with extinction in Figs. 15 & 16, especially when $D_e$ is increasing with extinction.  In this way, updrafts are implicit here with relatively weak updrafts associated with the apparent Twomey effect.  New text has been added:

*While higher updrafts over land could also enhance INP concentrations, note that updraft effects are implicit in Figs. 9, S11, and 10. That is, higher updrafts are associated with higher IWC (Hu et al., 2021; Mitchell, 1988) and higher extinction is associated with higher IWCs. The apparent Twomey effect here is associated with $α_{ext}$ < 0.3 km$^{-1}$ where updrafts are expected to be relatively weak over both ocean and land.*

Figure 17:

I see no difference between the dashed and solid lines, they seem to be parallel. Therefore, one of them could be removed from the plot for clarity.

Having a step increment of 0.1 (rather than 0.2 as suggested by the reviewer) provides more precise knowledge, and for some this may be important, so we have retained this information.

Figure 20:
Why is there a seasonality in the parameterization for the Northern Hemisphere (discontinuously moving from 30° to 60°N) but not for the Southern Hemisphere?

There is a seasonality for both hemispheres (i.e., a discontinuity moving from 30° to 60° latitude) as described in the text (lines 522-525 and lines 550-564 of preprint).  This appears to be due to changes in mineral dust concentration as discussed in this section.

Why is the word "corrected" in the title?

This is to refer to the correction factor required over land (see Eq. 6). We have removed it.

Figure 21:
Why do we need 4 temperature bins when the behavior seems to be pretty much the same in each of the bins (also, we are at figure 21 already, and it's starting to get harder to keep the focus).

We used four temperature bins in Fig. 21 (now Fig. 15) to show that we see the same behavior in each instance. We now show only one temperature bin for illustration (at 229 K) and say that the same behavior is seen at the other temperatures as described in the new text below:

Because this was unexpected, we examine in Fig. *15* the dependence of $N_i$, $D_e$, and IWC on extinction for the four seasons *for the temperature bin at 229 K. Other temperature intervals (having a mid-temperature of 233, 225, and 221 K) exhibited the same behavior.*

Figure 22 is, in my opinion, the key figure of the manuscript, but unfortunately it gets lost a bit due to the large amount of information presented.
Another comment, similar to the one above: I don't think it's fair to lump together in situ cirrus and cirrus of liquid origin. Only in situ cirrus can be modified. So the readers might want to have numbers of hom vs. het for in situ cirrus only.

The reviewer brings up a common misconception among cloud physicists that this study addresses. The theoretical and AIDA chamber results presented in Figs. 7 and 8 (revised paper; triangles and squares) are based on the homogeneous freezing of solution haze droplets and not cloud droplets, and since they coincide with the highest retrieved $N_i$ and IWC (for a given temperature level), this suggests that hom in natural cirrus clouds (both in situ and WB cirrus) is predominantly from the homogeneous freezing of solution haze droplets. As described above, we now discuss that mixed phase clouds are predominantly all-ice for T < -34° C (< 239 K), implying that even near 235 K (where cloud droplets would freeze if they existed) homogeneous freezing proceeds primarily through the freezing of solution haze droplets. In the revised paper we have now discussed that, if cloud droplet freezing was important, an abrupt increase in median $N_i$ should be evident in Fig. 12 (revised paper) near 235 K, but this is not evident. Using WB cirrus as a proxy for liquid origin cirrus, it is fair to include both in situ cirrus and WB cirrus in Fig. 22 (Fig. 16 in revised

version) since hom primarily proceeds through the same mechanism in each cloud type. Moreover, both in situ cirrus and WB cirrus can be modified via CCT for this same reason. New text has been added to indicate this:

*Although hom in WBC clouds (and thus liquid origin cirrus) has been predicted to occur mostly through the freezing of cloud droplets (Gasparini et al., 2018), evidence for this was not found in Fig. 12 (as discussed in Sect. 3.4.1). Thus, it appears that hom proceeds through the freezing of solution haze droplets for both hom cirrus categories in Fig. 16, making both in situ hom cirrus and hom WBC clouds susceptible to modification by increasing the concentration of INPs, which is the physical basis of CCT (discussed in Sect. 5).*

Section 4:
This is a valuable section, but there's a lot of speculation. I am very intrigued by the results presented in Figure 24. These should be verified in the future with other observational datasets and model studies to confirm or reject the proposed interpretation of cirrus cloud properties relative to updraft.
To increase the relevance of this section, the authors could compare their hypotheses with parcel model studies of cirrus, e.g. work by Bernd Kärcher, if possible.

We looked at some of the recent papers by Kärcher et al. but did not find any compelling reason to cite this work (i.e., it did not add significant value to the paper in our view).

Figure 25:
I assume that the INP number for Figure 25 and its description should be fixed. This should be stated explicitly.
Does the model in Figure 25 hold for all cirrus or only for in situ cirrus?
A caveat to this interpretation is that cirrus data could also be explained as cirrus at different stages of cloud evolution.

As stated previously, our results (and thus the model in Fig. 25 of the preprint) pertain to cirrus formed by the freezing of solution haze droplets for both in situ cirrus and WBC clouds. This figure (new Fig. 18) is now shown at mid-latitude only by separating in situ cirrus and WBC clouds, showing that the correspondence between Tr-$T_{top}$ and maximum $D_e$ is more evident in WBC than in situ cirrus clouds which are geometrically thinner.

Lines 645-6 (in the preprint) have been modified to read:
*This proposed explanation of Figs. 17 and 18 is summarized in the schematic in Fig. 19 for the warmest WBC layer, presenting a conceptual model of how cirrus cloud thickness might evolve with increasing cloud updraft and IWC for a fixed INP number concentration.*

*Section 5:*
*It seems a bit odd to have a separate section mentioning the otherwise very relevant study by Froyd et al., 2022.*

Journal articles often have a section titled something like "Comparison with other studies", and the ACP abstract format specifically asks authors to describe how their work differs from previous work on the topic addressed. Nevertheless, we understand that having this short section dedicated to only one study is odd. The comparison with the work by Froyd et al. (2022) has been shortened and included in Sect. 3.4.1 and Sect. 5 of the pre-print has been removed.

I don't think the Froyd et al., 2022 study presented any ice residual measurements (unlike e.g. Cziczo et al., 2013).

Yes, that is correct. This sentence was removed from the shortened text now in Sect. 3.4.1.

Figures 26 and 27:
Figures 26 and 27 are very useful for the purpose of scientific presentations, but lack some more content to qualify for a proper scientific publication. For example, can we be sure that the clouds shown are at temperatures below the homogeneous freezing temperature of water?
In the best case, one could find a photo of a cirrus cloud with a coincident CALIPSO overpass and the analysis as done for the rest of the paper.

Figures 26 and 27 in the preprint exhibit photos of het and hom cirrus clouds. Since most people are familiar with common het cirrus clouds, Figure 26 has been removed but Fig. 27 (new Fig. 20) showing an example of hom cirrus clouds has been retained. Many scientists consider lenticular or wave cirrus clouds to be very limited in areal coverage (and hence not radiatively important; see Krämer et al., 2016, ACP), but observations by the lead

author who lives immediately downwind of the Sierra Nevada Mountain range have consistently shown that this is true only for the relatively low wave cirrus. At higher levels, these wave cirrus exhibit extensive areal coverage as shown in Fig. 27 of the preprint and are thus relevant to radiation (which is why Fig. 27 is retained). These wave cirrus clouds induced by orographic gravity waves (OGWs) have relatively high updrafts and thus are more likely to be hom cirrus clouds (Joos et al., 2008, JGR; Joos et al., 2014, ACP; Barahona et al., 2017, Nature; M2024). They also tend to exhibit higher cloud fractions (see Fig. 4 in Matus and L'Ecuyer, 2017, JGR). These OGW cirrus are characterized by relatively high Ni and, perhaps due to the oscillation of the OGW, are evident far downwind from mountain ranges in North America, Patagonia, and Antarctica during winter (Fig. 2 and S5). The paragraph describing Fig. 27 in the preprint (new Fig. 20) has been revised and moved to the new Sect. 6 (Sect. 7 of the pre-print) after the discussion regarding the importance of including OGWs for ice nucleation in the models:

*An example of what OGW cirrus clouds often look like is given in Fig. 20, which are optically thicker than het cirrus clouds. Cirrus clouds induced by OGWs, often called wave cirrus, have relatively high updrafts and thus are more likely to be hom cirrus clouds (Barahona and Nenes, 2008; Joos et al., 2008; Joos et al., 2014; Barahona et al., 2017;* Lyu et al., 2023; *M2018). They also tend to exhibit higher cloud fractions (see Fig. 4 in Matus and L'Ecuyer, 2017). These OGW cirrus are characterized by relatively high $N_i$ (M2018, Gryspeerdt et al., 2018) and, perhaps due to the oscillation of the OGW, are evident far downwind from mountain ranges in North America, Patagonia, and Antarctica during winter (Figs. 2 and S5). Note that hom cirrus are not restricted to OGW cirrus and may form under other conditions having relatively high updrafts and/or low INP concentrations.*

Lines 730-731:
Or, more physically, increase the vertical resolution of the model.
Or use some kind of subgrid cloud fraction in the vertical (similar to what is done in the horizontal dimension in coarse GCMs).
In any case, the pre-existing ice formulation tends to be less important when using high resolution models, and may become less important in future cirrus modeling studies.

We agree with these comments and this sentence has been modified as follows:

*To address this issue, the model's vertical resolution could be increased, or $q_{i,pre}$ could be attenuated by a factor that best represents $q_i$ near cloud top in the "nucleation zone".*

Lines 754-755:

The study by Froyd et al. 2022 used only an idealized model setup (based on dust measurements, but the trajectories were based on reanalysis data).

This sentence has been modified as follows:

*From a global observational purview, this was done for the first time by Froyd et al. (2022) who used global measurements of dust concentration from aircraft in the UT to initialize a detailed cirrus cloud formation model that used reanalysis data in the dust trajectory simulations.*

---

## Author Comment (AC2)

**Author responses to Reviewer 2**

Format: The reviewers' comments are in black font while author responses are in red font.  Text in red font italics indicates revised/added text in the revised manuscript.

We understand that reviewing this paper took a lot of time and effort, and we sincerely thank you for your comments that have improved this paper.  Below are our responses to the general and specific comments:

**General comments from Reviewer 2:**

Advances in CALIPSO (IIR) cirrus cloud property retrievals – Part2:
Global estimates of the fraction of cirrus clouds
affected by homogeneous ice nucleation

David L. Mitchell and Anne Garnier

**General:**

The manuscript contains a new global climatology of cirrus clouds derived from satellite observations. It is an impressive work with very extensive analyses of various properties of cirrus, as cloud ice particle number concentration ($N_i$), effective diameter ($D_e$), ice water content (IWC), shortwave extinction coefficient ($\alpha_{ext}$), optical depth ($\tau$), and cloud radiative temperature. The study includes innovative data analyses that lead to new perspectives and a deeper understanding of cirrus clouds. In particular, the observations are analyzed to determine whether the cirrus formed homogeneously or heterogeneously. Further, the fraction of hom-affected cirrus clouds is determined and $\tau$ distributions are used to establish a proxy for cloud net radiative effect (CRE) of the hom affected cirrus. Finally, a conceptual model of cirrus cloud characterization is proposed. Altogether, this study has the potential to become a new standard work on cirrus properties.

Unfortunately, however, I have some concerns, which I will list in the following. I know that the authors are experienced scientists with many publications and therefore write their articles the way they like it best. Nevertheless, I would like to add some comments, because I feel that otherwise the extensive and thorough study may not get the attention it deserves.

**(G 1)** It took me quite a while to work through the long, sometimes complicated text and the equally complicated figures. To my opinion, the interesting, but complex results could be presented more simply and shorter to make them easier for the reader to understand. Otherwise, I fear readers will be discouraged from reading the article.

So, overall, I think it might be good to consider shortening the main part of the paper and only showing the most important figures in that part. Everything else could be moved to the Appendix or Supplementary Material.

Reviewer 1 made a similar recommendation, along with specific instructions. We have followed the advice from Reviewer 1 and Reviewer 2 to simplify this paper, reducing the number of figures in the main part of the paper from 27 to 20, greatly reducing the number of panels is some of the figures, and changing the labeling convention in some figures.

**(G 2)** I have some suggestions for simplifications - but not for the text, for that I can only ask the authors to go through the manuscript again and simplify and shorten the descriptions.

For example, I recommend

**(a)** to introduce a small table with abbreviations, which contains for example:

$T_r$: cloud layer radiative temperature, approximately in the middle between $T_{top}$ and $T_{bottom}$

IAB: CALIOP 532 nm layer integrated attenuated backscatter

IAB < 0.01 sr -1 = ~ 0.01 < $\tau$ < ~ 0.3 optically thinner cirrus – thin - subvisible cirrus*

IAB > 0.01 sr -1 = ~ 0.3 < $\tau$ < ~ 3 optically thicker cirrus – opaque cirrus

* subvisible cirrus ($\tau$ < 0.03), thin cirrus (0.03 < $\tau$ < 0.3)

**(b)** to use either the $\tau$ ranges throughout the manuscript to identify the cirrus type, or (I think even better) the name (subvisible cirrus, thin cirrus, opaque cirrus), IAB doesn't give an intuitive impression on the type.

We appreciate the need to provide a more intuitive impression for cloud type, and a new table (Table 1) has been added in Sect. 2.1 which contains the information mentioned above by Reviewer 2.  These descriptions of cirrus clouds are then used throughout the article (i.e., thin, thick, and all cirrus clouds).

**(c)** to simplify the figures:

I strongly recommend revising all figures so that the recurring headings above each panel be incorporated into a general figure title, so that only the specific information appear above the panels (in the current version it is hard to find out the differences between the panels).  Further, I would also include information that is now somehow hidden in figure captions in the Figure title.

As an example of the simplification of the figures here **Figure 18 (**I modified the figure for my own understanding):  <not shown>

This has now been done for all of the applicable figures.

**(G 3)** Retrieval of liquid origin cirrus

Line 110 ff: ··· *cirrus clouds with $T_{base}$ warmer than 235 K (and $T_r$ colder than 235K), hereafter called liquid origin cirrus. ···*
*This method is an approximation that may underestimate liquid origin cirrus clouds*

*somewhat (overestimating in situ cirrus) since cloud condensate from below the 235 K isotherm may be advected across this isotherm upwind of the CALIOP nadir view when there is no cloud at nadir below this isotherm.'*

This method sorts not only liquid origin as in-situ origin cirrus, but likely also in-situ origin as liquid origin, as explained in the following: Warm conveyer belts (but also convective systems) consist from bottom to top of layers of liquid, mixed-phase and cirrus clouds. The mixed-phase clouds appear in the cirrus region as liquid origin clouds, but above these, in-situ cirrus usually also form due to the lifting of the air masses. An example is shown by Luebke et al. (2016) (see Figure below, top panel). The vertical structure of liquid origin and in-situ origin cirrus is clearly recognizable.

If these clouds were classified as described in this paper (i.e. if the clouds reach down to temperatures warmer than -38C they are liquid origin cirrus), the whole in-situ origin cirrus umbrella would be misclassified as liquid origin cirrus.

I would recommend doing some case studies to test the classification. A trajectory analysis, as done for example by Luebke et al. (2016), would be best suited for this. This is the most reliable method to classify cirrus of in-situ and liquid origin. I think it is crucial to check the classification method, as all results on in-situ origin and liquid origin cirrus depend on the correctness of the sorting - and my concern is that many of the in-situ cirrus at the top of WCBs, MCS or convective cells will be classified as liquid origin.

We have changed our cirrus cloud classification wording to describe only what the CALIOP lidar measures, with clouds having a radiative temperature $T_r \leq 235$ K and a cloud base temperature $T_{base} > 235$ K classified as "warm-base cirrus clouds", or WBC clouds.  The "liquid origin cirrus" classification is no longer used.  We provide evidence showing that WBC and liquid origin cirrus are not equivalent but also provide evidence showing that WBC may be used as an approximation or proxy for liquid origin cirrus as done in Gasparini et al. (2018, J. Climate).

If there is an in situ origin cirrus umbrella with a clear layer between this "umbrella" and a WBC cloud below, then the lidar will likely detect two layers and the scene will not be included in our sampling because only single layer clouds are sampled. But if an in situ cirrus cloud and a WBC cloud somehow become connected and "bridge" so that the lidar detects only one layer, then this method will classify this vertically continuous cloud layer as a WBC cloud. Our methodology is not capable of determining such bridging phenomenon.

**(G 4)** Hom-affected and het-only cirrus clouds (Sections 3.2 and 3.3 etc.)

E.g. Line 293ff: *Characteristic in all plots* (Figure 12 and subsequent figures) *is a broad region on the left side (relatively low $\alpha_{ext}$) where $N_i < 30 \ L^{-1}$, apparently corresponding to het only. To the right of this region is a gradient of increasing $N_i$, culminating in values of $N_i > 2000 \ L^{-1}$. This gradient region is likely produced by varying degrees of hom activity.*

I am not convinced by this classification and would interpret this central point of the paper differently, as I will explain in the following.

I agree that there are two cirrus regimes, as described in Kramer et al. (2016, 2020). Here $\alpha_{ext}$ (color coded by $N_i$) is used to make this visible. The region of high $\alpha_{ext}$ and $N_i$ ('hom acitivity') corresponds approximately to the area of high IWC and Ni in Fig. 6 (top panel) of Kramer et al. (2020). In this region, both in-situ and liquid-origin cirrus clouds are present. For the in-situ cirrus clouds, the interpretation that they are formed by hom (of soluble aerosol particles) is correct, but not for the liquid-origin cirrus. Hom (of cloud drops) can occur within liquid-origin cirrus clouds, but is rather rare outside the tropics. **The predominant freezing mechanism of liquid origin cirrus is het**, nevertheless, they can have high $N_i$.

Dekoutsidis et al. (2023) show that RHi conducive to hom (RHi > 140%) occurs near cloud top irrespective of whether the cirrus clouds are in situ or liquid origin. This is consistent with our findings that show hom contributes to both in situ and liquid origin cirrus (LOC) clouds.

The region with low $\alpha_{ext}$ and $N_i$ is defined here as 'het only'. However, I think the composition of the cirrus clouds in this region are much more complex. First to mention, in this region there are also both in-situ and liquid origin cirrus present. Further, the in-situ origin cirrus could have formed either hom or het, since hom also produces only few ice crystals at warm temperatures and low updrafts.

Agreed.  We acknowledge this in text added to Sect. 3.2 that is given next in italics.

But, most importantly, the concentration of hom cirrus with initially high $N_i$ (and thus $\alpha_{ex}$) decreases quite rapidly in the warming phases of the ubiquitous mesoscale temperature fluctuations where the environment is subsaturated (Jensen et al., 2024). This means that they are moving from the hom affected regime to what is now defined as het only. This can be seen also in Fig. 6 (bottom panel) of Kramer et al. (2020) - the thinner the cirrus (and the lower $N_i$) the more frequent the cirrus clouds are in a subsaturated environment.

   The fact that the **cirrus clouds with low $\alpha_{ext}$ and $N_i$ are in a subsaturated environment** is also indicated by the decreasing $D_e$ to the left of $D_{max}$ (Figures 15 and **16), because under this condition, the thinner the cirrus clouds and the lower the $N_i$,** the smaller the ice particles.

   In an at least saturated or supersaturated environment, the ice particles would be larger with decreasing Ni, i.e. there would be no maximum in $D_e$, but an increase, maybe with a change of the slope during the transition from one to the other regime.

   In summary, I believe that this region is a mixture of in-situ origin cirrus clouds of different ages, which could have formed either het or hom, and aged liquid origin in the dissolution stage.

   I recommend reconsiding the naming and the discussion of the 'het only' cirrus regime.

What I wonder (although I know it would be a lot of work) is whether this analysis would be better done separately for in-situ and liquid origin cirrus (the derivations

presented in section 3.2 only apply to in-situ cirrus anyway)? Especially for in-situ cirrus, the interpretation of the freezing mechanisms would be much clearer, now the liquid origin probably blurs their features.

We thank the reviewer for making us aware of the ambiguity of our results concerning Fig. 12 of the preprint (now Fig. 7). This figure has been revised to show these results for both in situ cirrus and WBC clouds during winter only. It is seen that the results are very similar for both cloud types, with a strong $N_i$ gradient when related to extinction. Assuming WBC are mostly LOC, this indicates that hom is also important for LOC.

As indicated in our responses to Reviewer 1, the freezing mechanisms are now discussed and clarified, with hom ice nucleation in both in situ and WBC clouds proceeding primarily through homogeneous freezing of solution haze droplets. We agree that hom under warmer, low updraft conditions can also produce $N_i < 30$ $L^{-1}$, and the relevant section of text has been rewritten as:
"Characteristic in all plots is a broad region on the left side (relatively low $\alpha_{ext}$) where $N_i <$ 30 $L^{-1}$. *Although hom can produce such low concentrations at warmer temperatures and low updrafts (Krämer et al., 2016), hom tends to produce much higher $N_i$ (Barahona and Nenes, 2009).* To the right of this region is a gradient of increasing $N_i$, culminating in values of $N_i > 1000$ $L^{-1}$. This gradient region is likely produced by varying degrees of hom activity, *although het may also contribute to this gradient under conditions of relatively high INP concentration. It is evident that hom-affected cirrus clouds are common in both in situ cirrus and WBC. The main difference between these cloud types is in the tropics where in situ cirrus often appear to the right of the region predicted for pure hom (i.e., the triangles or squares) which will be discussed below. This may be due to deep convection overshooting the temperature level predicted for hom to activate, depositing moisture at lower temperatures where in situ cirrus subsequently form. Results like Fig. 7 are shown in Fig. S7 for land where this 'overshooting effect' is more evident for in situ cirrus outside the tropics, perhaps due to stronger orography-induced updrafts over land."*

In addition, the descriptor "het only" has been removed from the paper, and such clouds are now referred to as het cirrus or het dominated cirrus clouds.

**Specific comments:**

**(S 1) Line 35f:** ⋯ *liquid origin cirrus associated with cloudy air advected from lower levels (T>235 K) that often contains liquid cloud droplets.*

I don't think cirrus of liquid origin often contains liquid droplets - they are usually completely glaciated by the Wegener-Bergeron-Findeisen process when they enter the cirrus temperature regime (< -38C; see Costa et al., 2017, their Figure 15). Only at high vertical velocities (as in convective cirrus), or in the absence of INPs (e.g. in the Arctic), liquid droplets can rise to temperatures as low as -38 C, where they freeze homogeneously.

Completely agree.  Earlier text has been removed and replaced with new text as follows: "*with liquid origin cirrus associated with cloudy air advected from lower levels (T > 235 K) that is typically near ice saturation and completely glaciated by the Wegener-Bergeron-Findeisen process when this air enters the cirrus temperature regime (T ≤ 235 K; Luebke et al., 2016; Costa et al., 2017; Avery et al., 2020; Mitchell and d'Entremont, 2012).  However, at high vertical velocities liquid cloud droplets may be advected into the cirrus regime where they immediately freeze homogeneously (e.g., Rosenfeld and Woodley, 2000).*"

**(S 2) Line 51f**: *However, hom resulting from cloud droplet freezing dominated ice production in the lower part of cirrus clouds at all latitudes.*

This is not consistent with the measurements of Costa et al. (2017), see point (2).

Yes, we agree this is at variance with observations, but we are just reporting what other studies have done, and the observations cited earlier should alert the reader when discrepancies between predicted and observed features occur.  We challenge some of these modeling results in the next paragraph.

**(S 3)** All kinds of studies are cited in the introduction, but not with a specific focus. Then it is said - even without focus - what is in the paper.

A new sentence has been added to the beginning of the 3[rd] paragraph of the Introduction, stating: "*Since this study estimates the fraction of cirrus clouds strongly affected by hom, we briefly review similar estimates from modelling and observational studies here.*"  This provides the rationale for what follows, where

various modelling and observational studies are cited and related to the relative roles of het and hom in cirrus formation.

**(S 4) Line 130-132:** *When only clouds with τ >~0.3 are sampled over oceans* (solid red lines)*, liquid origin cirrus clouds prevail at mid- and high latitude (60% and 70%, respectively), but not in the tropics (32%).*

I think these numbers 60, 70 and 32% can be derived from those in the respective panels by taking the difference to 1, right? That is not easy to understand - please mention it in the text or write both numbers (for in-situ and liquid origin) in the panels.
*When all clouds are considered* (solid blue lines)*, the percentage of in situ cirrus increases by 18 to 25 % and they always prevail.*
I can't find these numbers in the panels ⋯

Regarding both of these two (S4) comments, this paragraph has been rewritten in response to other review comments, and the relevant (S4) text has been removed. However, the end of the second paragraph in Appendix A now reads "*From Fig. A2, the overall percentage of in situ cirrus for all sampled cirrus over oceans ranges from 86% in the tropics, to 62% in the midlatitudes, and 55% in the high latitudes. These percentages subtracted from 100 yield the percentage of WBC clouds.*"

**(S 5) Line 132ff:** *For these blue curves, it is seen that the liquid origin cirrus prevail at Tr larger than about 227 K, which is ~ 6 K higher than shown in Luebke et al., 2016 (their Fig. 13) and Dekoutsidis et al., 2023 (their Fig. 4).* ⋯
Note that the analysis of Luebke et al. (2016) and Dekoutsidis et al. (2023) are based on the same field experiment (ML-Cirrus) and represent only the meteorological conditions that prevailed during that time. I would recommend to compare the in-situ / liquid origin fractions with the analysis of Wernli et al. (2016), which covers 10 years of ERA5 data.

We have modified adjacent text to indicate both studies are based on the same field campaign: "*In LOC studies by Dekoutsidis et al. (2023) and Luebke et al. (2016), which are both based on the same field campaign, this transition occurs around 221 K and between 218 – 222 K, respectively.*" The study by Wernli et al. (2016) is based solely on model-generated ERA-Interim ice water content (IWC), liquid water content (LWC), temperature, and wind fields, and the temperature where in situ and liquid origin ice clouds are about equally frequent (based on their Fig. S4) is -34°C (239 K), which is much warmer than the cloud-type transition temperature found in this study and in the two ML-Cirrus studies. The authors admit that IWCs and LWCs are "produced by rather simplistic cloud physics" and that "The thermodynamic cloud phase in ERA-40 and ERA-Interim is parameterized simply as a function of temperature, with pure ice clouds below -23°C and mixed-phase clouds between -23°C and 0°C." These and other limitations preclude the use of this modeling study in our study for comparing in-situ / LOC (or WBC) fractions and transition temperatures.

**(S 6) Line 175:** *Global maps for each season are shown for median Ni, De, IWC, and Tr using the cloud sampling criteria described in section 2.1 and IAB $\geq$ 0.01 sr-1 (i.e., ~ 0.3 < $\tau$ < ~ 3,* thick cirrus*)* ⋯

Please introduce the cirrus category, thick cirrus', also in the Figure captions (or even more simply as the title of the figures: ,Median Ni / De / IWC / Tr , Tr < 235 K, thick cirrus (~ 0.3 < $\tau$ < ~ 3)', then only the time of year appears above the individual panels.

The sentence indicated above has been modified as: "*Global maps … are shown for median $N_i$, $D_e$, IWC, and cirrus cloud $T_r$, for thick cirrus (i.e., ~ 0.3 < τ < ~ 3) in Figs. 2, 3, 4, and 5, respectively.*"
The figure titles and captions have been modified as per the suggestions here.

**(S 7) Line 193ff (Figure A3):**
**\* panel b:** I wonder why the in-situ $D_e$ is of comparable size to that of liquid origin?

I think it should be smaller, because in-situ ice particles cannot grow as large as ice particles of liquid origin.

This is observed at a given temperature perhaps because hom activity is comparable for in situ and WBC clouds, as discussed above. In general, liquid origin ice particles will be larger than in situ ice particles since the latter form at lower temperatures as shown in Fig. A2.

**\* panel e:** I wonder why the in-situ $N_i$ of thick cirrus (red curves) increases at warm T - could this be a misclassification?

Our results show that hom is more active in thick cirrus and at relatively warm cirrus temperatures for NH midlatitudes over oceans during DJF (corresponding to Fig. 12b and Fig. A3). Panel e in Fig. A3 is consistent with Fig. 12b. But this hom activity varies with season and location, as shown in Fig. 12. New text has been added at the end of Appendix A: "*As shown in Fig. 12, the temperature dependence of the fraction of hom cirrus varies with latitude and season, indicating that results in Fig. A3 may vary somewhat for different latitude zones and seasons.*"

In addition, a misclassification does not appear likely due to the consistency of in situ optically thick cirrus $N_i$ in Fig. A3 panel e with other Fig. A3 panels and other figures. For example, the geometric thickness (Fig. A3d) of these optically thick in-situ cirrus decrease as T increases and approaches 235 K while their optical depth is about constant (see Fig. A3g). The increase of $N_i$ appears consistent with this increase in extinction. Extinction is proportional to $IWC/D_e$, and this in situ $D_e$ is quasi-constant at these warmer temperatures while IWC increases. As shown in Fig. 6 of Sect. 3.1, $N_i$ and IWC tend to track each other in optically thick cirrus and this section also suggests that these cirrus tend to be relatively thin cirrus having relatively high $N_i$.

**\* panel f and h** (IWC and IWP): I wonder why the in-situ IWC and IWP (red curves) is of comparable size to the liquid origin ones? I think they should be smaller. Misclassification?

When all cirrus clouds are considered (blue curves), indeed this is true (i.e., in situ IWC and IWP are lower than corresponding WBC values). But orographic gravity wave (OGW) cirrus may contribute substantially to the optically thick in situ cirrus (red curves), possibly resulting in higher IWCs (for a given Tr) relative to optically thick WBC clouds.  And OGW in situ cirrus may tend to be geometrically thicker than typical in situ cirrus (see Fig. A3d), perhaps explaining (along with higher IWCs) why optically thick in situ cirrus have only slightly lower IWPs than optically thick WBC. The difference might also be attenuated because the clouds of optical depth larger than 3 are not sampled.

**(S 8) Line 237f:** *… two different  $\tau$  categories: ~ 0.01 <  $\tau$  < ~ 0.3 (IAB < 0.01 sr-1) and ~0.3< $\tau$ <~3 (IAB>0.01sr-1); henceforth categories 1 and 2.*
Instead of categories 1 and 2, you could say 'thin cirrus' and ,thick cirrus', which is more specific and informative.

Agreed; text has been changed here as suggested above: "*Figures 6 provides a means of investigating this question, evaluating $N_i$, IWC, and $D_e$ for optically thin (~ 0.01 <$\tau$ < ~ 0.3) and thick (~ 0.3 <$\tau$ < ~ 3) cirrus clouds.*  Only retrievals over ocean are considered since variable land emissivities preclude retrievals over land *for the thin clouds* ….
"*Figure 6a shows that for thick cirrus, the highest $N_i$ (resulting from hom due to its magnitude) is found in relatively geometrically thin clouds (consistent with Fig. 10 in M2018), while Fig. 6d for  optically thin cirrus is almost featureless with $N_i$ < 100 $L^{-1}$ in general.*"

**(S 9) Line 251f:** *⋯ highest* (IWC) *values in* (geometrically) *thinner clouds in Category 2 for a given Tr, were not anticipated.*

Possibly the higher IWC, especially at warmer temperatures in the geometrically thinner cirrus clouds, indicates that these are young cirrus which have not yet lost any ice particles through evaporation in temperature fluctuations (see Jensen et al., 2023). During aging ice particles are lost by evaporation and sedimentation, so the geometrical thickness increases and the IWC decreases.

The above sentence has been replaced by these two sentences: "*Jensen et al. (2024) show that $N_i$ and IWC are higher in younger tropical cirrus clouds due to mesoscale temperature fluctuations from gravity waves that act to decrease them over time. This may help explain these results if hom cirrus are associated with young cirrus.*"

**(S 10) Line 253ff:** *For a given $T_r$, $D_e$ tends to be quasi-constant, although usually decreasing for the thinnest clouds in both categories, possibly due to entrainment. But this $D_e$ decrease could also be due to hom in Category 2* (thick cirrus)*…*
It could also be that the larger ice crystals in geometrically thicker cirrus clouds, especially in the tropics at warm temperatures, indicate liquid origin cirrus clouds. I think that is more likely than the occurrence of hom.

This speculation has been removed from this sentence, which now reads: "*For a given $T_r$, $D_e$ tends to be quasi-constant, although decreasing for the thinnest clouds in both $\tau$ categories.*" Moreover, as shown in Fig. A3, in situ $D_e$ is of comparable size to that of WBC $D_e$.

**(S 11) Line 440ff:** *Most evident when comparing Figs. 15 and 16 for $\alpha_{ext} < 0.3\ km_{-1}$ (where het is expected to prevail) is that median $N_i$ is higher over land (up to a factor of 10), presumably due to higher INP concentrations over land. …*
Or stronger updrafts → enhanced hom over land ?
⋯. higher INP over land (which can also be enhanced by stronger updrafts) may be producing a "Twomey effect" in het cirrus clouds over land.

This is very speculative (over-interpreted?) …. to make this hypothesis more information about INP and updrafts would be necessary.

New text has been added at the end of Sect. 3.3.2 to address this concern: *"While higher updrafts over land could also enhance INP and Ni concentrations, note that updraft effects are implicit in Figs. 9, S11, and 10. That is, higher updrafts are associated with higher IWC (Hu et al., 2021; Mitchell, 1988) and higher extinction is associated with higher IWCs. The apparent Twomey effect here is associated with $\alpha_{ext} < 0.3$ km$^{-1}$ where updrafts are expected to be relatively weak over both ocean and land."*

**(S 12) Line 476f:** *When $D_e >$ sensitivity limit, we set the sample as het-only.*
Here I have strong concerns, as outlined in point **G 4**.

"Het-only" in the above sentence has been changed to "het cirrus". New text has been added immediately below the sentence cited above that may address this concern: *"Applying this $D_e$ sensitivity limit to all cirrus cloud samples over oceans reduces the estimated hom fraction by less than 8 % at $T_r < 211$ K. The largest changes are for $T_r$ within the 223-235 K interval, where the hom fraction is reduced by 50 % in the tropics, and by 22-25 % in the extra-tropics."*

**(S 13) Line 550ff:** *Also of interest are the seasonal changes in hom fraction between 30°N and 60°N in Fig. 20. Relative dust contributions of the world's main dust source regions are ⋯ more likely to reach cirrus cloud levels in the UT due to ascent within frontal systems, orographic uplift, and dry convection. …*
As in point S 11, seasonal changes in hom fraction might also be due to changes in updrafts and not only to be related to INP.

A new paragraph has been added below the paragraph indicated above, stating:
*"It can also be argued that the above seasonal differences in the hom fraction can be attributed to seasonal differences in vertical velocities at cirrus cloud levels. However, this appears less likely when one considers that strong orographic lifting*

*occurs over the southern Andes Mountains during all seasons, and these vertical motions should be much greater than other vertical motions at cirrus levels in this region (excepting deep convection in summer, but the thick anvils affected by such convection are not sampled by this method). If the hom fraction changes are sensitive to changes in updraft strength, the hom fraction over the southern Andes should not change much between DJF and JJA, but it does. A similar argument can be made for the Himalayas and the Rocky Mountains. Finally, two studies (Sporre et al., 2022; Lin et al., 2025) have documented large microphysical changes in cirrus clouds that were impacted by volcanic aerosol, which are consist with this reasoning.*"

**(S 14) Line 575, Figure 22:** I wonder if it wouldn't be easier to understand if LO and IS were the same color (LO grey, IS red) and hom / het the color shades ? And then to plot LO (het/hom) over IS (het/hom).

In this paper, our objective is to estimate the fraction of cirrus clouds strongly affected by hom and conversely those clouds strongly affected by het. This distinction is more readily apparent in our view by assigning one color type to these het cirrus (grey) and another color type (pink/red) to the hom cirrus, where WB and IS are indicated by color intensity. Therefore, we prefer to keep this figure as it is.

**(S 15) Line 604f:** *Interpreting these w regimes as het and hom regimes, respectively (which was not done in Kramer et al., 2016) ···.*
It has been discussed by Kramer et al. (2016) that het dominates in slow updrafts with low IWCs and hom in fast updrafts with high IWCs, see their Figure 6 and corresponding text.

The text here has been modified as follows: "The slow w regime *where het dominates* was characterized by IWC < median IWC, relatively low $N_i$, and relatively large mean mass volume radius $R_v$. Conversely, the fast w regime *where hom dominates* was characterized by IWC > median IWC, relatively high $N_i$, and relatively small $R_v$. Interpreting these w regimes as het and hom regimes, respectively, this

study reports similar findings in Figs. 9 and 10 *(where $\alpha_{ext}$ correlates strongly with IWC)."*

**(S 16) Line 650, Figure 25:** Considering point G 4, does this scheme fit?
I'll stop commenting here (but have read the rest of the paper); I think there are enough points to be revised, after which the remaining parts could have changed. So I will wait for the next version of the manuscript.

The relationships shown in Sect. 4 (relating $T_r$-$T_{top}$ maxima to $D_e$ maxima) have now been evaluated separately for in situ cirrus and WBC clouds. While the cloud thickness differs considerably between these cloud types, the relationships observed for WBC clouds still appear applicable to in situ cirrus, although more explanation is needed to see this. This section has been largely rewritten.

**References:**
Costa, A., Meyer, J., Afchine, A., Luebke, A., Gunther, G., Dorsey, J. R., Gallagher, M. W., Ehrlich, A., Wendisch, M., Baum gardner, D., Wex, H., and Kramer, M.: Classification of Arctic, midlatitude and tropical clouds in the mixed-phase temperature regime, Atmospheric Chemistry and Physics, 17, 12 219–12 238, https://doi.org/10.5194/acp-17-12219-2017, 2017.

Jensen, E. J., Karcher, B., Woods, S., Kramer, M., & Ueyama, R. (2024). The impact of gravity waves on the evolution of tropical anvil cirrus microphysical properties. Journal of Geophysical Research: Atmospheres, 129, e2023JD039887. https://doi.org/10.1029/2023JD039887

Wernli, H., M. Boettcher, H. Joos, A. K. Miltenberger, and P. Spichtinger (2016), A trajectory-based classification of ERA-Interim ice clouds in the region of the North Atlantic storm track, Geophys. Res. Lett., 43, 6657–6664, doi:10.1002/2016GL068922.

Table 1. Notations used for CALIOP retrievals of cirrus clouds.

| Notation | Definition or interpretation |
|---|---|
| $T_{top}$, $T_{base}$ | Temperature at cloud top, temperature at cloud base |
| $T_r$ | Cloud layer radiative temperature, on average in the middle between $T_{top}$ and $T_{base}$. |
| $\tau$ | Visible cloud optical depth |
| IAB | CALIOP 532 nm layer integrated attenuated backscatter |
| Thin cirrus | IAB < 0.01 sr$^{-1}$ $\sim 0.01 < \tau < \sim 0.3$ : optically thin cirrus including some subvisible cirrus* |
| Thick cirrus | IAB >0.01 sr$^{-1}$ $\sim 0.3 < \tau < \sim 3$ : optically thick cirrus, but semi-transparent to the lidar |
| All cirrus | $\sim 0.01 < \tau < \sim 3$ ; includes optically thin and thick cirrus clouds |

*subvisible cirrus ($\tau < 0.03$)

---

## Referee Report (RR1)

**Second review of**

Advances in CALIPSO (IIR) cirrus cloud property retrievals – Part2: Global estimates of the fraction of cirrus clouds affected by homogeneous ice nucleation

David L. Mitchell and Anne Garnier

**General:**

The manuscript has improved considerably. The authors put in a lot of work and incorporated almost all of the suggestions, for example, the Figures in the new version are much much better! However, there are still a few points for discussion regarding the previous comments and some new recommendations, which are all listed below.

**Old G 4** Hom-affected and het-only cirrus clouds (Sections 3.2 and 3.3 etc.)

E.g. Line 293ff: Characteristic in all plots (Figure 12 and subsequent figures) is a broad region on the left side (relatively low  $\alpha$ ext) where Ni < 30 L -1, apparently corresponding to het only. To the right of this region is a gradient of increasing N i, culminating in values of Ni >2000 L-1. This gradient region is likely produced by varying degrees of hom activity.

I am not convinced by this classification and would interpret this central point of the paper differently, as I will explain in the following.

I agree that there are two cirrus regimes, as described in Krämer et al. (2016, 2020). Here  $\alpha_{ext}$  (color coded by  $N_{ij}$  is used to make this visible. The region of high  $\alpha_{ext}$  and  $N_{i}$  (,hom acitivity') corresponds approximately to the area of high IWC and  $N_{i}$  in Fig. 6 (top panel) of Krämer et al. (2020). In this region, both in-situ and liquid-origin cirrus clouds are present. For the in-situ cirrus clouds, the interpretation that they are formed by hom (of soluble aerosol particles) is correct, but not for the liquid-origin cirrus. Hom (of cloud drops) can occur within liquid origin cirrus clouds, but is rather rare outside the tropics. **The predominant freezing mechanism of liquid origin cirrus is het**, nevertheless, they can have high  $N_{i}$ .

Dekoutsidis et al. (2023) show that RHi conducive to hom (RHi > 140%) occurs near cloud top irrespective of whether the cirrus clouds are in situ or liquid origin. This is consistent with our findings that show hom contributes to both in situ and liquid origin cirrus (LOC) clouds.

**Old G4, new comment**: this is true – hom can contribute to liquid-origin cirrus in a second step: liquid-origin cirrus enter the cirrus temperature range completely glaciated by het freezing. When RHi rises again up to the homogeneous freezing threshold due to further ascent, a new (in situ!) ice nucleation event occurs.

I recommend to explain this in your paper, otherwise readers might be confused about homogeneous freezing in liquid-origin cirrus (which can be only drop freezing at 235 K).

By the way, these liquid-origin cirrus are a special case, a mixture between in situ and liquid-origin cirrus. This is discussed in a new article by Gasparini et al. (2025) https://egusphere.copernicus.org/preprints/2025/egusphere-2025-203/ (their Figure 2)

and these cirrus clouds are referred to as an 'dual-origin'. Maybe you want to mention this in your paper.

The region with low  $\alpha$ ext and Ni is defined here as ,het only'. However, I think the composition of the cirrus clouds in this region are much more complex. First to mention, in this region there are also both in-situ and liquid origin cirrus present. Further, the in-situ origin cirrus could have formed either hom or het, since hom also produces only few ice crystals at warm temperatures and low updrafts.

**Agreed. We acknowledge this in text added to Sect. 3.2 that is given next in italics.**

But, most importantly, the concentration of hom cirrus with initially high N i (and thus  $\alpha$ ex) decreases quite rapidly in the warming phases of the ubiquitous mesoscale temperature fluctuations where the environment is subsaturated (Jensen et al., 2024). This means that they are moving from the hom affected regime to what is now defined as het only. This can be seen also in Fig. 6 (bottom panel) of Kramer et al. (2020) - the thinner the cirrus (and the lower N i) the more frequent the cirrus clouds are in a subsaturated environment.

The fact that the **cirrus clouds with low αext and Ni are in a subsaturated environment** is also indicated by the decreasing De to the left of D max (Figures 15 and 16), because under this condition, the thinner the cirrus clouds and the lower the Ni, the smaller the ice particles. In an at least saturated or supersaturated environment, the ice particles would be larger with decreasing Ni, i.e. there would be no maximum in De, but an increase, maybe with a change of the slope during the transition from one to the other regime.

**Old G4 Reiteration**: My comment was:**

,In summary, I believe that this region is a mixture of in-situ origin cirrus clouds of different ages, which could have formed either het or hom, and aged liquid origin in the dissolution stage.

I recommend reconsiding the naming and the discussion of the 'het only' cirrus regime.'

It is now mentioned in manuscript that hom might als produce a low number of ice crystals. However, that the cirrus with low  $\alpha$ ext and Ni coud also be aged liquid origin or dissolving cirrus is missing.

The naming of this cirrus type has changed from 'het only' to ,het'. However, in all discussions, the impression is still that these cirrus originated purely heterogeneously. While it would be ideal if the cirrus types could be clearly separated, unfortunately, this is not the case. Therefore, I recommend that this cirrus class be better described and named (maybe ,het-mixed'?).

**I also wrote:**

`What I wonder (although I know it would be a lot of work) is whether this analysis would be better done separately for in-situ and liquid origin cirrus (the derivations presented in section 3.2 only apply to in-situ cirrus anyway)? Especially for in-situ cirrus, the interpretation of the freezing mechanisms would be much clearer, now the liquid origin probably blurs their features.'

Any thoughts about this?

We thank the reviewer for making us aware of the ambiguity of our results concerning Fig. 12 of the preprint (now Fig. 7). This figure has been revised to show these results for both in situ cirrus and WBC clouds during winter only. It is seen that the results are very similar for both

cloud types, with a strong Ni gradient when related to extinction. Assuming WBC are mostly LOC, this indicates that hom is also important for LOC.

As indicated in our responses to Reviewer 1, the freezing mechanisms are now discussed and clarified, with hom ice nucleation in both in situ and WBC clouds proceeding primarily through homogeneous freezing of solution haze droplets. We agree that hom under warmer, low updraft conditions can also produce  $N_i < 30~L^{-1}$ , and the relevant section of text has been rewritten as:

"Characteristic in all plots is a broad region on the left side (relatively low  $\alpha_{\rm ext}$ ) where  $N_i < 30$   $L^{-1}$ . Although hom can produce such low concentrations at warmer temperatures and low updrafts (Krämer et al., 2016), hom tends to produce much higher Ni (Barahona and Nenes, 2009). To the right of this region is a gradient of increasing  $N_i$ , culminating in values of  $N_i > 1000 \ L^{-1}$ . This gradient region is likely produced by varying degrees of hom activity, although het may also contribute to this gradient under conditions of relatively high INP concentration. It is evident that hom-affected cirrus clouds are common in both in situ cirrus and WBC. The main difference between these cloud types is in the tropics where in situ cirrus often appear to the right of the region predicted for pure hom (i.e., the triangles or squares) which will be discussed below. This may be due to deep convection overshooting the temperature level predicted for hom to activate, depositing moisture at lower temperatures where in situ cirrus subsequently form. Results like Fig. 7 are shown in Fig. S7 for land where this 'overshooting effect' is more evident for in situ cirrus outside the tropics, perhaps due to stronger orography-induced updrafts over land ."

In addition, the descriptor "het only" has been removed from the paper, and such clouds are now referred to as het cirrus or het dominated cirrus clouds. See new comment above.

**Old S 11: Line 440ff:** Most evident when comparing Figs. 15 and 16 for  $\alpha_{ext} < 0.3 \text{ km}^{-1}$  (where het is expected to prevail) is that median  $N_i$  is higher over land (up to afactor of 10), presumably due to higher INP concentrations over land. ...

Or stronger updrafts  $\rightarrow$  enhanced hom over land?

.... higher INP over land (which can also be enhanced by stronger updrafts) may be producing a "Twomey effect" in het cirrus clouds over land.

This is very speculative (over-interpreted?) .... to make this hypothesis more information about INP and updrafts would be necessary.

New text has been added at the end of Sect. 3.3.2 to address this concern:

"While higher updrafts over land could also enhance INP and Ni concentrations, note that updraft effects are implicit in Figs. 9, S11, and 10. That is, higher updrafts are associated with higher IWC (Hu et al., 2021; Mitchell, 1988) and higher extinction is associated with higher IWCs. The apparent Twomey effect here is associated with  $\alpha_{ext} < 0.3$  km-1 where updrafts are expected to be relatively weak over both ocean and land."

**Old S11, new comment**: It would be nice to mention that updraft and hom produced Ni (and thus IWC) are strongly correlated; updraft being the parameter that most strongly triggers Ni production, more than temperature or INP concentration (see Kaercher and Lohmann 2002, 2003; JGR). That is the reason why I am a bit persistent here...

**Old S 13: Line 550ff**: Also of interest are the seasonal changes in hom fraction between 30°N and 60°N in Fig. 20. Relative dust contributions of the world's main dust regions are ... more likely to reach cirrus cloud levels in the UT due to ascent within frontal systems, orographic uplift, and dry convection. ...

As in point S 11, seasonal changes in hom fraction might also be due to changes in updrafts and not only to be related to INP.

A new paragraph has been added below the paragraph indicated above, stating:

"It can also be argued that the above seasonal differences in the hom fraction canbe attributed to seasonal differences in vertical velocities at cirrus cloud levels. However, this appears less likely when one considers that strong orographic lifting occurs over the southern Andes Mountains during all seasons, and these vertical motions should be much greater than other vertical motions at cirrus levels in this region (excepting deep convection in summer, but the thick anvils affected by such convection are not sampled by this method). If the hom fraction changes are sensitive to changes in updraft strength, the hom fraction over the southern Andes should not change much between DJF and JJA, but it does. A similar argument can be made for the Himalayas and the Rocky Mountains. Finally, two studies (Sporre et al., 2022; Lin et al., 2025) have documented large microphysical changes in cirrus clouds that were impacted by volcanic aerosol, which are consist with this reasoning."

**Old S 13, new comment** (see also my new comment to S11): A change in  $N_i$  over mountains between DJF and JJA is also reported by Sourdeval et al. (2018). They explain  $_iN_i$  ... in the midlatitude storm tracks and orographic regions are found to be higher during winter months, consistent with stronger jets (Gryspeerdt et al., 2018a)."

What I'm only now noticing is that Figure 8 in Sourdeval et al. (2018) is not consistent with your Figure 20/14 (old/new version of the paper), where the hom fraction is very low over the Andes during winter and over India during summer. In Sourdeval et al. (2018), Ni(5um) is higher over the Andes in winter (stronger jets) and higher over India in summer (convection). High Ni(5um) strongly suggests hom freezing...

I now understand the different arguments regarding the influence of updraft or INPs, this comes from the differing impressions from the figures in Sourdeval et al. (2018) and here. I think these differences should be discussed ...

**New General 1:** you argue here and later in the manuscript that strong orographic lifting occurs over the southern Andes Mountains during all seasons. This is an important point for interpreting the results, but is not clearly documented here. This would need to be addressed, or the conclusions based on this argument should be formulated more speculatively.

**New comments (to version with tracked changes):**

**New General 2**: The number of Figures has been reduced, but the text now frequently refers to figures in the appendix or supplement. This often interrupts the flow of reading, and also refers to two different places. Where possible, I would reduce these references and also consider creating either an appendix or a supplement.

**New S 1: Line 922 ff:** 'Hom may occur through (1) the freezing of haze solution droplets (Koop et al., 2000) and (2) the freezing of supercooled cloud droplets advected across the isotherm  $\sim 235$  K (e.g., Rosenfeld and Woodley, 2000). If (1) and (2) are comparable in their frequency of occurrence, an abrupt increase in median Ni should be evident in Fig. 12 in the

231 - 235 K range (given typical non-convective cirrus updrafts of 10 to 30 cm s -1). **Since such an abrupt increase is not evident in Fig. 12,** it appears that (2) does not contribute significantly to Ni, even in the tropics.'

As can be seen in Fig. 12,  $N_i$  increases significantly towards higher temperatures in the TRO from approximately T=220~K, which is not visible in the MID and HIGH regions. I would attribute this to droplet freezing, which is also consistent, e.g. Costa et al. (2017) observe liquid droplets coexisting with ice particles at temperatures close to 235 K more frequently in TRO than in MID or HIGH.

**New S2: Figure 16:** Why is the hom fraction so low in the tropics - that's not very likely given the high updrafts there.? I think it's written somewhere... but readers here probably don't remember it.

---

## Author Response (AR2)

**Referee #1, 2nd review of Part 2 of CALIPSO paper**

**Author responses to the Reviewer 1**

Format: The reviewers' comments are in normal font while author responses are in red font. Text in red font italics indicates revised or added text in the revised manuscript.

Again, we thank the reviewers for their thoughtful comments and the considerable amount of time they invested for improving this manuscript. Below are our responses to the comments concerning their second review:

**General comments from Reviewer 1:**

Review of Mitchell and Garnier, 2024, round 2

I would like to thank the authors for addressing many of the comments. In my opinion, the manuscript is now ready for publication. The authors did a better job of condensing their figures in the revised manuscript. I believe this is high-quality, innovative work of great scientific importance that deserves immediate publication in ACP.

However, due to its length and the large number of figures and subplots, I still believe that the broader community will not fully appreciate or understand the important science presented by the authors.

Please not that I still added a number of additional minor comments that the authors may want to consider including in the manuscript:

The two studies led by Tim Lüttmer that examine the origin of ice clouds by tracing each source of ice separately in a high-resolution model could be mentioned in the introduction (or other parts of the manuscript):

Lüttmer et al., 2025, <a href="https://acp.copernicus.org/articles/25/4505/2025/">https://acp.copernicus.org/articles/25/4505/2025/</a>, see their Fig. 6 for the example of deep convection.

Lüttmer et al., 2025, in typesetting right now, preprint

at: <a href="https://egusphere.copernicus.org/preprints/2025/egusphere-2025-185/">https://egusphere.copernicus.org/preprints/2025/egusphere-2025-185/</a> for the example of a WB cirrus.

Appendix A, but maybe also lines 130–141 and other discussions related to the classification of WBC and in situ cirrus, particularly in the tropics:

In my opinion, Figure 7a shows that the WBC classification does not work very well in the

tropics if its purpose is to highlight the microphysical origin of cirrus.

There are many samples to the right of the two theoretical curves. This is less problematic for other climate bands.

One option would be to state in the Appendix that discerning the microphysical origin is beyond the scope of such a simple classification, and to justify the use of the current classification scheme.

We have classified cirrus in terms of in situ and WBC clouds because of (1) its similarity to the in situ vs. LOC classification and cloud property differences thereof and (2) the practicality of implementing this classification for CALIPSO sampling. The motivation for using this classification derives from (1) and we explored whether similar differences in cloud properties are observed. Similar differences were observed (namely IWC) but it turns out that the largest differences were IWP and  $\tau$  (which was not evident in previous studies based on in situ sampling from aircraft). This was the main finding for this classification. We agree with the reviewer that our study was not designed to assess microphysical origin, and we have added this disclaimer near the top of the  $3^{rd}$  paragraph in the Appendix: "While our cloud classification method (in situ vs. WBC) was not designed for evaluating the cloud's microphysical origin (i.e., using back-trajectory analysis) like the in situ-LOC classification was, it may be sufficiently similar to capture some of the microphysical distinctions observed between the in situ and LOC classifications."

**Figure 3:**

The authors could cut the colormap at around 80 microns. This would provide greater contrast for the most commonly encountered ice crystal sizes. In its current form, it is difficult to spot large size differences.

In the current colormap, we use 10-µm steps between 10 and 80 µm and only one color between 80 and 137 µm, to show where  $D_e$  is found larger than 80 µm while keeping sufficient resolution for  $D_e$  < 80 µm. It is not clear that cutting the colormap at around 80 µm will improve the figure.

**Lines 270-272:**

Isn't it simply the detrainment of ice formed at warmer temperatures at higher altitudes, which appears to violate the Clausius-Clapeyron limitations? I think it is incorrect to talk about 'moisture', as deep convection and freshly detrained clouds typically have a relative humidity (RH) of about 100%. What we really have there is the transport of ice from lower to higher levels.

**The word "moisture" has been replaced by "ice mass".**

Lines 286-300:

It would be useful to include the full derivation of the equations used in the appendix, particularly equations 2 and 4.

How does one derive equation 2 from the standard Clausius–Clapeyron equation? While this may sound obvious, it is still good practice to document it.

There is a very eloquent derivation of Eq. 2 in Ch. 3 of Lamb and Verlinde (2011). Therefore, we have modified the sentence introducing Eq. 2 as follows:

"Reformulating Eq. (1) as described in Sect. 3.3.1 of Lamb and Verlinde (2011) gives an expression useful in calculating the supersaturation required for homogeneous ice nucleation:"

Similarly, an intermediate step could be included to clarify equation 4, e.g. by explicitly writing the ideal gas laws: Rho = e/RvT and Rho\_(hom) =  $e_(hom)/RvT$ , Rho\_(si) =  $e_(si)/RvT$ . These derivations could go in the appendix.

The sentence introducing Eq. 4 has been modified as:

Noting that the water vapor pressure at  $S_i^f$ ,  $e_{hom}$ , is simply  $e_{hom} = S_i^f e_{si}$ , water vapor densities are obtained from the Gas Law ( $\rho_{hom} = e_{hom}/(R_v T)$  and  $\rho_{si} = e_{si}/(R_v T)$ ,  $R_v = gas$  constant for water vapor) to predict the maximum IWC resulting from hom:

Lines 324-326:

Because of close agreement, do we really need both versions in all the figures?

Because we evaluate different quantities in Fig. 7 (Ni) and in Fig. 8 (IWC and De), it makes sense to show the empirical and theoretical predictions for hom. However, in the supplement showing the equivalent of Figs. 7 and 8 but over land (Figs. S7 and S8), only the theoretical prediction for hom is used. Similarly, only one version is shown in Fig. 11, and in Fig. S12 which is the equivalent of Fig. 11 but over land.

**Section 3.3.1**

I think the language could be simplified here.

We have rewritten this section to simplify it and make it clearer.

**Figure 13:**

Is there a need for the Hom fraction axis and information in this figure? It's already plotted in Figure 12.

Including the hom fraction profile allows the reader to see how it influences the  $\tau$  difference between the het  $\tau$  profile and the overall  $\tau$  profile. This would not be very evident if the hom fraction profile was only in Fig. 12.

Also, it's tiring to fight through 12 subplots in 6 of the manuscript's figures. This clearly doesn't help convey your key messages, as it dilutes the message too much. Readers lose focus and motivation to continue.

We show these because we think there are often significant differences between latitude bands and/or seasons. Some readers may be intrigued by this. We could have submitted to Science or Nature, but we chose ACP since they allow more research findings to be published.

**Note on some of the figures:**

It might be better to show NH-JJA and SH-DJF close to each other because we want to be able to compare summers and winters in both hemispheres.

We think that the current figure arrangements allow readers to do this adequately.

**Lines 575-576:**

I think it's clear that homogeneous drop freezing isn't the main source of ice in WBC. The fact that most GCMs simulate this phenomenon is likely due to the biases in numerical models, especially low-resolution GCMs.

The Lüttmer et al., 2025 study (<a href="https://egusphere.copernicus.org/preprints/2025/egusphere-2025-185/">https://egusphere.copernicus.org/preprints/2025/egusphere-2025-185/</a>) may be a more updated and trustworthy modeling reference.

Even after considering one of the key figures (Figure 16, specifically the COD-weighted part), there's still a lot of content in Figures 17 (again, 12 (!) subplots) and 18. I know that this is important for proposing the cirrus cloud formation and evolution idea and the Figure 19, but that could also be published e.g. in a letter format, as a separate publication.

We think that the context in which the cirrus cloud characterization idea is presented adds significantly to the concept. Therefore, we prefer not to separate this concept from the paper.

Figure 20b should include a legend. Additionally, the cloud scene in the photo could be highlighted in the satellite image.

A red "X" was added in the satellite image to highlight the location of the cloud scene in the photo.

The caption for Fig. 20b (right) was amended as follows: "Low clouds are white-grey while high clouds (e.g., blue colors) tend to have lower brightness temperatures (BT), with lowest BT in dark blue and green. The red "X" indicates the location of the cloud scene in the photo."

**Author responses to the Second Review from Referee #2**

The format below is identical to that used by Referee #2 in her/his second review, except that our **responses to this second review are in green font**. That is, the first review from Referee #2 is in black font, followed by our responses in red, followed by the second review from Referee #2 in blue, followed by our second author responses in green. New text added is in italics (green font).

**Second review of**

Advances in CALIPSO (IIR) cirrus cloud property retrievals – Part2: Global estimates of the fraction of cirrus clouds affected by homogeneous ice nucleation

David L. Mitchell and Anne Garnier

**General:**

The manuscript has improved considerably. The authors put in a lot of work and incorporated almost all of the suggestions, for example, the Figures in the new version are much much better! However, there are still a few points for discussion regarding the previous comments and some new recommendations, which are all listed below.

**Old G 4** Hom-affected and het-only cirrus clouds (Sections 3.2 and 3.3 etc.)

E.g. Line 293ff: Characteristic in all plots (Figure 12 and subsequent figures) is a broad region on the left side (relatively low  $\alpha$ ext) where Ni < 30 L -1, apparently corresponding to het only. To the right of this region is a gradient of increasing N i, culminating in values of Ni >2000 L-1. This gradient region is likely produced by varying degrees of hom activity.

I am not convinced by this classification and would interpret this central point of the paper differently, as I will explain in the following.

I agree that there are two cirrus regimes, as described in Kramer et al. (2016, 2020). Here  $\alpha_{ext}$  (color coded by Ni) is used to make this visible. The region of high  $\alpha_{ext}$  and Ni (hom acitivity') corresponds approximately to the area of high IWC and Ni in Fig. 6 (top panel) of Kramer et al. (2020). In this region, both in-situ and liquid-origin cirrus clouds are present. For the in-situ cirrus clouds, the interpretation that they are formed by hom (of soluble aerosol particles) is correct, but not for the liquid-origin cirrus. Hom (of cloud drops) can occur within liquid origin cirrus clouds, but is rather rare outside the tropics. **The predominant freezing mechanism of liquid origin cirrus is het**, nevertheless, they can have high Ni.

Dekoutsidis et al. (2023) show that RHi conducive to hom (RHi > 140%) occurs near cloud top irrespective of whether the cirrus clouds are in situ or liquid origin. This is consistent with our findings that show hom contributes to both in situ and liquid origin cirrus (LOC) clouds.

**Old G4, new comment**: this is true – hom can contribute to liquid-origin cirrus in a second step: liquid-origin cirrus enter the cirrus temperature range completely glaciated by het freezing. When RHi rises again up to the homogeneous freezing threshold due to further ascent, a new (in situ!) ice nucleation event occurs.

I recommend to explain this in your paper, otherwise readers might be confused about homogeneous freezing in liquid-origin cirrus (which can be only drop freezing at 235 K).

By the way, these liquid-origin cirrus are a special case, a mixture between in situ and liquid origin cirrus. This is discussed in a new article by Gasparini et al. (2025) https://egusphere.copernicus.org/preprints/2025/egusphere-2025-203/ (their Figure 2) and these cirrus clouds are referred to as an 'dual-origin'. Maybe you want to mention this in your paper.

Figure 2 of Gasparini et al. (2025) shows dual-origin cirrus mostly near the top of a simulated tropical anvil cloud, where in-situ nucleation has occurred more recently than detrainment. Figure 7 of our revised paper (May version) is shown for in situ cirrus clouds (panels a-c) and for warm-base cirrus clouds (WBC, panels d-f) and indicates that hom is very active in WBCs even at the warmest cirrus radiative temperatures. This can be understood if the cloud is glaciated by het freezing at T > 235 K and supersaturation  $s_i$  is already near the hom threshold. Then some cloud parcels are transported upwards across the 235 K isotherm with haze solution droplets freezing as they cross this isotherm. We have added new text to make this clear at Line 285: "For WBC, hom appears quite active even at the highest temperatures. This can be understood if the cloud is glaciated by het freezing at T > 235 K and ice supersaturation  $s_i$  is already near the hom threshold. Then some cloud parcels are transported upwards across the 235 K isotherm, increasing  $s_i$ , with haze solution droplets freezing as they cross this isotherm."

The region with low aext and Ni is defined here as 'het only'. However, I think the composition of the cirrus clouds in this region are much more complex. First to mention, in this region there are also both in-situ and liquid origin cirrus present. Further, the in-situ origin cirrus could have formed either hom or het, since hom also produces only few ice crystals at warm temperatures and low updrafts.

**Agreed. We acknowledge this in text added to Sect. 3.2 that is given next in italics.**

But, most importantly, the concentration of hom cirrus with initially high Ni (and thus  $\alpha$ ext) decreases quite rapidly in the warming phases of the ubiquitous mesoscale temperature fluctuations where the environment is subsaturated (Jensen et al., 2024). This means that they are moving from the hom affected regime to what is now defined as het only. This can be seen also in Fig. 6 (bottom panel) of Kramer et al. (2020) - the thinner the cirrus (and the lower N i) the more frequent the cirrus clouds are in a subsaturated environment.

The fact that the **cirrus clouds with low aext and Ni are in a subsaturated environment** is also indicated by the decreasing De to the left of Dmax (Figures 15 and 16), because under this condition, the thinner the cirrus clouds and the lower the Ni, the smaller the ice particles. In an at least saturated or supersaturated environment, the ice particles would be larger with decreasing Ni, i.e. there would be no maximum in De, but an increase, maybe with a change of the slope during the transition from one to the other regime.

**Old G4 Reiteration:** My comment was:**

'In summary, I believe that this region is a mixture of in-situ origin cirrus clouds of different ages, which could have formed either het or hom, and aged liquid origin in the dissolution stage. I recommend reconsiding the naming and the discussion of the 'het only' cirrus regime.'

It is now mentioned in manuscript that hom might also produce a low number of ice crystals. However, that the cirrus with low aext and Ni coud also be aged liquid origin or dissolving cirrus is missing.

The naming of this cirrus type has changed from 'het only' to ,het'. However, in all discussions, the impression is still that these cirrus originated purely heterogeneously. While it would be ideal if the cirrus types could be clearly separated, unfortunately, this is not the case. Therefore, I recommend that this cirrus class be better described and named (maybe 'het-mixed'?).

We agree with this concern and we now define the het category to include dissipating cirrus clouds. Most of the first paragraph of Sect. 3.3 has been rewritten as: "To address this question, a method is presented that estimates the fraction of hom-affected cirrus clouds such that these clouds are strongly altered microphysically relative to cirrus associated with relatively low  $N_i$ , such as cirrus formed through het and aged, dissipating/sublimating cirrus clouds (formed either through hom or het). Henceforth the former will be referred to as hom cirrus and the latter as het cirrus. These dissipating cirrus clouds are microphysically described in Fig. 6 of Krämer et al. (2020) where  $RH_i < 100\%$ . These two categories will now be described and finally quantified."

**I also wrote:**

'What I wonder (although I know it would be a lot of work) is whether this analysis would be better done separately for in-situ and liquid origin cirrus (the derivations presented in section 3.2 only apply to in-situ cirrus anyway)? Especially for in-situ cirrus, the interpretation of the freezing mechanisms would be much clearer, now the liquid origin probably blurs their features.'

Any thoughts about this?

We followed the suggestion and separated in situ and WBC clouds in Fig. 7 (Sect. 3.2). Previously, these two cloud types were lumped together in this figure. On line 268 (May version), it states: "It is evident that hom-affected cirrus clouds are common in both in situ cirrus and WBC."

We also followed this suggestion for Fig. 9 (oceans) and Fig. 10 (land), which are for in situ cirrus clouds only, while the results for WBC are shown in Figs. S9 and S10, respectively. However, we did not comment clearly on the similarities between the in situ and WBC results. This sentence is now added in the second paragraph of Sect. 3.3: "Note that these observations apply to both in situ and WBC clouds, suggesting that similar physics applies to both cloud types."

We also followed this suggestion for Fig. 18.

We thank the reviewer for making us aware of the ambiguity of our results concerning Fig. 12 of the preprint (now Fig. 7). This figure has been revised to show these results for both in situ cirrus and WBC clouds during winter only. It is seen that the results are very similar for both cloud types, with a strong Ni gradient when related to extinction. Assuming WBC are mostly LOC, this indicates that hom is also important for LOC.

As indicated in our responses to Reviewer 1, the freezing mechanisms are now discussed and clarified, with hom ice nucleation in both in situ and WBC clouds proceeding primarily through homogeneous freezing of solution haze droplets. We agree that hom under warmer, low updraft conditions can also produce  $N_i < 30 L_{-1}$ , and the relevant section of text has been rewritten as:

"Characteristic in all plots is a broad region on the left side (relatively low  $\alpha_{\text{ext}}$ ) where  $N_i < 30$  L-1. Although hom can produce such low concentrations at warmer temperatures and low updrafts (Kramer et al., 2016), hom tends to produce much higher Ni (Barahona and Nenes, 2009). To the right of this region is a gradient of increasing  $N_i$ , culminating in values of  $N_i > 1000$  L-1. This gradient region is likely produced by varying degrees of hom activity, although het may also contribute to this gradient under conditions of relatively high INP concentration. It is evident that hom-affected cirrus clouds are common in both in situ cirrus and WBC. The main difference between these cloud types is in the tropics where in situ cirrus often appear to the right of the region predicted for pure hom (i.e., the triangles or squares) which will be discussed below. This may be due to deep convection overshooting the

temperature level predicted for hom to activate, depositing moisture at lower temperatures where in situ cirrus subsequently form. Results like Fig. 7 are shown in Fig. S7 for land where this 'overshooting effect' is more evident for in situ cirrus outside the tropics, perhaps due to stronger orography-induced updrafts over land."

In addition, the descriptor "het only" has been removed from the paper, and such clouds are now referred to as het cirrus or het dominated cirrus clouds. See new comment above.

This has been addressed as described above.

**Old S 11: Line 440ff:** Most evident when comparing Figs. 15 and 16 for  $\alpha_{\text{ext}} < 0.3 \text{ km}_{-1}$  (where het is expected to prevail) is that median  $N_i$  is higher over land (up to a factor of 10), presumably due to higher INP concentrations over land. ...

Or stronger updrafts → enhanced hom over land?

.... higher INP over land (which can also be enhanced by stronger updrafts) may be producing a "Twomey effect" in het cirrus clouds over land.

This is very speculative (over-interpreted?) .... to make this hypothesis more information about INP and updrafts would be necessary.

New text has been added at the end of Sect. 3.3.2 to address this concern: "While higher updrafts over land could also enhance INP and Ni concentrations, note that updraft effects are implicit in Figs. 9, S11, and 10. That is, higher updrafts are associated with higher IWC (Hu et al., 2021; Mitchell, 1988) and higher extinction is associated with higher IWCs. The apparent Twomey effect here is associated with  $\alpha_{\rm ext} < 0.3$  km-1 where updrafts are expected to be relatively weak over both ocean and land."

**Old S11, new comment**: It would be nice to mention that updraft and hom produced Ni (and thus IWC) are strongly correlated; updraft being the parameter that most strongly triggers Ni production, more than temperature or INP concentration (see Kaercher and Lohmann 2002, 2003; JGR). That is the reason why I am a bit persistent here...

All that is true, but the mentioned correlations between hom Ni vs. updraft and hom IWC vs. updraft apply to the hom regime. This discussion about the Twomey effect in Sect. 3.3.2 applies only to the het regime. The first sentence of Sect. 3.3.2 has been slightly modified to make this more clear: "Most evident when comparing Figs. 9 and 10 (or Figs. S9 and S10) for  $\alpha_{\rm ext} < 0.3 \, {\rm km}^{-1}$  (i.e., the het cirrus regime) is that median  $N_i$  is higher over land (up to a factor of 10), presumably due to higher INP concentrations over land."

**Old S 13: Line 550ff**: Also of interest are the seasonal changes in hom fraction between 30°N and 60°N in Fig. 20. Relative dust contributions of the world's main dust regions are ...

more likely to reach cirrus cloud levels in the UT due to ascent within frontal systems, orographic uplift, and dry convection. ...

As in point S 11, seasonal changes in hom fraction might also be due to changes in updrafts and not only to be related to INP.

A new paragraph has been added below the paragraph indicated above, stating: "It can also be argued that the above seasonal differences in the hom fraction canbe attributed to seasonal differences in vertical velocities at cirrus cloud levels. However, this appears less likely when one considers that strong orographic lifting occurs over the southern Andes Mountains during all seasons, and these vertical motions should be much greater than other vertical motions at cirrus levels in this region (excepting deep convection in summer, but the thick anvils affected by such convection are not sampled by this method). If the hom fraction changes are sensitive to changes in updraft strength, the hom fraction over the southern Andes should not change much between DJF and JJA, but it does. A similar argument can be made for the Himalayas and the Rocky Mountains. Finally, two studies (Sporre et al., 2022; Lin et al., 2025) have documented large microphysical changes in cirrus clouds that were impacted by volcanic aerosol, which are consist with this reasoning."

**Old S 13, new comment** (see also my new comment to S11): A change in Niover mountains between DJF and JJA is also reported by Sourdeval et al. (2018). They explain "Ni... in the midlatitude storm tracks and orographic regions are found to be higher during winter months, consistent with stronger jets (Gryspeerdt et al., 2018a)."

What I'm only now noticing is that Figure 8 in Sourdeval et al. (2018) is not consistent with your Figure 20/14 (old/new version of the paper), where the hom fraction is very low over the Andes during winter and over India during summer. In Sourdeval et al. (2018), Ni(5um) is higher over the Andes in winter (stronger jets) and higher over India in summer (convection). High Ni(5um) strongly suggests hom freezing...

In our Fig. 14, the hom fractions over the southern Andes (south of about 15 °S) are higher in winter (JJA) than in summer (DJF) and  $N_i$  in Fig. 2 is also higher in winter than in summer, consistent with the top two panels of Fig. 8 of Sourdeval et al. (2018) which are for temperatures between -60 and -50 °C. North of 15° °S in south America,  $N_i$  in Fig. 2 is larger in DJF (summer) than in JJA (winter), consistent with Sourdeval et al. However, the hom fractions in Fig.14 are smaller in summer than in winter, because the search for maximum  $D_e$  yielded larger extinction thresholds in DJF than in JJA (see panels d and j of Fig. S12) and as a result, hom fractions are lower in summer than in winter.

In the tropics, the differences between  $N_i$  from our work and from Sourdeval et al. (2018) is explained by the fact that we do not sample most of the convective clouds because our retrieval is limited to clouds having optical depth  $\leq$  3. We believe that this explains why  $N_i$  in Fig. 2 is smaller than in Sourdeval et al. (2018) in some areas. Indeed, Sourdeval et al. (2018) state that "Strong variations are found in the tropics and along the ITCZ, where large cloud structures are

convectively driven. High Ni values are noted in these regions during summer seasons, which strengthens their link to freezing events associated with deep convective structures. These values typically decrease by a factor of 2–3 during winter seasons". This is why in the first paragraph of Sect. 2.3, we wrote: "The seasonal dependence of  $N_i$  in Fig. 2 was also found by Sourdeval et al. (2018) at mid-to-high latitudes for winter vs. summer." For clarity, we added the following sentence: "The regions with large  $N_i$  in the tropics in Sourdeval et al. (2018) are absent in Fig. 2 possibly because thick cirrus here have  $\tau < \sim 3$  and the deep convective cores (where hom is often active) are not sampled".

Hom fractions over India are smaller in summer (JJA) than in winter (DJF), which is again due to the larger extinction thresholds in summer than in winter (see panels a and g of Fig. S12).

I now understand the different arguments regarding the influence of updraft or INPs, this comes from the differing impressions from the figures in Sourdeval et al. (2018) and here. I think these differences should be discussed ...

**New General 1:** you argue here and later in the manuscript that strong orographic lifting occurs over the southern Andes Mountains during all seasons. This is an important point for interpreting the results, but is not clearly documented here. This would need to be addressed, or the conclusions based on this argument should be formulated more speculatively.

Thank you for your persistence. We investigated this more, and the article by Hoffmann et al. (2016) shows strong evidence for higher winds at cirrus levels during winter over the Southern Ocean. The paragraph in question has been rewritten as:

"It can also be argued that the above seasonal differences in the hom fraction can be attributed to seasonal differences in vertical velocities at cirrus cloud levels. This appears likely over the Southern Ocean when one considers the stronger orographic gravity waves occurring there during the winter season (Jiang et al., 2002; Hoffmann et al., 2016). Moreover, Gryspeerdt et al. (2018) note that  $N_i$  is higher during winter in the midlatitude storm tracks, consistent with stronger jets. Nonetheless, two studies (Sporre et al., 2022; Lin et al., 2025) have documented large microphysical changes in cirrus clouds that were impacted by volcanic aerosol, which suggests that the observed seasonal differences can be attributed in part to seasonal differences in INP concentrations."

**New comments (to version with tracked changes):**

**New General 2**: The number of Figures has been reduced, but the text now frequently refers to figures in the appendix or supplement. This often interrupts the flow of reading, and also refers to two different places. Where possible, I would reduce these references and also consider creating either an appendix or a supplement.

The purpose of the Supplement was generally to reinforce the findings presented in the main manuscript. This was done by showing maps for additional seasons, maps for in situ clouds only ( $N_i$  and  $D_e$ ; contasting these with maps combining WBC and in situ cirrus), results for WBC clouds (for comparison with in situ results shown in the main manuscript), and figures over land (for comparison with similar figures over ocean). The main purpose of the Appendix was not to reinforce findings but to present new findings concerning the microphysical and macrophysical differences between in situ and WBC clouds. There has been considerable interest among cirrus cloud investigators regarding such differences, but we wanted to offer another perspective. Therefore, we placed our findings on this topic in an appendix so this would not distract the reader from the theme of our manuscript, while also serving the interest of the readership concerning these microphysical and macrophysical differences. To merge the supplement and appendix into a single document (either an appendix or supplement) would confuse the purposes for which they were created. We have reviewed many papers for ACP, and our supplement that has 12 figures is comparable to other supplements published in ACP (consider, for example, the recently published cirrus cloud paper by Ngo et al. (2025) that has 12 figures in its supplement).

We think that it is important to make the reader aware that more details or more figures are available in the manuscript itself or in the Supplement; hence references were provided accordingly.

We agree that the text in the last paragraph of Sect. 2.3 contained excessive references to Appendix A and specific panels in Appendix A, which interrupted the flow of reading unnecessarily. Such references have been reduced, and the modified section is given here: "This suggests that the  $D_e$  differences seen between Fig. 3 (for thick WBC & in situ cirrus) and the thick in situ cirrus in Fig. S6 (residing at a lower  $T_r$  on average) are due to temperature differences. Temperature could also be a factor in explaining the lower  $N_i$  in Fig. 2 (for thick WBC & in situ cirrus) relative to the thick in situ clouds in Fig. S5 since the later tend to form at lower temperatures (Fig. A2) and hom depends strongly on temperature (e.g., Lamb and Verlinde, 2011). Figure A3 shows that both the geometric thickness and the equivalent thickness seen by IIR are on average greater for WBC clouds than for in situ cirrus clouds. This is the main reason that IWP and  $\tau$  tend to be substantially greater for WBC clouds. Finally, IWC and  $N_i$  are both higher in WBC, in agreement with Luebke et al. (2016), only when clouds with  $\tau < \sim 0.3$  are included (blue curves)."

**New S 1: Line 922 ff:** 'Hom may occur through (1) the freezing of haze solution droplets (Koop et al., 2000) and (2) the freezing of supercooled cloud droplets advected across the isotherm ~ 235 K (e.g., Rosenfeld and Woodley, 2000). If (1) and (2) are comparable in their frequency of occurrence, an abrupt increase in median Ni should be evident in Fig. 12 in the 231 – 235 K range (given typical non-convective cirrus updrafts of 10 to 30 cm s -1). **Since such an abrupt increase is not evident in Fig. 12,** it appears that (2) does not contribute significantly to Ni, even in the tropics.'

As can be seen in Fig. 12,  $N_i$  increases significantly towards higher temperatures in the TRO from approximately T = 220 K, which is not visible in the MID and HIGH regions. I would attribute this to droplet freezing, which is also consistent, e.g. Costa et al. (2017) observe liquid droplets coexisting with ice particles at temperatures close to 235 K more frequently in TRO than in MID or HIGH.

**Notes:** Yes, in the tropics, the median  $N_i$  associated with hom cirrus does increase with increasing radiative temperature for  $T_r > 215$  K, but it applies only to a very small percentage of samples (2 % at 231-235K and less than 7 % at 219-223 K)) as shown by the hom fraction curves in the tropics. These hom fraction curves tend to increase with increasing  $T_r$ , which cloud droplet freezing does not explain.

This paragraph has been rewritten (mostly at the end) as: "Hom may occur through (1) the freezing of haze solution droplets (Koop et al., 2000) and (2) the freezing of supercooled cloud droplets advected across the isotherm ~ 235 K (e.g., Rosenfeld and Woodley, 2000). If (1) and (2) are comparable in their frequency of occurrence, an abrupt increase in median all cirrus  $N_i$  should be evident in Fig. 12 in the 231 – 235 K range (given typical non-convective cirrus updrafts of 10 to 30 cm s -1). Since such an abrupt increase is not evident in Fig. 12, it appears that (2) does not contribute significantly to  $N_i$  even in the tropics. This is consistent with Avery et al. (2020), Costa et al. (2017), and Mitchell and d'Entremont (2012) where it was shown that liquid water is rare in clouds over the range 239 – 235 K. This applies to both in situ cirrus and WBC clouds and is consistent with the definition of LOC in Luebke et al. (2016) where it is stated that LOC are restricted to pure ice clouds having T < 250 K (-23° C). When considering only hom cirrus  $N_i$ , an increase in hom  $N_i$  with  $T_r$  is seen in the tropics for  $T_r > 215 \text{ K}$ , which might be attributed to (2), but it applies to only 2 % of the sampled cirrus at 231-235 K, as shown by the hom fraction curve."

**New S2: Figure 16:** Why is the hom fraction so low in the tropics - that's not very likely given the high updrafts there.? I think it's written somewhere... but readers here probably don't remember it.

The explanation is that we do not sample convective columns because we only sample cirrus having optical depth smaller than 3. See response to "Old S 13, new comment".